# Reconstruction of the lymphatic system by transplantation of a centrifuge-based bioengineered lymphatic tissue

Shu Obana[1], Shoko Itakura [2], Mutsunori Murahashi [3], Makiya Nishikawa [2] & Kosuke Kusamori [1]✉

The increase in cancer incidence has accelerated the need for secondary lymphedema treatments after lymphadenectomy (LD) because lymph nodes cannot be regenerated. We demonstrate that bioengineered tissues with a lymphatic network containing lymphatic endothelial cells and mesenchymal stem/stromal cells (MSCs) fabricated by a centrifugal cell stacking technique effectively treat secondary lymphedema. Centrifuge-based bioengineered lymphatic tissues (CeLyTs) with MSCs outside the tissue, prepared using mouse or human cells, survive long after transplantation and restore lymphatic flow in LD mice. CeLyTs transplanted into LD mice form a lymph node-like structure and suppress lymphedema in LD mice for 100 days post-transplantation, in contrast to conventional standard treatments including compression therapy. Lymph node-like structures composed of transplant- and host-derived cells, including immune cells, generate immune responses to an immunostimulant CpG1018. Here we show CeLyTs composed of lymphatic endothelial cells and MSCs reconstruct a lymph node and may represent a promising therapy for secondary lymphedema.

Lymphatic vessels, which are mainly composed of lymphatic endothelial cells (LECs), connect hundreds of lymphoid organs, transport tissue metabolites, and peripheral antigens[1,2]. Lymph nodes are the organs responsible for immune surveillance and immunomodulatory functions and are connected by numerous lymphatic vessels[3]. Dysfunction of lymphatic vessels caused by congenital mutations or cancer treatment results in primary and secondary lymphedema, respectively[4–6]. Secondary lymphedema is often caused by lymphadenectomy (LD) during the surgical treatment of cancer and parasitic infection related to filariasis[7,8]. Because lymph nodes do not regenerate once removed, a patient's quality of life is severely impaired by recurrent complications, including interstitial fluid retention and limb cellulitis[9]. Compression therapy, which is the gold standard treatment[9], liposuction[10], manual lymphatic drainage[11], and lymphatic anastomosis are treatment options for secondary lymphedema, but their edema-suppressing effect is limited[12,13]. Additionally, autologous lymph node transplantation for lymph node reconstruction and vascular endothelial growth factor (VEGF)-C gene therapy has been attempted, but there is insufficient evidence of a therapeutic effect[14,15].

Cellular medicine is an innovative cell-based therapy that has demonstrated therapeutic potential for intractable diseases. For secondary lymphedema, mesenchymal stem cell (MSC) transplantation has been performed in medical institutions and has received much attention as a solution for lymphedema. However, there is no evidence of lymphatic regeneration or a therapeutic effect on lymphedema[16,17]. Recently, many attempts have been made to treat secondary lymphedema by forming lymphatic vessels using three-dimensional cellular structures[18–20]. Of these, three-dimensional cellular structures composed of LECs and fibroblasts fabricated using a cell stacking technique by coating functional proteins on the cell surface were

[1]Laboratory of Cellular Drug Discovery and Development, Faculty of Pharmaceutical Sciences, Tokyo University of Science, Katsushika, Tokyo, Japan. [2]Laboratory of Biopharmaceutics, Faculty of Pharmaceutical Sciences, Tokyo University of Science, Katsushika, Tokyo, Japan. [3]Division of Oncology, Research Center for Medical Sciences, The Jikei University School of Medicine, Tokyo, Japan. ✉e-mail: kusamori@rs.tus.ac.jp

reported to form a lymphatic network inside the structures, demonstrating the formation of a lymphatic lumen structure after transplantation in mice[21,22]. However, the cellular coating process with functional proteins is quite complicated and their immunogenicity may be a major problem for clinical applications. Furthermore, this cellular structure has not been effective for the treatment of secondary lymphedema. Therefore, lymph node regeneration or reconstruction using therapeutic cells has not been achieved, and the development of a better therapeutic method is desired.

This study aims to develop a bioengineered three-dimensional tissue composed of LECs and MSCs, which has immunomodulatory functions and can prolong the survival of transplants[23–25] for lymph node reconstruction. To fabricate the bioengineered tissue simply, we establish a centrifugal cell stacking technique with no additives, including functional proteins and complicated handling (Fig. 1). This bioengineered tissue, termed "centrifuge-based bioengineered lymphatic tissue" (CeLyT), forms a lymphatic network inside the tissue during culture for several days. CeLyTs induce the formation of lymph node-like structures, with characteristics similar to lymph nodes, after transplantation into mice, and the formation of this lymph node-like structure suppress edema in LD mice. Therefore, CeLyTs composed of LECs and MSCs might be a cell-based therapeutic strategy for secondary lymphedema.

## Results

### Fabrication and characteristics of CeLyTs

To fabricate the CeLyT, we optimized the centrifugal conditions for cells after seeding on the insert of a Transwell plate by observing the cells using a fluorescence microscope. Under the evaluated centrifugation conditions (centrifugation speed from $50 \times g$ to $1000 \times g$ and centrifugation time from 10 s to 3 min), seeded green fluorescent protein (GFP)-expressing mouse MSC C3H10T1/2 cells (mMSCs) settled down uniformly and reproducibly on the bottom of the insert of the Transwell plate at a speed of $750 \times g$ and a time of 30 s

(Supplementary Fig. 1a, b). In addition, these conditions stacked mMSC/GFP cells uniformly and reproducibly regardless of the number of centrifugations (Fig. 2a). Thus, we termed the centrifugation-based cell stacking process a centrifugal cell stacking technique. Then, we fabricated a three-layered bioengineered tissue composed of LECs (internal) and MSCs (external) by stacking MSCs, LECs, and MSCs in this order using the centrifugal cell stacking technique. MSCs were used because they support and prolong the survival of a transplant[23–25], and they should be located outside of the bioengineered tissue. The fabricated bioengineered tissues were cultured for 5 days to form a lymphatic network in the tissue with reference to a previous report[22]. Figure 2b, c shows an image of the bioengineered tissue from the top of the well and a microscopic cross-sectional image of the tissue (hLEC +mMSC tissue and hLEC+hMSC tissue) fabricated using primary human LECs (hLECs) and mMSCs or primary human MSCs (hMSCs). These images show that hLEC+MSC tissues had a uniform sheet-like structure of 30–50 μm thickness. To examine differences between hLEC+MSC tissues prepared by the centrifugal cell stacking technique and mMSC tissues or hLEC/mMSC mixed tissues prepared by single seeding without centrifugation, the lymphatic endothelial cell markers Prox-1 and VEGF receptor (VEGFR)−3 were detected in these tissues by immunofluorescence staining. Significantly, the expressions of Prox-1 and VEGFR-3 in the bioengineered tissues prepared by the centrifugal cell stacking technique were higher than those in other bioengineered tissues (Fig. 2d and Supplementary Fig. 1f). Of note, the mouse fibroblast cell line NIH3T3 did not form a sheet-like structure using the centrifugal cell stacking technique, and mMSCs seeded at the same cell number formed a non-uniform sheet-like structure (Fig. 2b and Supplementary Fig. 1c). Therefore, an optimal centrifugal cell stacking technique is required to produce a uniform bioengineered tissue. In addition, Z-stack images of hLEC+mMSC tissues showed that the locations of CellTracker™ Green-labeled mMSCs and CellTracker™ Orange-labeled hLECs were outside (detected at a depth of 0–35 μm from the top) and inside (detected at a depth of 10–30 μm, especially

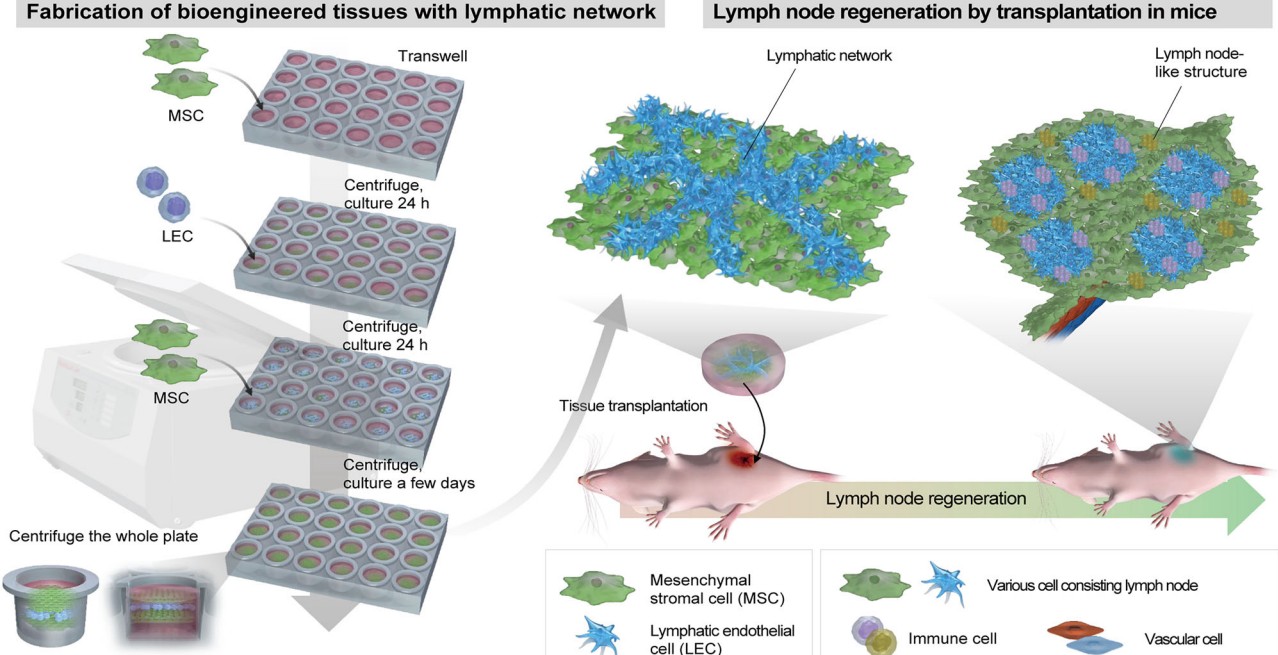

**Fig. 1 | Schematic illustration of the fabrication of centrifugation-based bioengineered lymphatic tissues (CeLyTs) for lymph node reconstruction.** Schematic showing the fabrication of a three-dimensional cellular structure with a lymphatic network composed of LECs and MSCs using a centrifugal cell stacking technique. MSCs, LECs, and MSCs are seeded and layered by centrifugation in that order. The CeLyT is completed by culturing for several days to form a lymphatic network. CeLyTs can survive for a long time after transplantation in LD mice and they form a lymph node-like structure that functions as a lymph node. Figure 1 was created by Science Graphics Inc. (Kyoto, Japan) under a commissioned agreement. All rights have been transferred to the corresponding author.

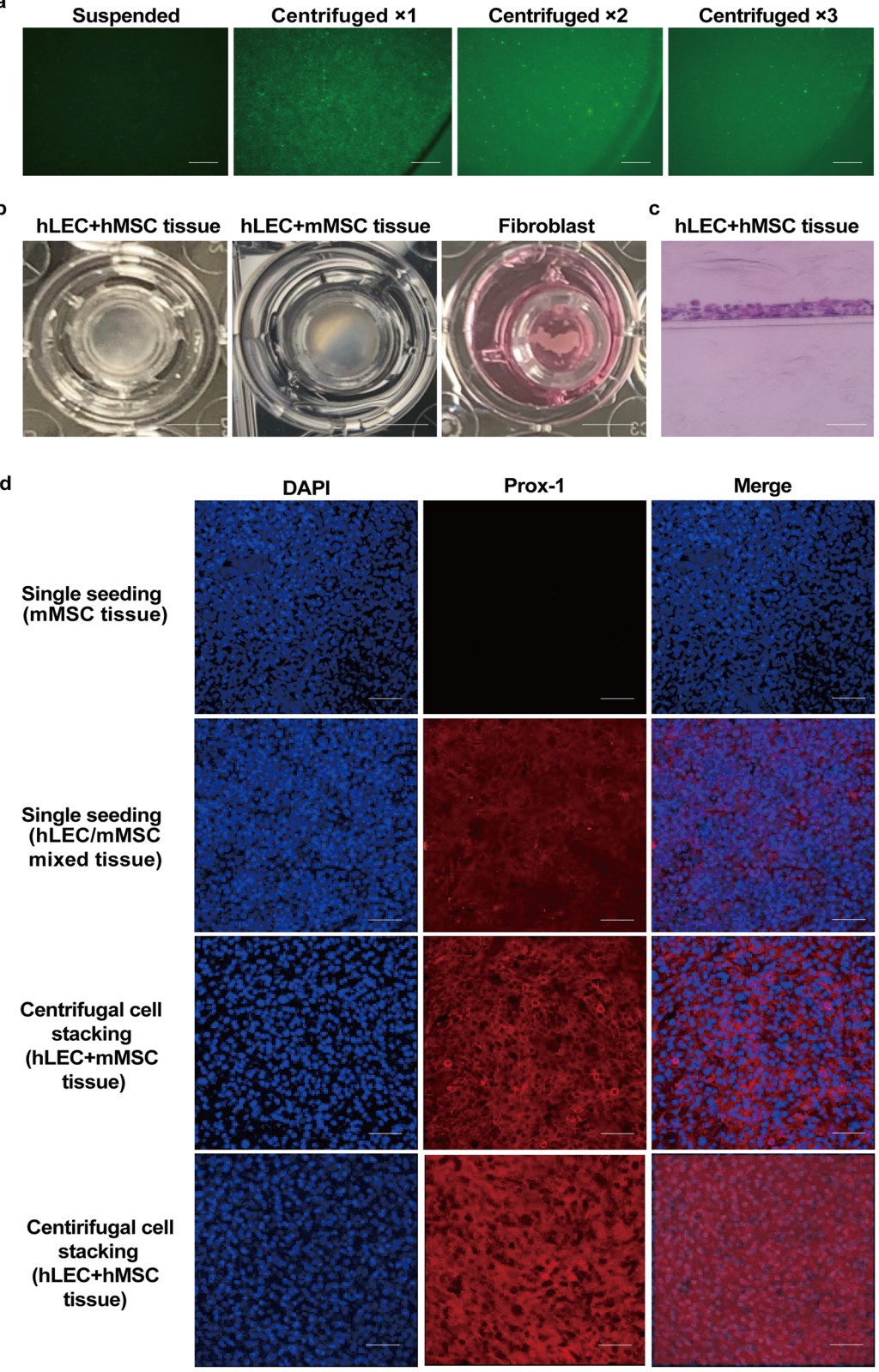

**Fig. 2 | Fabrication of bioengineered tissues. a** Fluorescence microscopic observation of suspended and centrifuged mMSC/GFP at different times after seeding (day 1) of > 3 independent experiments. Scale bars, 500 μm. **b** Typical images of hLEC+hMSC tissue, hLEC+mMSC tissue, and NIH3T3 (fibroblasts) fabricated by a centrifugal cell stacking technique (day 5). Scale bars, 5 mm. **c** Paraffin-section image of an hLEC+hMSC tissue (day 5) of > 3 independent experiments.

Scale bars, 50 μm. **d** Immunofluorescence images of a lymphatic endothelial cell marker Prox-1 in mMSC tissues (by single seeding without centrifugation), hLEC/mMSC mixed tissue (by single seeding without centrifugation), hLEC+mMSC tissue (by centrifugal cell stacking technique), and hLEC+mMSC tissue (by centrifugal cell stacking technique) of > 3 independent experiments. Blue, DAPI; Red, Prox-1. Scale bars, 100 μm.

10–20 μm from the top) the tissue, respectively (Fig. 3a). These data demonstrated that a three-layered bioengineered tissue composed of LECs and MSCs was successfully fabricated by the centrifugal cell stacking technique. To observe the structure of the bioengineered tissues in more detail, hLEC+mMSC tissues fabricated using Cell-Tracker™ Green-labeled mMSCs and CellTracker™ Orange-labeled hLECs were observed in two or three dimensions by fluorescence microscopy. hLECs formed a lymphatic network in the tissue as expected (Fig. 3b and Supplementary Fig. 1d, e). Centrifuge-based bioengineered lymphatic tissues composed of LECs and MSCs will be referred to as CeLyTs and represented as LEC + MSC tissues (the former cells are inside and the latter cells are outside the tissue, respectively) unless otherwise indicated.

To evaluate the survival of the internal and external cells during the culture period to form lymphatic networks, a NanoLuc luciferase (Nluc)-expressing mMSC/Nluc+mMSC tissue was prepared. The relative light units derived from live cell-releasing Nluc in the culture supernatant remained at similar levels throughout the culture period, with no statistically significant differences (Fig. 3c). A similar result was obtained for the luminescence images of a firefly luciferase (fluc)-expressing mMSC/fluc+mMSC tissue, where luminescence signals were detected throughout the tissue (Fig. 3d). In addition, the fluorescence microscopic observation of an hMSC+hMSC/carboxy-fluorescein succinimidyl ester (CFSE; for live cell staining) tissue showed that the external mMSCs were also alive (Fig. 3e). These results demonstrated that the CeLyTs formed a lymphatic network by culturing for several days without the induction of cell death. To evaluate the characteristics of the CeLyTs, the expressions of two lymphatic endothelial cell marker genes, Podoplanin (*PDPN*, also known as *GP38* and *T1α*) and VEGFR-3 (*FLT4*), were compared with those of other cells or tissues. The expressions of *PDPN* and *FLT4* genes, important factors of lymphangiogenesis in hLEC+mMSC tissues, were significantly increased compared with those in an hLEC/mMSC mixed tissue prepared by single seeding without centrifugation, and were also higher than those in monolayered hLEC/mMSC (Supplementary Fig. 2a). In addition, the gene expressions in hLEC+hMSC tissues increased with time during the culture period (Supplementary Fig. 2b). Flow cytometric analysis showed that the geometric mean fluorescence intensity of the hLEC/CFSE+hMSC tissue was significantly higher than that of the hLEC/CFSE/hMSC mixed tissue prepared by single seeding without centrifugation (Supplementary Fig. 2c), indicating the survival of internal LECs was maintained in CeLyTs. This result corresponded with the expression level of Prox-1 protein in the bioengineered tissues (Fig. 2d). Next, we evaluated the contribution of MSCs to the network formation of LECs because MSCs expressed and produced collagen 1A2, which is important for the formation of endothelial cell vascular networks[26]. The expression of collagen 1A2 was observed throughout the mMSC sheet prepared by centrifugation and was higher than that in suspended mMSCs (Supplementary Fig. 2d–f). The shape of seeded hLECs on the mMSCs sheet changed to a thick spindle shape 5 days after seeding (Supplementary Fig. 2g).

## Lymphatic structure formation after CeLyT transplantation

To confirm the usefulness of CeLyTs fabricated with MSCs for survival after transplantation, the survival periods of suspended mMSC/Nluc cells and various bioengineered tissues were evaluated by in vivo imaging. The same number of Nluc-expressing cells were transplanted, and the survival rates after transplantation were evaluated relatively by adjusting the color scale of each group. An mMSC/Nluc+NIH3T3 tissue was prepared as a control as previously reported[21]. The luminescence signals of suspended mMSC/Nluc cells and mMSC/Nluc+NIH3T3 tissues disappeared 3–4 days after transplantation, and those of mMSC/Nluc+hMSC and hLEC+mMSC/Nluc tissues persisted for at least 21 days (Fig. 4a, b), indicating the long-term survival of internal and external cells in CeLyTs. Figure 4c shows the higher and longer survival period

of internal mMSC/Nluc cells in mMSC/Nluc+hMSC tissues compared with that of suspended mMSC/Nluc cells, indicating that the internal cells in CeLyTs survived over a long period when fabricated with MSCs. Next, we evaluated the formation of lymphatic vessels after subcutaneous transplantation of hLEC+mMSC tissues to mice. The expressions of Prox-1 and an endothelial marker CD31 were detected in skin tissues transplanted with hLEC+mMSC tissues 14 days after transplantation, although the expression of Prox-1 in normal skin was beyond the limit of detection (Fig. 4d). Furthermore, the transplanted hLECs survived and were located between the dermis and peritoneum (Fig. 4e). In addition, the shape of LECs expressing Prox-1 and CD31 in the hLEC+mMSC tissue-transplantation group showed the formation of luminal structures by lymphatic cells, as previously reported[22].

## Restoration of lymphatic flow in LD mice after CeLyT transplantation

We evaluated the therapeutic effect of CeLyTs. First, we used LD mice with impaired lymphatic flow by removing the popliteal and inguinal lymph nodes, which caused lymphedema. Then, we evaluated the lymphatic flow in these mice using a lymphatic flow testing reagent indocyanine green (ICG) by in vivo imaging. ICG signals were detected in the popliteal lymph nodes of normal mice (no treatment) 1 h after ICG injection, whereas no signals were detected in the popliteal lymph nodes of ICG-injected LD mice 1, 7, or 14 days after popliteal and inguinal lymph node removal (Supplementary Fig. 3a). ICG fluorescence signals in the popliteal lymph nodes (the transplantation site) in LD mice transplanted with suspended mMSC/Nluc cells or various bioengineered tissues were analyzed 1 h after ICG administration by in vivo imaging. hLEC+mMSC/Nluc tissue- and hLEC+hMSC tissue-transplanted LD mice showed fluorescence signals at the transplantation site on day 21, whereas other groups showed no signal, indicating CeLyTs restored the lymphatic flow in LD mice (Fig. 5a). To confirm the persistence of CeLyTs after transplantation into LD mice, the luciferase-derived luminescence of hLEC+mMSC/Nluc tissues was detected by in vivo imaging. Luminescence signals were detected at the transplantation site in LD mice transplanted with hLEC+mMSC/Nluc tissue at 1 and 14 days after transplantation, whereas signals in LD mice transplanted with suspended mMSC/Nluc cells were barely detectable 14 days after transplantation (Fig. 5b). In addition, hLEC+mMSC tissue formed a lymph node-like structure at the transplantation site in LD mice 21 days after transplantation (Fig. 5c and Supplementary Fig. 3b). To confirm that the lymph node-like structure functioned as a lymph node, ICG was administered to hLEC+mMSC tissue-transplanted LD mice (day 21). ICG fluorescence signals were detected in the lymph node-like structure in hLEC+mMSC tissue-transplanted LD mice (Fig. 5d), indicating that ICG was delivered to the lymph node-like structure formed by the transplanted hLEC+mMSC tissue. Consistent with these results, lymph nodes or lymph node-like structures harvested from normal mice and hLEC+mMSC tissue- or hLEC+hMSC tissue-transplanted LD mice after ICG administration showed similar fluorescence intensities (Fig. 5e–g).

## Therapeutic effect of CeLyT transplantation on lymphedema in LD mice

The CeLyTs restored lymphatic flow in LD mice, and therefore we treated lymphedema in LD mice by transplantation with various tissues. LD mice were used and various suspended cells and bioengineered tissues were used for transplantation (Supplementary Fig. 4a [top]). The percentage change in the thickness of paws and legs of LD mice to that of normal mice (no treatment) increased until days 5–7, reaching 20 and 6%, respectively, and then remained almost constant until day 21 (Fig. 6a–c). The suspended hLECs transplantation group and suspended mMSCs transplantation group had a low therapeutic effect, which was equivalent to that of the LD mouse group. However, the hLEC+mMSC and hLEC+hMSC tissue groups had a suppressive

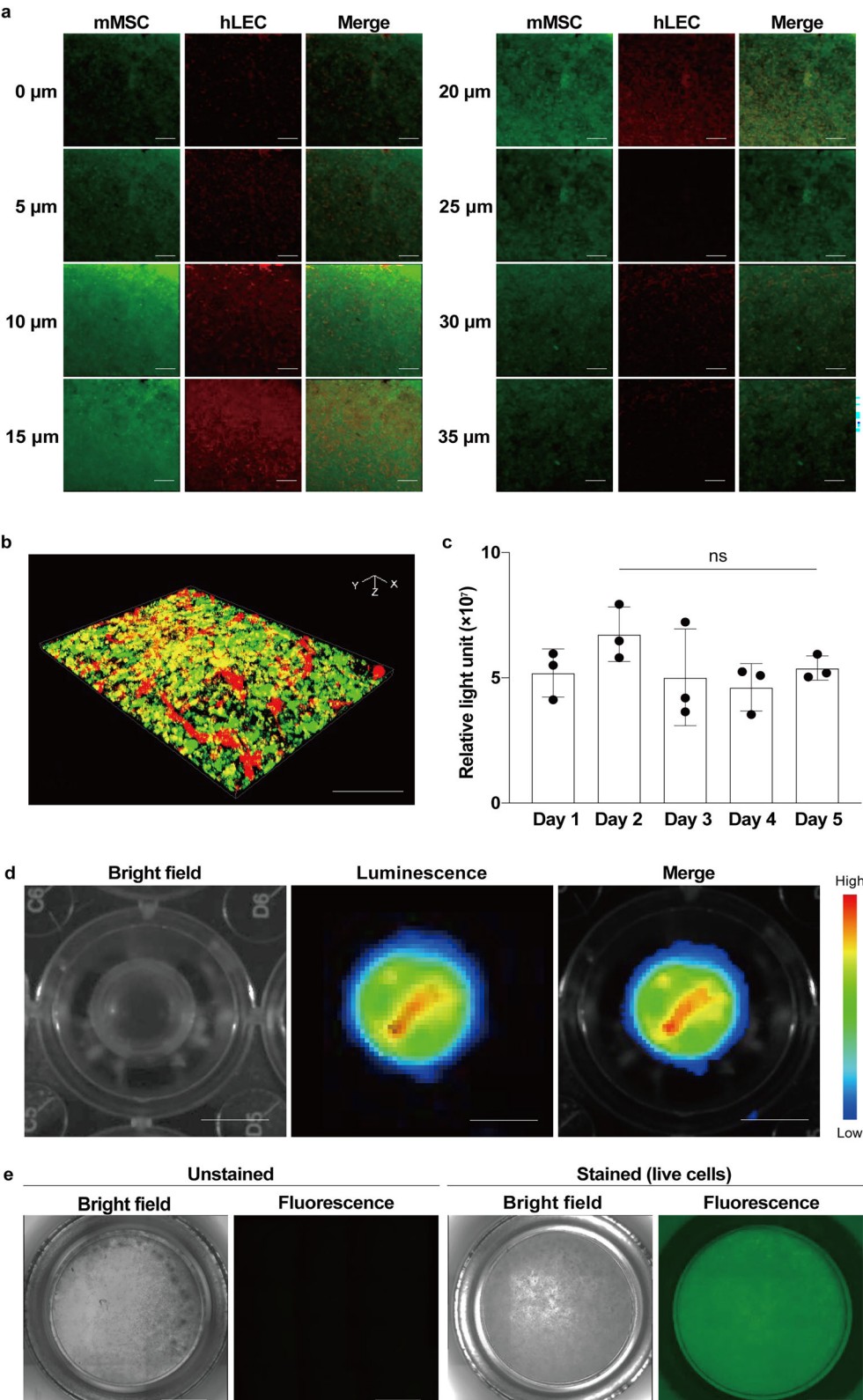

**Fig. 3 | Characteristics of CeLyTs. a** Z-stack images taken every 5 µm in an hLEC +mMSC tissue of > 3 independent experiments. Green, CellTracker™ Green-labeled mMSC; Red, CellTracker™ Orange-labeled hLEC. Scale bars, 100 µm. **b** Typical 3D image of an mMSC+hLEC tissue. Green, CellTracker™ Green-labeled mMSC; Red, CellTracker™ Orange-labeled hLEC. Scale bar, 300 µm. **c** Relative light units in the culture supernatant of an mMSC/Nluc+mMSC tissue as an indicator of the viability of mMSC/Nluc inside the tissue. Data represent the mean ± standard deviation from 3 independent experiments, and P-values were determined by two-sided Dunnett's test. *P < 0.05 was considered statistically significant for Day 1. ns, not significant. **d** Luminescence images of an mMSC/fluc+mMSC tissue to evaluate the viability of mMSC/Nluc inside the tissue (day 5). Scale bar, 5 mm. **e** Fluorescence microscopic images of an mMSC+mMSC/CFSE tissue to evaluate the viability of mMSC/CFSE outside the tissue (day 5). Green, CFSE. Scale bar, 5 mm.

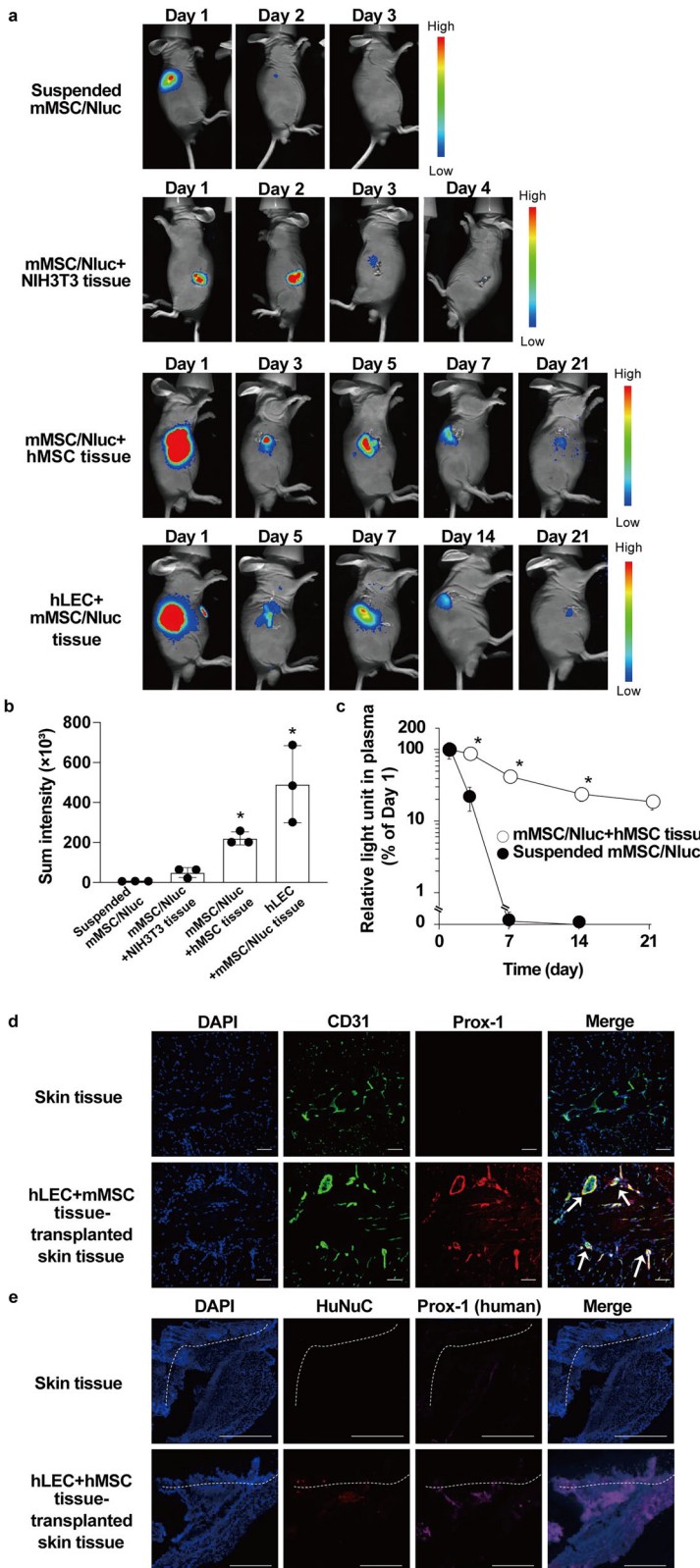

effect on the change ratio after a transient increase, which finally was almost equivalent to that of normal mice. The results demonstrated that CeLyTs exhibited a high therapeutic effect on the treatment of LD mice compared with MSC transplantation, which is anticipated to be a potential therapeutic strategy for lymphedema. Consistent with these results, sectioned images of paws showed that only hLEC+mMSC tissue and hLEC+hMSC tissue groups suppressed the accumulation of

interstitial fluid in the paws of the LD mice (Fig. 6d). We then treated lymphedema after its formation in LD mice transplanted with hLEC +hMSC tissues on day 5 after LD (Supplementary Fig. 4a, bottom). Consistent with the above results, hLEC+hMSC tissues suppressed the increased percentage change in the thickness of paws and legs in LD mice, inhibited the accumulation of interstitial fluid, and promoted ICG influx into the lymph node-like structure formed at the

**Fig. 4 | Survival of CeLyTs and formation of lymphatic structures after transplantation. a** In vivo bioluminescence imaging of suspended mMSC/Nluc, mMSC/Nluc+NIH3T3, mMSC/Nluc+hMSC, and hLEC+mMSC/Nluc tissues. Luciferase-derived signals were observed in live cells. **b** Sum intensity of region of interest signals quantified from in vivo bioluminescence imaging (Fig. 4a) at the transplanted area. Suspended mMSC/Nluc, day 3; mMSC/Nluc+NIH3T3 tissue: day 4; mMSC/Nluc+hMSC tissue, day 7; hLEC+mMSC/Nluc tissue: day 7. Data represent the mean ± standard deviation from 3 independent experiments, and P-values were determined by the two-sided Student's *t*-test. *P < 0.05 was considered statistically significant for the suspended mMSC/Nluc group. **c** Relative light units in plasma

after the transplantation of suspended mMSC/Nluc and mMSC/Nluc+hMSC tissues. Data represent the mean ± standard deviation from 3 independent experiments, and P-values were determined by two-tailed Student's *t*-test. *P < 0.05 was considered statistically significant for the suspended mMSC/Nluc group. **d** Typical immunofluorescence images of hLEC+mMSC tissues 14 days after subcutaneous transplantation into mice. Blue, DAPI; Green, CD31; Red, Prox-1. Arrows, luminal structures formed with lymphatic cells. Scale bars, 100 μm. **e** Representative immunofluorescence images of hLEC+mMSC tissues 14 days after subcutaneous transplantation into mice. Blue, DAPI; Red, HuNuC; Magenta, human-reactive Prox-1. The dermis is indicated above the dotted lines. Scale bars, 1 mm.

transplantation site (Supplementary Fig. 4b–e). To confirm the therapeutic effect of CeLyTs on lymphedema, mMSC tissues and hLEC/mMSC mixed tissues prepared by single seeding without centrifugation were transplanted into LD mice. Although these tissues formed lymph node-like structures, the ICG influx into the lymph node-like structure, the therapeutic effect on lymphedema, and the suppressive effect on the accumulation of interstitial fluid were poor (Supplementary Fig. 5a–e). In addition, although compression therapy represents the gold standard for the treatment of lymphedema in clinical practice, it was observed that this approach transiently delayed the swelling of edema in the paws of LD mice. By contrast, CeLyTs were more effective at suppressing lymphedema than treatment with compression therapy (Supplementary Fig. 6a, b). CeLyTs also exhibited therapeutic effects in a more severe chronic lymphedema model, which was prepared by removing the iliac, inguinal, and popliteal lymph nodes (Supplementary Fig. 7a). Moreover, they demonstrated a greater lymphedema-suppressive effect compared with bioengineered tissues fabricated by other tissue engineering methods, such as hLEC/hMSCs spheroids and hMSC+NIH3T3 tissues generated by the cell coating method (Supplementary Fig. 7b, c). Furthermore, we confirmed the effect of the CeLyTs on the hallmarks of lymphedema, including skin thickening and adipose tissue accumulation. The hLEC+mMSC tissue groups had a suppressive effect on the thickness of skin compared with the suspended mMSC and hLEC groups (Supplementary Fig. 8a, b). In addition, immune cell populations were present in the dermis, which showed signs of inflammation[27], including decreased T cell numbers and increased macrophage numbers in the LD group. However, the hLEC+hMSC tissue-transplanted groups showed a recovery of immune cell populations, with T cell and macrophage numbers reaching levels nearly equivalent to those of normal mice (Supplementary Fig. 8c). The adipose tissue accumulation in the skin was also reduced by transplantation of the CeLyTs (Supplementary Fig. 8d). These findings suggested CeLyTs containing LECs formed functional lymph node-like structures and had a marked therapeutic effect on lymphedema.

## Analyses of lymph node-like structures after CeLyT transplantation

Finally, we analyzed the characteristics of lymph node-like structures formed after the transplantation of CeLyTs. We evaluated whether cells forming lymph node-like structures were derived from the transplanted CeLyTs, the host animals, or both. Immunostaining of CD31 and Prox-1 using different cross-species antibodies (human and mouse) showed that the lymph node-like structures formed by hLEC+hMSC tissue transplantation were composed of cells derived from the hLEC+hMSC tissues and host animal, indicating CeLyTs induced the formation of lymph node-like structures in mice after transplantation (Fig. 7a). In addition, lymphatic vessels derived from the host animal and endogenous LECs from the donor co-existed in a mixed state within the tissue (Fig. 7b). To clarify the contribution of transplanted LECs in the formation of lymphatic vessels and lymph node-like structures, CeLyTs prepared using hLECs expressing green fluorescent protein (hLEC/GFP) and GFP fluorescence was tracked (Supplementary Fig. 9a). A few host-derived LECs were observed at day 7. By day 14,

transplanted and host-derived LECs co-localized, and by day 21, they were connected through the extracellular matrix, with some evidence of cell fusion (Supplementary Fig. 9b, c). In contrast, host-derived LECs were rarely observed in hMSC tissue transplants at day 21, highlighting the importance of LECs in CeLyTs for the formation of lymphatic vessels and lymph node-like structures (Supplementary Fig. 9d). If these lymph node-like structures function as lymph nodes, immune cells must be present. Therefore, we immunostained lymph node-like structures formed after hLEC+hMSC tissue transplantation, with CD3 (T cells), B220 (B cells), CD11b (macrophages), and CD11c (dendritic cells) antibodies, and confirmed their presence (Fig. 7c and Supplementary Fig. 10a–c). Interestingly, B cells within the lymph node-like structures formed a follicle-like structure, similar to that in the lymph nodes of normal mice (Supplementary Fig. 10a). Lymph node-like structures formed after hLEC+hMSC tissue transplantation were also immunostained with HuNuC antibody, and this was supported by part of the lymphatic vessels in the lymph node-like structures formed by transplanted cells (Supplementary Fig. 10d). In addition, we evaluated the immune responses of the lymph node-like structures using immunostimulatory CpG1018. The lymph node-like structures produced various proinflammatory cytokines including interferon (IFN)-γ, interleukin (IL)-6, IL-12, and tumor necrosis factor (TNF)-α, the levels of which were equivalent to those in the lymph nodes of normal mice treated with CpG1018 (Fig. 7d). Furthermore, the distribution of cells within the lymph node-like structures formed after hLEC+hMSC tissue transplantation was evaluated. Various cells, including LECs and immune cells, were distributed throughout the overall lymph node-like structures, and follicle-like structures were beginning to form (Fig. 8a–c, Supplementary Fig. 10e). Therefore, high-magnification hematoxylin and eosin staining of the reconstructed lymph node-like structures revealed distinct lymphoid structures, including lymphoid follicles and sinus-like spaces. Immunofluorescence staining for CD3, B220, CD11b, and LYVE-1 (LECs) demonstrated partial segregation of T cell and B cell zones; however, both cell types were intermingled, and the boundary between the cortex and the medulla was not clearly defined. Within a region consistent with the medulla, macrophages were abundantly distributed, supporting the presence of a medulla-like structure. Lymphatic structures were broadly distributed throughout the tissue. These findings indicate that the reconstructed tissues possess key features of a lymph node, such as follicles, sinuses, and a medulla region, but remain structurally immature compared with mature lymph nodes (Supplementary Fig. 11a–c). Wide-field views confirmed the location of the lymph node-like structure within the host hindlimb, adjacent to skeletal muscle. LECs were distributed throughout various tissue compartments but were most densely localized in the central region of the lymph node-like structure, consistent with lymphatic lineage enrichment in the core of the graft (Supplementary Fig. 11d). In addition, the immune cell populations of the lymph node-like tissues were analyzed 7, 14, and 21 days after the transplantation of hLEC+hMSC tissues in LD mice. CD3 positive T cell numbers were increased at 7 days, then decreased at 14 days, but increased to the same level as in the lymph nodes 21 days after transplantation. Numbers of CD19 positive B cells and CD11c positive dendritic cells also rapidly increased to the same levels as in lymph nodes

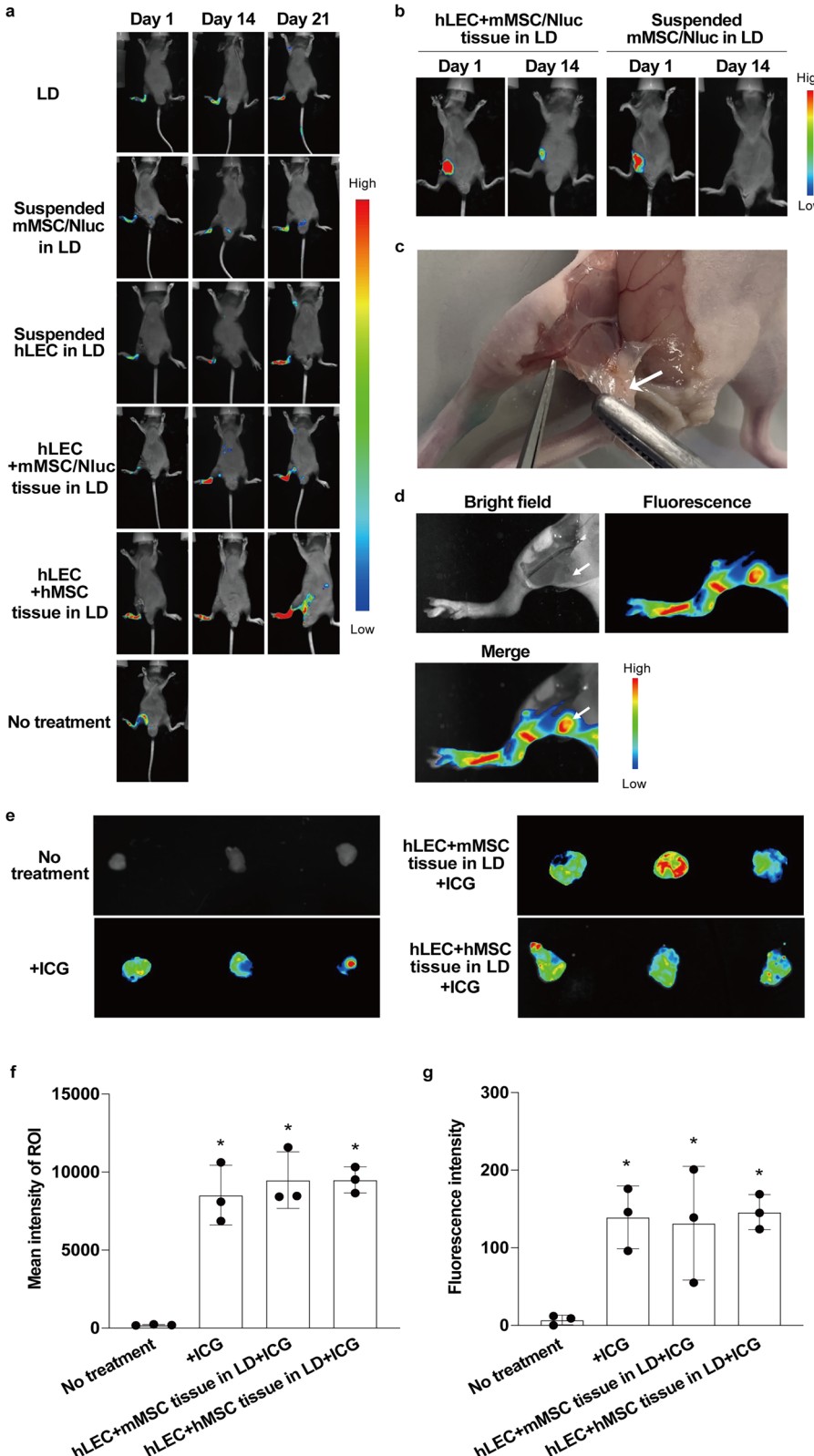

21 days after transplantation. However, only the numbers of F4/80 positive macrophages had increased 7 days after transplantation, which might be an early immune response. These then increased further to become more abundant than in the lymph nodes 14 and 21 days after transplantation. Images showing abundant immune cells in the lymph node-like structures are shown in Fig. 8d. To confirm the lymph node-like structures functioned as lymph nodes, ICG influx was also analyzed after ICG administration to hLEC+hMSC tissue-transplanted mice (day 21) by in vivo imaging. ICG fluorescence signals at the transplantation site of hLEC+hMSC tissues increased with time starting 10 min after ICG administration, and then the signals migrated to the adjacent lymph nodes, indicating that ICG that flowed into the lymph node-like structure was subsequently drained (Supplementary Fig. 12a). Furthermore, after the removal of distant lymph nodes and

**Fig. 5 | Restoration of lymphatic flow in LD mice after the transplantation of CeLyTs. a** In vivo fluorescence imaging after the injection of ICG to footpads of normal (no treatment) or LD mice after cell transplantation. LD mice had their inguinal and popliteal lymph nodes removed. Bioengineered tissues were transplanted to the site of the removed popliteal lymph nodes. **b** In vivo bioluminescence imaging of hLEC+mMSC/Nluc tissue and suspended mMSC/Nluc in LD mice after transplantation. **c** Formation of a lymph node-like structure 21 days after the transplantation of hLEC+mMSC tissues in LD mice. Arrow, a formed lymph node-like structure. **d** In vivo fluorescence imaging after the injection of ICG to footpads of hLEC+mMSC tissue-transplanted LD mice. Arrow, site of ICG influx into the formed lymph node-like structure. **e** Ex vivo imaging of a lymph node or lymph node-like structure after ICG administration to normal mice (+ICG), hLEC+mMSC tissue-transplanted mice, and hLEC+hMSC tissue-transplanted LD mice (day 21). **f** Sum intensity of region of interest (ROI) signals quantified from in vivo fluorescence imaging (Fig. 5e). Data represent the mean ± standard deviation from 3 independent experiments, and P-values were determined by two-sided Dunnett's test. *P < 0.05 was considered statistically significant for the No treatment group. **g** Fluorescent intensity of excised and homogenized lymph nodes or lymph node-like structures after ICG administration. Data represent the mean ± standard deviation from 3 independent experiments, and two-sided Dunnett's test determined P-values. *P < 0.05 was considered statistically significant for the No treatment group.

lymph node-like structures, ICG fluorescence signals in the lymph node-like structures increased at 1 h, while at 3 h after ICG administration, the signals were decreased in the lymph node-like structures and increased in the iliac lymph nodes (Supplementary Fig. 12b, c). The use of fluorescence-labeled beads to assess the filtering ability of the lymph node-like structures demonstrated that fluorescence signals in the lymph node-like structures were diminished, whereas a slight increase in signals was observed in the iliac lymph nodes 3 h after fluorescence-labeled bead administration (Supplementary Fig. 12d). These findings confirm that the lymphatic flow behavior and filtration capacity of the lymph node-like structures were comparable to those of popliteal lymph nodes in mice. In addition, the lymph node-like structure remained 100 days after the transplantation of CeLyTs (Supplementary Fig. 13a), and the improvement of lymphatic flow and suppression of lymphedema by CeLyTs transplantation were also sustained for 100 days (Supplementary Fig. 13b, c), indicating that the therapeutic effect of CeLyTs was significantly higher than that of MSC transplantation over a long period. Furthermore, the transplanted cells and various immune cells remained in the formed lymph node-like structure after 100 days (Supplementary Fig. 13d, e).

## Discussion

In this study, we successfully fabricated CeLyTs composed of LECs and MSCs and demonstrated their usefulness for the treatment of secondary lymphedema, for which no effective radical treatment is available. Many previous reports have attempted to develop cell-based therapy for lymphedema and have reported excellent results, such as the formation of lymphatic vessels in vivo[19,20,28]. However, the reconstruction and regeneration of lymph nodes or functional lymph node-like structures and their therapeutic effects on lymphedema have never been achieved. To achieve this, we focused on the survival and presence of functional LECs in a three-dimensional cellular structure after transplantation. The centrifugal cell stacking technique can fabricate a three-layered cellular structure composed of an internal layer of LECs and two external layers of MSCs, which can prolong the survival of LECs after transplantation without unnecessary additions and severe cell damage. In addition, this structure can induce the formation of a lymphatic network of LECs within the structure. Then, CeLyTs form a lymph node-like structure after transplantation into mice, resulting in the restoration of lymph node functions in LD mice.

For cell-based therapy, cell survival is an important factor related to the therapeutic effect, as we previously reported[29–31]. This is because the transplanted cells release therapeutic cytokines and contribute to the reconstruction of the tissues themselves. In most cases, immunosuppressive drugs such as tacrolimus are used to prolong the survival of transplanted cells in cell-based therapy[32–34]. However, the long-term use of immunosuppressive drugs causes systemic adverse effects such as renal disorder, myelosuppression, and increased risk of infectious diseases, leading to a reduced quality of life for patients[35–37]. MSCs are multipotent stromal cells that can differentiate into multiple lineages and support cell survival by releasing immunosuppressive cytokines without systemic side effects in the case of local transplantation. We hypothesized that a three-dimensional cellular structure composed of

MSCs on the outside would have long-term survival. Indeed, previous reports showed the long-term survival of a sheet-like cellular structure composed of MSCs[38–40]. Here, mMSC/Nluc+hMSC, hLEC+mMSC/Nluc, and mMSC tissues had long-term survival and formed lymph node-like structures after transplantation, whereas suspended mMSCs or mMSC/Nluc+NIH3T3 tissues disappeared within a few days (Fig. 4a–c, Fig. 5b, c). Furthermore, we speculated that the survival of the internal cells of CeLyTs was more important for the treatment of lymphedema, because the mMSC tissues or hLEC/mMSC mixed tissues prepared by single seeding without centrifugation had a low therapeutic effect on lymphedema (Supplementary Fig. 5c–e). This result indicated that LECs were necessary to reconstruct the lymph node-like structures that functioned as lymph nodes. We evaluated the survival of internal cells in the respective bioengineered tissues and demonstrated that the survival of the internal cells in CeLyTs (mMSC/Nluc+mMSC tissues or hLEC/CFSE+hMSC tissues) was maintained longer compared with that in the hLEC/CFSE/hMSC mixed tissues (Fig. 3c and Supplementary Fig. 2c). CeLyTs had a suppressive effect on the hallmarks of lymphedema induced by chronic inflammation (Supplementary Fig. 8). Suspended MSC transplantation also showed a tendency to improve skin thickening and adipose tissue accumulation (Supplementary Fig. 8a, b, d), indicating the immunoregulatory ability of MSCs may also contribute to improving inflammation associated with lymphedema by the transplantation of CeLyTs. In addition, CeLyTs were shown to exert a sustained therapeutic effect for 100 days, which can be attributed to the long-term survival of the tissue associated with the presence of MSCs (Supplementary Fig. 13). Therefore, the presence of MSCs in the bioengineered lymphatic tissues and the centrifugal cell stacking technique contributed to the long-term survival of LECs after the transplantation of CeLyTs.

Previously reported three-dimensional cellular structures containing LECs promoted an internal lymphatic network in the structure[22,41,42]. In general, endothelial cells, including lymphatic endothelial cells, can form a vascular network related to external topological effects and the presence of surrounding collagen, which exerts the original functions[26,43,44]. Therefore, we cultured the fabricated bioengineered tissues for 5 days to form a lymphatic network in the tissue (Fig. 3b and Supplementary Fig. 1d, e). The immunostaining of collagen 1A2 in the bioengineered tissue showed that mMSCs played the role of a collagen 1A2 donor and mMSC sheets with a high expression of collagen 1A2 provided a collagen 1A2-rich environment (Supplementary Fig. 1d, e). In addition, the expressions of lymphatic endothelial cell marker genes, *PDPN* and *FLT4*, in the hLEC+mMSC tissues increased with time during culture (Supplementary Fig. 2b), and were higher than those in the hLEC/mMSC mixed tissue (Supplementary Fig. 2a). These results indicated that the centrifugal cell stacking technique induced the formation of a lymphatic network in the structure, and the appropriate culture time led to the functionalization of LECs, resulting in the formation of functional lymph node-like structures after the transplantation of CeLyTs.

Lymph nodes can be characterized by their active immune response mediated by abundant immune cells and lymphatic inflow and outflow[45–48]. The transplantation of CeLyTs into LD mice formed a

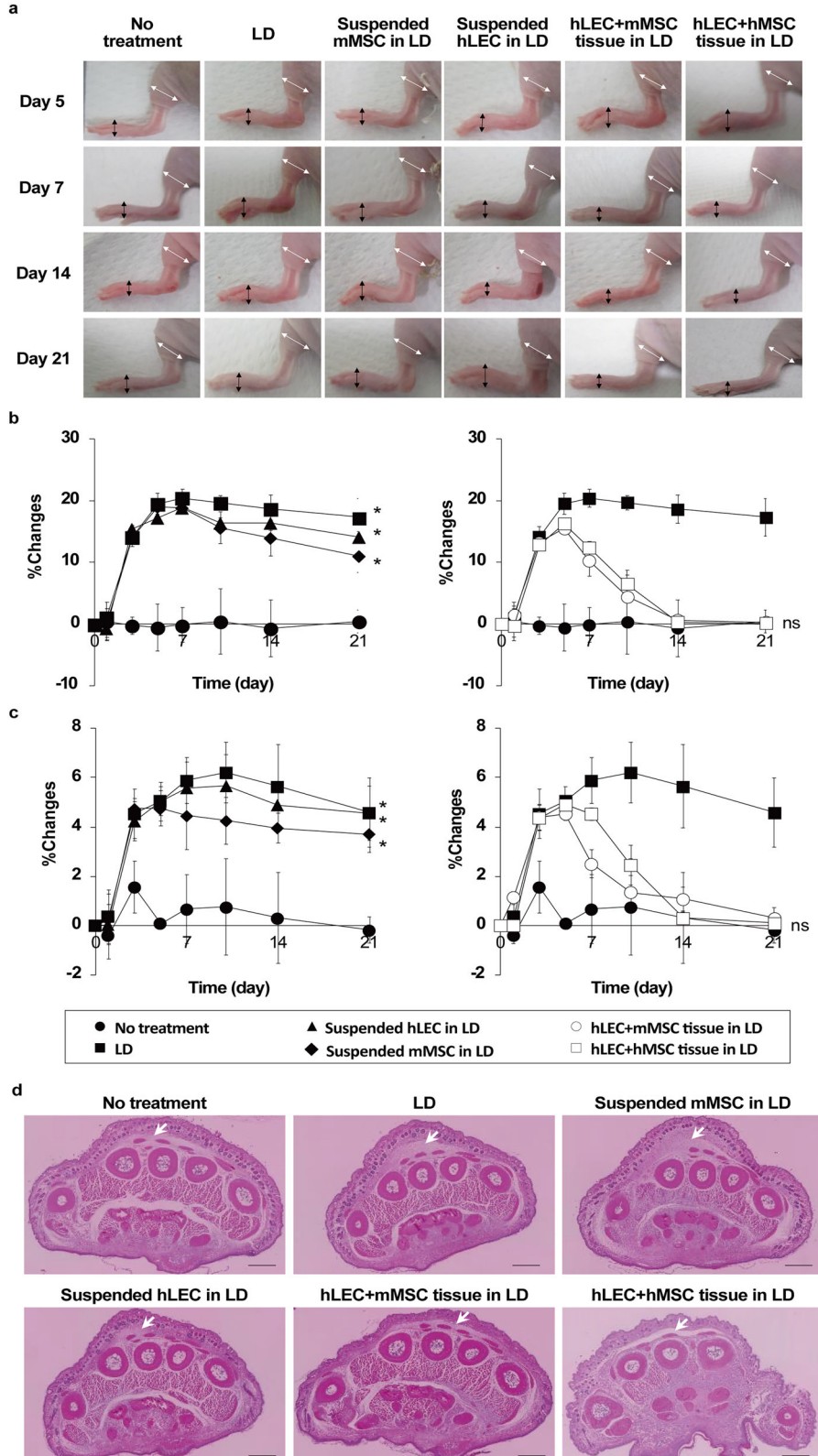

**Fig. 6 | Therapeutic effect of the transplantation of CeLyTs on lymphedema in LD mice. a** Typical images of the lower limbs of bioengineered tissue-transplanted LD mice. Arrows indicate the measurement site (white, legs; black, paws) in the lower limbs for the evaluation of edema. **b,c** Change in the size of the paws (**b**) and legs (**c**) of bioengineered tissue-transplanted LD mice. The thicknesses of the paws and legs were measured and the change rates from day 0 were calculated. Data represent the mean ± standard deviation from 3 independent experiments, and P-values were determined by two-sided Dunnett's test. *P < 0.05 was considered statistically significant for the No treatment group. ns, not significant. **d** Paraffin-section images of the paws of bioengineered tissue-transplanted LD mice (day 21). Arrows indicate the interstitial fluid. Scale bars, 500μm.

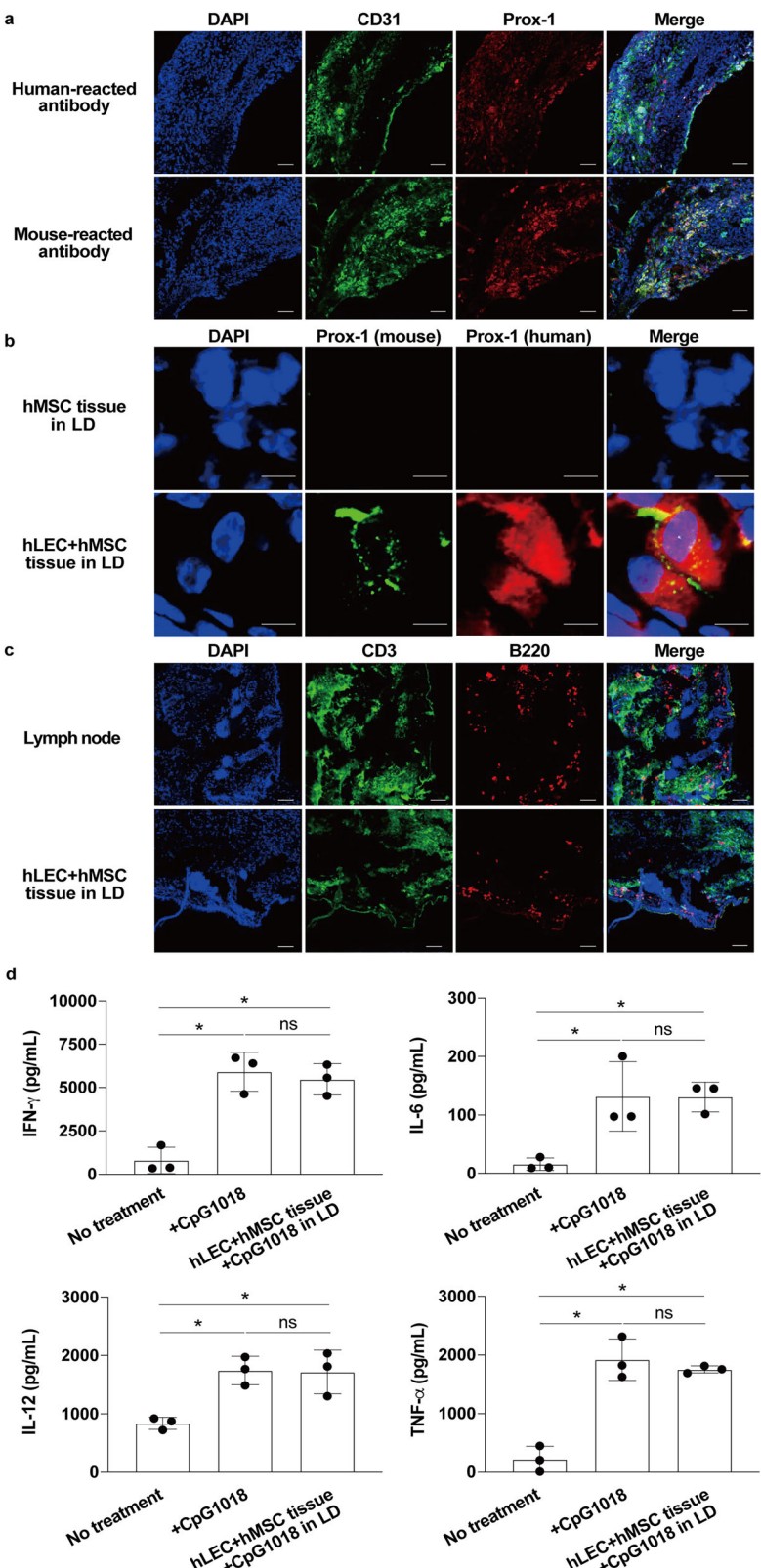

lymph node-like structure containing a wide range of immune cells, which promoted an immune response to CpG1018, and the inflow and outflow of ICG (Fig. 7c, d, Supplementary Fig. 10 and Supplementary Fig. 12a). Furthermore, the lymph node-like structures were connected to distant lymph nodes and integrated into a lymphatic network with filtration capacity (Supplementary Fig. 12b–d). These results indicate that the lymph node-like structure functions as a lymph node.

Although the lymph node-like structures were immature compared with the lymph nodes, they contained lymphatic vessels, and lymph node-specific structures such as follicular structures, cortex, and medulla were beginning to form (Fig. 8a–c, Supplementary Fig. 10e and Supplementary Fig. 11). The lymph node-like structures were missing structures that exist in mature lymph nodes, such as a T cell zone, B cell zone, and follicular structures. However, the function of

**Fig. 7 | Formation of a lymphatic node-like structure after the transplantation of CeLyTs. a** Representative immunofluorescence staining for an endothelial marker CD31 and lymphatic endothelial cell marker Prox-1 in a lymph node-like structure formed in hLEC+hMSC tissue-transplanted LD mice (day 21) of > 3 independent experiments. Blue, DAPI; Green, CD31: Red, Prox-1. Scale bars, 100 μm. Human- or mouse-reactive antibodies were used in the upper panel and lower panel, respectively. **b** Representative immunofluorescence staining for a lymphatic endothelial cell marker Prox-1 in a lymph node-like structure formed in hLEC+hMSC tissue-transplanted LD mice (day 21) of > 3 independent experiments. Blue, DAPI; Green, mouse-reactive Prox-1: Red, human-reactive Prox-1. Scale bars, 50 μm.

**c** Representative immunofluorescence staining for a T cell marker CD3 and B cell marker B220 in a lymph node or lymph node-like structure formed in hLEC+hMSC tissue-transplanted LD mice (day 21) of > 3 independent experiments. Blue, DAPI; Green, CD3: Red, B220. Scale bars, 100 μm. **d** Cytokine production in lymph nodes or lymph node-like structures of hLEC+hMSC tissue-transplanted LD mice (day 21) 3 h after CpG1018 administration (footpad). Data represent the mean ± standard deviation from 3 independent experiments, and P-values were determined by two-sided Dunnett's test. *P < 0.05 was considered statistically significant. ns, not significant.

the lymph node-like structures was almost equivalent to those of lymph nodes, and LECs and various immune cells were present, indicating that the lymph node-like structures were at a stage prior to maturation. Future studies are needed to elucidate the mechanisms underlying the maturation of lymph node-like structures. The changes in immune cell populations in the lymph node-like structures included a temporary increase in T cell and macrophage numbers that might be caused by immune rejection 7 days after the transplantation of CeLyTs, followed by a re-increase related to lymph node-like structure formation 21 days after transplantation. However, the numbers of B cells and dendritic cells increased rapidly 21 days after transplantation (Fig. 8d). Lymph node-like tissues contained abundant immune cells 21 days after transplantation, but it will be necessary to study changes in immune cell populations after 21 days. Further investigation is required to clarify the mechanism of lymph node-like tissue formation and changes in immune cell populations. Because the structure was composed of transplant- and host animal-derived cells (Fig. 7a), the bioengineered tissues can induce the formation of functional lymph node-like structures, and LECs in the tissues are the key to lymph node regeneration. In addition, lymph nodes do not regenerate in LD mice and suspended LECs barely formed lymph node-like structures (Supplementary Fig. 3b), indicating that the long-term survival of LECs or presence of MSCs with LECs may be essential for the formation of functional lymph node-like structures. Of note, CeLyTs did not show an immediate therapeutic effect in LD mice, which occurred 7–10 days after transplantation. Thus, CeLyTs might initially induce lymphangiogenesis with angiogenesis around the transplantation site, leading to the establishment of an immature lymph node-like structure formed by incorporating host animal-derived cells into the tissue within several days, followed by its maturation and ability to function as a lymph node 7–10 days after transplantation. During this process, various immune cells migrate to the lymph node-like structure through the newly formed lymphatic vessels that form a vascular network in the structure. In addition, CeLyTs exhibited a therapeutic effect that emerged 10–14 days after transplantation and completely suppressed lymphedema for 28 days in chronic lymphedema model mice, in which the iliac lymph nodes were removed in addition to the inguinal and popliteal lymph nodes (Supplementary Fig. 7b, c). Additional removal of the iliac lymph nodes prevents lymphatic bypass through the deep medical system and connections between the deep and superficial lymphatic systems into the inguinal region. The therapeutic effect of CeLyTs in chronic lymphedema model mice took longer to manifest compared with that in LD mice. This suggests that CeLyTs transplantation in LD mice led to an earlier therapeutic effect because of its connection with the iliac lymph nodes. This study has revealed connections between the iliac lymph nodes and lymph node-like structures (Supplementary Fig. 11b, c). However, bypass formation may be more challenging in chronic lymphedema model mice. A more detailed study of this hypothesis is required in the future.

To evaluate the contribution of the transplanted LECs to the formation of the lymphatic vessels and lymph node-like structures, we established hLEC/GFP and directly tracked GFP fluorescence in the CeLyTs prepared using hMSCs and hLEC/GFP (Supplementary Fig. 9a). The transplanted LECs fused with the host-derived LECs to reconstruct

lymphatic vessels (Supplementary Fig. 9b, c). Additionally, some lymphatic luminal structures were formed independently by the transplanted LECs, while others were formed collaboratively with host-derived LECs. These findings are consistent with previous reports and suggest that transplanted LECs play a critical role in the formation of the lymphatic vessels[49]. The host-derived LECs may be induced by cytokines such as VEGF-A, VEGF-D, and IL-8, which were reported to be produced by MSCs in co-culture with LECs[50]. On day 7 after transplantation, only a few host-derived LECs were observed, whereas by day 14, a larger number had appeared and were fused or connected with the transplanted LECs. Furthermore, VEGFR-3 was upregulated in the LECs within CeLyTs, suggesting that the transplanted LECs may contribute to lymphatic vessel reconstruction in vitro and in vivo by acting as guide cells for the host-derived LECs under the influence of VEGF-C/D derived from MSCs (Supplementary Fig. 2a, b). Regarding the formation of lymph node-like structures, transplanted LECs formed immature lymph node-like structures by forming lymphatic luminal structures together with host-derived LECs on day 14 after transplantation. This process likely facilitates immune cell infiltration, supporting the development of lymph node-like structures. In contrast, transplantation of MSC-only tissue did not induce sufficient LECs to form lymphatic vessels (Supplementary Fig. 9d), suggesting that lymph node-like structures are unlikely to form under such conditions. Recently, it has been reported that LECs induce immune cells by producing chemokines such as CCL21. The formation of lymphatic luminal structures composed of LECs may be important in the formation of lymph node-like structures[51,52]. Therefore, chemokines such as CCL21 likely play a crucial role in lymph node development, influencing the formation of lymph node-like structures by both transplanted and host LECs. Based on this evidence, the long-term survival of LECs may be critical for successful lymph node regeneration. Further studies are needed to elucidate the functional roles of transplanted LECs in the formation of lymph node-like structures.

This study had some limitations. First, obtaining primary LECs from mice tissues is challenging. Because primary LECs can only be isolated and harvested in limited quantities from mouse tissues, human-derived primary LECs were purchased and used instead. Second, immunodeficient mice were used in animal experiments, allowing CeLyTs composed of hMSCs and hLECs to survive and form lymph node-like structures in the mice. Finally, our chronic lymphedema model involves resection of three key lymph nodes (iliac, inguinal, and popliteal) to suppress both superficial and deep lymphatic drainage, aiming to induce a more persistent and clinically relevant form of lymphedema. While this triple-node resection strategy offers a reproducible and scalable model, it may not completely prevent the development of compensatory lymphatic bypasses or spontaneous lymphatic regeneration, potentially leading to partial edema resolution over time. Alternative approaches, such as local irradiation or fascial dissection, have been used to more stringently inhibit lymphangiogenesis, but these techniques are technically challenging and require specialized equipment not always available in preclinical settings. Thus, although our surgical model balances technical feasibility with chronicity, the possibility of lymphatic rerouting and spontaneous recovery remains a limitation. Accordingly, our interpretation of

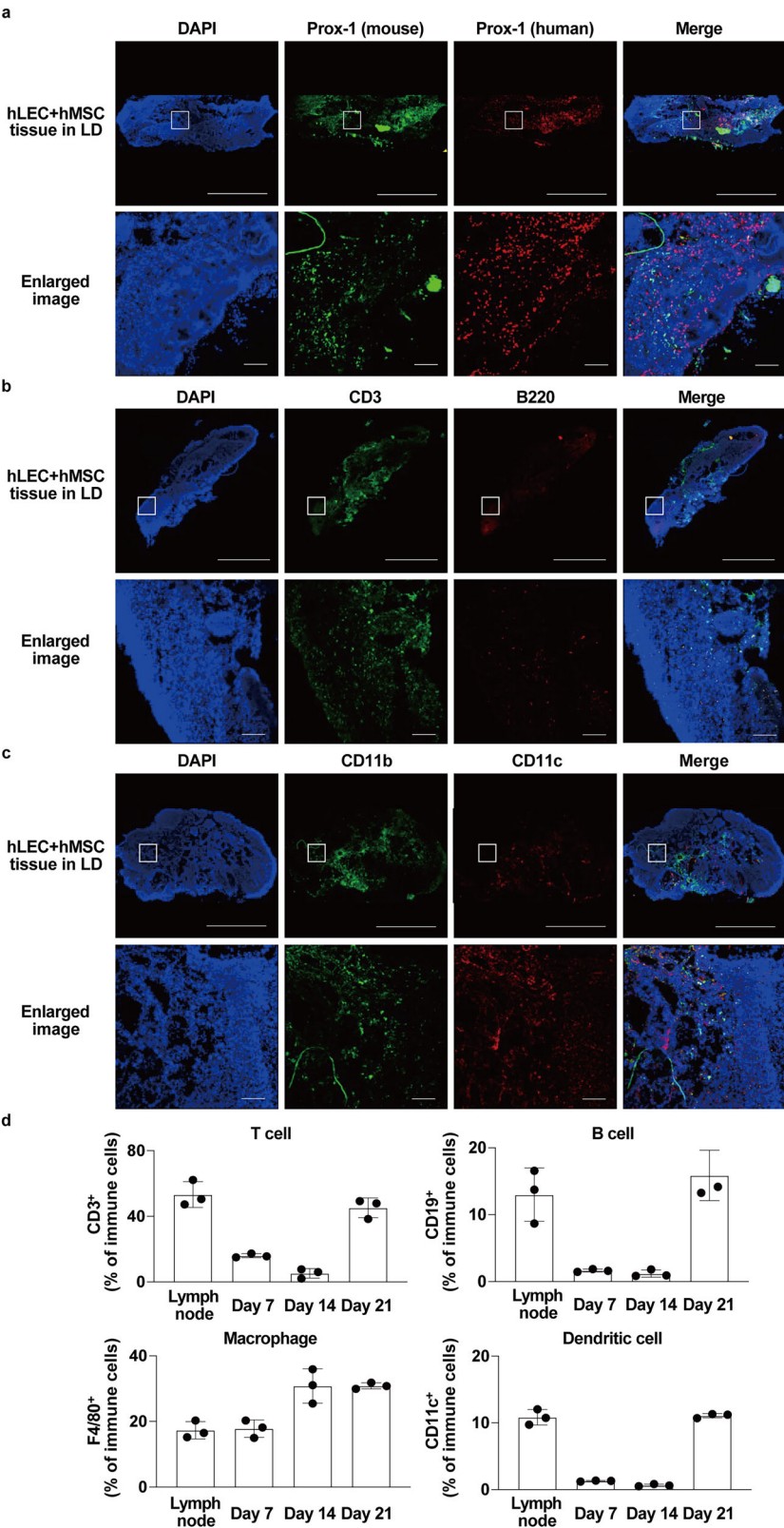

the current findings is deemed to be valid only within the constraints of this model, and we have sought to avoid overstatement by acknowledging its inherent limitations. Future studies combining additional interventions, including pharmacological treatments, lymphovenous anastomosis surgery, or the transplantation of VEGF-C overexpressing MSCs, may be required to more definitively suppress lymphatic regeneration and better mimic chronic lymphedema. To achieve the radical treatment of lymphedema, the reconstruction or regeneration of functional lymph nodes is required. In light of the current standard care for lymphedema, compression therapy and MSC transplantation have been employed and have demonstrated some degree of therapeutic effect in patients. In this regard, our findings indicate that these treatments did not result in the regeneration of lymph nodes, were only a temporary cure, and could not be considered a radical

**Fig. 8 | Distribution of cells in lymphatic node-like structures. a** Representative immunofluorescence staining for human- or mouse-reactive Prox-1 in a lymph node-like structure formed in hLEC+hMSC tissue-transplanted LD mice (day 21) of > 3 independent experiments. Blue, DAPI; Green, mouse-reactive Prox-1: Red, human-reactive Prox-1. Scale bars, 1 mm (upper panel) and 200 μm (lower panel). **b** Representative immunofluorescence staining for CD3 and B220 in a lymph node-like structure formed in hLEC+hMSC tissue-transplanted LD mice (day 21) of > 3 independent experiments. Blue, DAPI; Green, CD3: Red, B220. Scale bars, 1 mm (upper panel) and 200 μm (lower panel). **c** Representative immunofluorescence staining for CD11b and CD11c in a lymph node-like structure formed in hLEC+hMSC tissue-transplanted LD mice (day 21) of > 3 independent experiments. Blue, DAPI; Green, CD11b: Red, CD11c. Scale bars, 1 mm (upper panel) and 200 μm (lower panel). **d** Proportion of various immune cell populations in lymph nodes or lymph node-like structures of hLEC+hMSC tissue-transplanted LD mice (days 7, 14, and 21). CD3⁺ cells, CD19⁺ cells, CD11c⁺ cells, and F4/80⁺ cells were gated by SSC and FSC. Data represent the mean ± standard deviation from 3 independent experiments.

treatment (Supplementary Figs. 3, 6). Conversely, CeLyTs resulted in the reconstruction of a lymph node-like structure and long-term suppression of lymphedema, exhibiting a high therapeutic effect compared with conventional treatments over 100 days (Supplementary Fig. 13). To the best of our knowledge, this is the first study to demonstrate that cell transplantation can regenerate functional lymph nodes and suppress lymphedema over an extended period. CeLyTs composed of LECs and MSCs might reconstruct lymph nodes in LD patients and thus be a promising therapy for secondary lymphedema.

## Methods

### Materials and animals
The mMSC line C3H10T1/2 was obtained from Japanese Collection of Research Bioresources (JCRB) (Osaka, Japan, JCRB9080). The mouse fibroblast cell line NIH3T3 (fibroblast) was obtained from RIKEN Bioresource Center (RIKEN BRC) (Ibaraki, Japan, RCB2767). Human primary adipose-derived stem cells (hMSC) were kindly provided by Rohto Pharmaceutical Co., Ltd. (Osaka, Japan). The hMSCs were isolated from adipose tissue from a single donor. Human primary lymphatic endothelial cells (hLEC) were purchased from Takara Bio Inc. (Tokyo, Japan, C-12216, lot: 467Z001.2). The hLECs were isolated from juvenile foreskin (different locations) from a single donor. Lenti-X 293 T cells were purchased from Takara Bio Inc. All other chemicals used were of the highest commercially available grade. Male nude BALB/c Slc-nu/nu mice (5–8 weeks old) were purchased from Sankyo Labo Service Co., Inc. (Tokyo, Japan) and maintained under specific pathogen-free conditions. All animals were housed under a 12 h light/ 12 h dark cycle at an ambient temperature of $23 \pm 2\,°C$ and a relative humidity of $50 \pm 10\%$. All animals were treated under anesthesia using isoflurane, and euthanized by excess anesthesia after experiments.

### Cell culture
mMSCs, firefly luciferase (fluc) gene-expressing mMSCs (mMSC/fluc), NanoLuc luciferase (Nluc) gene-expressing mMSCs (mMSC/Nluc), and green fluorescent protein (GFP) gene-expressing mMSCs (mMSC/GFP) cells were cultured in DMEM supplemented with 15% heat-inactivated FBS and 1% penicillin-streptomycin-L-glutamine solution. NIH3T3 and Lenti-X 293 T cells were cultured in DMEM supplemented with 10% heat-inactivated FBS and 1% penicillin-streptomycin-L-glutamine solution. hMSCs were cultured in R-STEM Medium for hMSC High Growth (Rohto Pharmaceutical Co., Ltd.). hLECs and hLEC/GFP were cultured in Endothelial Cell Media MV2 (Takara Bio Inc.) and maintained in a humidified atmosphere containing 5% $CO_2$ at 37 °C.

### Establishment of hLEC/ green fluorescent protein (GFP)
Lenti-X 293 T cells ($1 \times 10^7$ cells) were seeded in a 10 cm culture dish and cultured overnight at 37 °C in humidified air containing 5% $CO_2$. Lenti-X 293 T cells were co-transfected with the pLVSIN-GFP plasmid and the Lentiviral Packaging Mix (Takara Bio Inc.) by the calcium phosphate method. After 24 h, the medium was replaced with culture medium containing 10 μM forskolin, and cells were cultured for an additional 48 h. Then, the supernatant was collected and centrifuged at $900 \times g$ for 5 min. Furthermore, the supernatant was filtered, mixed with PEG solution (32 w/v % polyethylene glycol #6000, 400 mM NaCl, and 40 mM HEPES), and centrifuged at $2500 \times g$ for 40 min. The precipitated viruses were collected by Opti-MEM (Thermo Fisher

Scientific, Inc., Waltham, MA, USA) and transduced into hLECs. After 24 h, the medium was replaced, and the cells were cultured in normal culture medium for a few days. These cells were cloned and incubated until they were confluent. hLEC/GFP were observed with BZ-X800 fluorescence microscope (Keyence, Osaka, Japan).

### Fabrication of bioengineered lymphatic tissues by cell stacking
mMSCs, hMSCs, mMSC/GFP, mMSC/Nluc, carboxyfluorescein succinimidyl ester (CFSE)-labeled mMSCs (mMSC/CFSE), or NIH3T3 ($7.5 \times 10^5$ cells) were collected with a trypsin-ethylenediaminetetraacetic acid (EDTA) solution (Nacalai Tesque Inc., Kyoto, Japan), seeded into 24-well Transwell inserts (Corning, NY, USA), and set into a culture plate. The culture plate with seeded cells was immediately centrifuged under various conditions (centrifugation speed from $50 \times g$ to $1000 \times g$ and centrifugation time from 10 s to 3 min) or $750 \times g$ and 30 s to settle the cells at the bottom of the inserts. After 24 h-incubation, hLECs, hLEC/CFSE, hLEC/GFP, mMSC/Nluc, and mMSC/fluc ($2 \times 10^5$ cells) were gently seeded on the layered MSCs and settled down on the layered cells in the inserts using the same procedure. After another 24 h-incubation, mMSCs, hMSCs, mMSC/GFP, and mMSC/Nluc or mMSC/CFSE ($7.5 \times 10^5$ cells) were gently seeded on LECs and MSCs and settled down on the layered cells in the inserts using the same procedure. Then, the cells were cultured for 5 days with changes of medium.

### Fabrication of bioengineered tissues by cell coating
Fibronectin and gelatin coating of cells was performed as previously reported[21]. Briefly, NIH3T3 cells ($1 \times 10^6$ cells) or mMSCs/Nluc ($2 \times 10^5$ cells) were collected with a trypsin-EDTA solution and suspended in 0.04 mg/mL fibronectin (Sigma-Aldrich, St. Louis, MO, USA) in 50 mM Tris-HCl buffer (pH 7.4) and incubated for 1 min with gentle rotation. Subsequently, the cells were suspended in 0.04 mg/mL of gelatin (Sigma-Aldrich) in 50-mM Tris-HCl buffer at pH 7.4, and incubated for 1 min with gentle rotation. After each procedure, the cells were washed with 50 mM Tris-HCl buffer (pH 7.4). This procedure was repeated four times, and cells were finally coated with fibronectin. After suspending the fibronectin and gelatin-coated NIH3T3 cells in DMEM, they were seeded into Transwell inserts. After 24 h-incubation, the fibronectin and gelatin-coated mMSCs/Nluc were seeded on the fibronectin and gelatin-coated NIH3T3 cells and attached to the bottom of the inserts by the same procedure. After another 24 h-incubation, the fibronectin and gelatin-coated NIH3T3 cells were gently seeded on the fibronectin and gelatin-coated mMSCs/Nluc and attached to the bottom of the inserts using the same procedure. Then, the cells were cultured for 5 days with changes of medium.

### Characterization of the bioengineered tissues
The survival of cells in the bioengineered tissues was evaluated by analyzing CFSE stained cells, measuring the luciferase activity of Nluc in the culture supernatant, and in vivo imaging of fluc-expressing cells. For CFSE staining, cells were prepared by incubation in medium containing 10 μM CFSE for 30 min, and the bioengineered tissues containing CFSE stained cells were observed with a BZ-X800 fluorescence microscope (Keyence). To measure luciferase activity, the culture supernatants of bioengineered tissues containing Nluc-expressing cells were collected daily and their luciferase activity was measured with a 2104 EnVision multilabel plate reader (PerkinElmer, Waltham, MA, USA). For in vivo imaging, the luminescence signals of

bioengineered tissues containing fluc-expressing cells were detected with VivoGLo Luciferin (Promega, Madison, WI, USA) using a bioluminescence imaging system (In-Vivo Xtreme II, Bruker, Billerica, MA, USA). The structure of the engineered tissues was evaluated by observing the bioengineered tissue composed of CellTracker™ Green (Thermo Fisher Scientific, Inc.)-labeled mMSCs and CellTracker™ Orange (Thermo Fisher Scientific, Inc.)-labeled hLECs using a Leica SP8 laser scanning confocal microscope with LAS X Life Science software (Leica Microsystems, Wetzlar, Germany) or a BZ-X800 fluorescence microscope (Keyence). mMSCs and hLECs were prepared by incubation with medium containing 10 μM CellTracker™ Green or Cell-Tracker™ Orange for 30 min, respectively.

## Preparation of paraffin sections and cryosections
Bioengineered tissues fabricated in Transwell inserts were fixed with 4% paraformaldehyde (PFA). The bottoms of the Transwell inserts were cut out, embedded in paraffin, and sectioned at 5 μm thickness. The preparation of paraffin sections was performed by Biopathology Institute Co. Ltd. (Oita, Japan). The paraffin sections were observed using a BZ-X800 fluorescence microscope (Keyence). The transplanted bioengineered tissues were excised using scissors, embedded in O.C.T. compound (Sakura Finetek, Torrance, CA, USA), and frozen using liquid nitrogen. The frozen tissues were stored at −80 °C until use. The frozen sections were cut to 7 μm thickness using a CM3050 S cryostat (Leica Biosystems, Wetzler, Germany).

## Immunohistochemical staining
Bioengineered tissues and cryo-sectioned tissues were fixed with 4% PFA, permeabilized with methanol, dimethyl sulfoxide, and a hydrogen peroxide mixture (6:1:1), and blocked with 1% bovine serum albumin for immunofluorescence staining. Bioengineered tissues and cryo-sectioned tissues were stained using primary antibodies and corresponding secondary antibodies. Nuclei were stained with Vectashield Antifade Mounting Medium with DAPI (Vector Laboratories Inc., Burlingame, CA, USA). Immunofluorescence images were captured using a Leica SP8 laser scanning confocal microscope with a LAS X Life Science software (Leica Microsystems). The primary and secondary antibodies were as follows: 1:200 PROX-1 (Proteintech, Rosemont, IL, USA, 11067-2-AP), 1:500 mouse PROX-1 (Abcam, Cambridge, UK, ab101851), 1:500 human PROX-1 (Bio-Techne, Minneapolis, MN, USA, AF2727), 1:500 CD31 (Bio-Techne, NB600-1475), 1:500 mouse CD31 (Abcam, ab256569), 1:500 human CD31 (Abcam, ab76533), 1:100 B220/CD45R (Bio-Techne, NBP2-53303), 1:100 CD3 (Arigo, Zhubei, Taiwan, ARG22819), 1:50 CD11b (Abcam ab8878), 1:50 CD11c (Thermo Fisher Scientific, PA5-90208), 1:500 Collagen Type I (Proteintech, 14695-1-AP), 1:400 LYVE-1 (Thermo Fisher Scientific, BS-1311R), 1:500 VEGFR-3 (Thermo Fisher Scientific, BS-1083R), 1:50 HuNuC (Sigma-Aldrich, MAB1281C3), 1:400 CD45 (Thermo Fisher Scientific, 56-0451-82), 1:100 CD3 (Bio-Techne, FAB4841G), 1:100 CD19 (Bio-Techne, NBP2-24965AF647), 1:100 F4/80 (Cell Signaling Technology Inc., Danvers, MA, USA, 52267), 1:200 CD11c (Bio-Techne, FAB69501R), 1:500 anti-rabbit Alexa 488 (Thermo Fisher Scientific, A21206), 1:500 anti-rat Alexa 488 (Thermo Fisher Scientific, A-11006), 1:500 anti-rabbit Alexa 647 (Thermo Fisher Scientific, A31573), and 1:500 anti-goat Alexa 647 (Thermo Fisher Scientific, A21447).

## mRNA expression in centrifuge-based bioengineered lymphatic tissues (CeLyTs)
The mRNA expressions of Podoplanin (*PDPN*, also known as *GP38* and *T1α*) and vascular endothelial growth factor (VEGF) receptor 3 (VEGFR-3) (*FLT4*) in CeLyTs were detected by quantitative real-time RT-PCR. Briefly, total RNA was isolated from the CeLyTs using Sepasol-RNA I Super G (Nacalai Tesque, Inc.). The cDNA was converted from the obtained RNA using ReverTra Ace qPCR RT Master Mix with gDNA Remover (Toyobo Co., Ltd., Osaka, Japan). The PCR fragments of each

cDNA were amplified by quantitative real-time RT-PCR using the following primers were synthesized by FASMAC (Kanagawa, Japan): human Podoplanin (forward primer: 5′-GAAGGTGTCAGCTCTGCTCT-3′; reverse primer: 5′-ACGTTGGCAGGGCGTAA-3′), human VEGFR3 (forward primer: 5′-AGCTCTCAGAGCTCAGAAGAG-3′; reverse primer: 5′-TTCTCTCTCTCTGCTTCAGCT-3′) and human GAPDH (forward primer: 5′-GCACCGTCAAGGCTGAGAA-3′; reverse primer: 5′-GCCTTC TCCATGGTGGTGAA-3′).

**Evaluation of the viability of LECs in CeLyTs**. mMSCs were collected with a trypsin-ethylenediaminetetraacetic acid (EDTA) solution (Nacalai Tesque Inc.), seeded into 24-well Transwell inserts (Corning), and set into the culture plate. The culture plate with seeded cells was immediately centrifuged at $750 \times g$ and 30 s to settle the cells at the bottom of the inserts. After 24 h-incubation, carboxyfluorescein succinimidyl ester (CFSE, Sigma-Aldrich Co.)-labeled hLECs (hLECs/CFSE) ($2 \times 10^5$ cells) were gently seeded on the layered mMSCs and settled down on the layered cells in the inserts using the same procedure. After another 24 h-incubation, mMSCs ($7.5 \times 10^5$ cells) were also gently seeded on hLECs/CFSE and mMSCs and settled down onto the layered cells in the inserts using the same procedure. Then, the cells were cultured for 5 days with changes in medium. The survival rate of hLECs/CFSE within CeLyTs was evaluated by flow cytometry using a BD FACSLyric (BD Biosciences Pharmingen, San Diego, CA, USA).

**Formation of a lymphatic network by LECs in the presence of collagen 1A2**. To evaluate changes in collagen expression associated with MSC sheet formation, $1 \times 10^6$ mMSC sheets were prepared. Immunofluorescence staining was performed using collagen type I (Proteintech Group, Inc.). Immunofluorescence images were captured using the 3D sectioning function of a BZ-X800 fluorescence microscope (Keyence) or Leica SP8 laser scanning confocal microscope with LAS X Life Science software (Leica Microsystems). Quantification of fluorescence was performed by BD FACSLyric (BD Biosciences Pharmingen). Then, $2 \times 10^5$ hLECs labeled with 10 mM CellTracker™ Green (Thermo Fisher Scientific) were seeded onto mMSC sheets to evaluate the lymphatic network formation ability of LECs. hLECs on mMSC sheets were observed by a Leica SP8 laser scanning confocal microscope with LAS X Life Science software (Leica Microsystems).

## Evaluation of bioengineered tissue survival after transplantation
Bioengineered tissues containing Nluc-expressing cells were subcutaneously transplanted to normal mice or the site of the removed popliteal lymph node in the right lower limb of LD mice. To detect the luminescence of cells, 5 mg Nano-Glo Luciferase Assay System (Promega) dissolved in 100 μL PBS was injected into the transplantation site of mice, and luminescence was detected using a bioluminescence imaging system (In-Vivo Xtreme II, Bruker). In addition, the intensity of the region of interest signal at the transplantation site was quantified using Bruker Molecular Imaging software with an In-Vivo Xtreme II system (Bruker). For the quantitative evaluation of cell survival, blood was collected daily from the posterior facial vein of mice after transplantation of bioengineered tissues containing Nluc-expressing cells, and the luciferase activity in the blood was measured using a 2104 EnVision multilabel plate reader (PerkinElmer).

## Evaluation of lymph flow and lymphedema size in LD mice
LD mice were induced by removing the popliteal and inguinal lymph nodes in the right lower limbs of mice as previously described[53]. Briefly, skin near the inguinal region of the right lower limb of mice was cut and the popliteal and inguinal lymph nodes were removed. Bioengineered tissues, suspended mMSC/Nluc ($2 \times 10^6$ cells), or suspended hLEC ($2 \times 10^6$ cells) were transplanted to the site of the removed popliteal lymph node. Thirty μg indocyanine green (ICG, MP Biomedicals, Irvine, CA, USA) dissolved in 30 μL PBS was administered

into the footpad of the right lower limb of mice 1, 14, and 21 days after transplantation of the bioengineered tissues. Then, fluorescence at the site of the removed popliteal lymph nodes (the transplantation site) was detected 1 h after ICG administration using In-Vivo Xtreme II (Bruker). In addition, lymph nodes or lymph node-like structures formed by transplantation of the bioengineered tissues were excised from normal mice or LD mice on day 21, and the fluorescence intensity of the homogenates was measured by a 2104 EnVision multilabel plate reader (PerkinElmer). For the qualitative evaluation of lymphedema in LD mice, the accumulation of interstitial fluid in the paws of LD mice with or without transplantation of cells or bioengineered tissues was observed by a BZ-X800 fluorescence microscope (Keyence). For quantitative evaluation, the thicknesses of the paws and legs of the right lower limbs of LD mice with or without transplantation of cells or bioengineered tissues were measured and the change rate relative to day 0 was calculated.

**Therapeutic effect of CeLyTs on lymphadenectomy (LD) mice.** LD mice were induced by removing the popliteal and inguinal lymph nodes in the right lower limbs of mice. Therapeutic effects were evaluated by implanting CeLyTs into lymph node-impaired mice 5 days after lymph node removal. The effect of improving lymph flow and the size of edema were evaluated. The lymph nodes or lymph node-like structures formed by the transplantation of CeLyTs were excised from normal mice or LD mice on day 33 after the administration of 30 μg ICG dissolved in 30 μL PBS, and the fluorescence was detected 1 h after ICG administration using a bioluminescence imaging system (In-Vivo Xtreme II, Bruker). For the qualitative evaluation of lymphedema in LD mice, the accumulation of interstitial fluids to the paws of LD mice with or without transplantation of cells or bioengineered tissues was observed by a BZ-X800 fluorescence microscope (Keyence). For quantitative evaluation, the thickness of the paws and legs of the right lower limbs of LD mice with or without transplantation of cells or bioengineered tissues was measured and the change rate to day 0 was calculated.

**Compression therapy for LD mice.** Following the removal of the lymph nodes from the mice, the paws of the LD mice were bandaged with medical surgical tape at high pressure every 3 days for 21 days as a form of compression therapy. To quantify the efficacy of this treatment, the thickness of the paws of the right lower limbs of LD mice with compression therapy was measured and the change rate to day 0 was calculated.

**Therapeutic effect of CeLyTs on chronic lymphedema model mice.** hLEC/hMSC spheroids were prepared by hLECs ($2 \times 10^5$ cells) and hMSCs ($1.5 \times 10^6$ cells) were seeded in 96 well ultra-low attachment plate (ThermoFisher Scientific) and incubated for 5 days. Chronic lymphedema model mice were induced by removing the iliac, popliteal, and inguinal lymph nodes in the right lower limbs of mice as previously described[53]. Under isoflurane anesthesia, a ~1.5 cm skin incision was made along the inguinal crease. Subcutaneous tissue and fascia were gently dissected to expose the lymph nodes. Vascular and lymphatic structures were carefully separated but not ligated, as bleeding and lymphorrhea were minimal and self-limiting in this model. The surgical site was irrigated with saline and closed using interrupted sutures. To confirm complete lymph node removal and lymphatic flow disruption, 1 mg/mL ICG was injected into the right footpad immediately after surgery, and lymphatic flow was assessed using the IVIS Lumina III (PerkinElmer). Bioengineered tissues were transplanted to the site of the removed popliteal lymph node. For quantitative evaluation, the thickness of the paws and legs of the right lower limbs of LD mice with or without transplantation of bioengineered tissues were measured for 42 days and the change rate to day 0 was calculated.

**Proportion of various immune cell populations in the dermis tissue.** Preparation of single-cell suspension from excised dermis was performed as previously reported[27]. Briefly, excised lymph nodes and lymph node-like structures were cut into small fragments and digested with DMEM/high glucose (not supplemented with antibiotics or FBS) containing 10 mg/mL dispase (Wako Pure Chemical Industries, Ltd., Osaka, Japan), 0.1 mg/mL collagenase P, and 0.1 mg/mL DNase I (both from Sigma-Aldrich). Cell suspensions were strained through a 70 μm filter and stained with fluorescently labeled antibodies. Data were acquired using a BD FACSLyric (BD Biosciences Pharmingen) and analyzed using FlowJo software version 10.7.2 (BD Biosciences Pharmingen).

**Lymphatic flow in the reconstructed lymph node-like structure.** CeLyTs were transplanted into LD mice. Then, fluorescence at the site of the removed popliteal lymph nodes (the transplantation site) was detected every 10 min after ICG administration on day 21 by an IVIS Lumina III (PerkinElmer) under continuous isoflurane anesthesia. In addition, the lymph nodes or lymph node-like structures formed by transplantation of the CeLyTs were excised from normal mice or LD mice on day 21. Then, fluorescence at the site of the lymph nodes or lymph node-like structures was detected immediately, 1, and 3 h after ICG administration using iBright FL1000 Imaging System (Thermo Fisher Scientific), and the fluorescence intensity of the homogenates was measured by a GloMax® Discover Microplate Reader (Promega). To evaluate the semi-circulatory function of lymph node-like structures, single-cell suspension was prepared from excised lymph nodes and lymph node-like structures formed by transplantation of the CeLyTs at immediately, 1 h, and 3 h after fluorescence-labeled beads (Fluo-Spheres, 0.1 μm, yellow-green, Thermo Fisher Scientific) administration into mice footpad. The total uptake of fluorescence-labeled beads in a monocyte population was calculated using a positive rate of fluorescein isothiocyanate (FITC). Data were acquired using a BD FACSLyric (BD Biosciences Pharmingen) and analyzed using FlowJo software version 10.7.2 (BD Biosciences Pharmingen).

**Measurement of cytokine production**

Bioengineered tissues were transplanted to the site of the removed popliteal lymph nodes of right lower limbs in LD mice. After 21 days, 10 nmol CpG1018 (sequence: 5′-TGACTGTGAACGTTCGAGATGA-3′ with a phosphorothioate-backbone) was synthesized by Integrated DNA Technologies (Coralville, IA, USA) and injected into the footpad. Three hours after injection, the popliteal lymph nodes or lymph node-like structures formed by transplantation of the bioengineered tissues with the lymphatic network were excised and homogenized. Then, the amount of interferon (IFN)-γ, interleukin (IL)-6, IL-12, and tumor necrosis factor (TNF)-α in the homogenates was quantified using mouse IFN-γ, IL-6, and IL-12/IL-23 (p40), and TNF-α ELISA MAX Deluxe Sets (BioLegend, San Diego, CA, USA).

**Flow cytometry analysis of the reconstructed lymph node-like structure.** Preparation of a single-cell suspension from excised lymph nodes and lymph node-like structures formed by transplantation of the CeLyTs was performed as previously reported[54]. Briefly, excised lymph nodes and lymph node-like structures were cut into small fragments and digested with RPMI-1640 medium containing 0.8 mg/mL dispase (Wako Pure Chemical Industries Ltd.), 0.2 mg/mL collagenase P, and 0.1 mg/mL DNase I (both from Sigma-Aldrich). Cell suspensions were strained through a 70 μm filter and stained with fluorescently labeled antibodies. Data were acquired using a BD FACSLyric (BD Biosciences Pharmingen) and analyzed using FlowJo software version 10.7.2 (BD Biosciences Pharmingen).

**Statistics & reproducibility**

Statistical differences were evaluated by one-way analysis of variance (ANOVA), followed by Dunnett's test for multiple comparisons or

Student's *t*-test for comparisons between two groups. Statistical significance was set at $p < 0.05$. Unless otherwise stated in the figure legends, each experiment was independently repeated at least three times with similar results. No statistical method was used to predetermine sample size. No data were excluded from the analyses. The experiments were not randomized. The investigators were not blinded to allocation during experiments and outcome assessment.

### Ethical Statement
The animal experiments were approved by the Institutional Animal Experimentation Committee of the Tokyo University of Science (approval number: Y23004, approved on August 8, 2024) and were conducted in accordance with the National Institutes of Health Guide for the Care and Use of Laboratory Animals and the ARRIVE guidelines. The sex of animals and cells was not considered in the study design, as it was not expected to affect the outcomes. No human participants or donors were involved in this study.

### Reporting summary
Further information on research design is available in the Nature Portfolio Reporting Summary linked to this article.

## Data availability
The data generated in this study are provided in the Source Data file included with this paper. Source data are provided as a Source Data file. No raw imaging data have been deposited in a public repository owing to file size and data management constraints. Requests for access to these imaging data should be directed to the corresponding authors, and will be fulfilled for academic research purposes within 4 weeks. The raw numerical data supporting the findings of this study are included in the Source Data file. Source data are provided with this paper.

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

## Acknowledgements

We would like to thank the Biopathology Institute Co., Ltd. for their kind support of some experiments, and Rohto Pharmaceutical Co., Ltd. for providing hMSCs. This work was supported by a Grant-in-Aid for Scientific Research (B) from the Japan Society for the Promotion of Science (grant number 23H03749 to K.K.), Project Seeds A from the Japan Agency for Medical Research and Development (grant number A409TS to M.M and K.K.), JST SPRING (grant number JPMJSP2151 to S.O.), a Noguchi Shitagau Research Grant from the Noguchi Institute (to K.K.), and Grant-in-Aid for JSPS Fellows (grant number 25KJ2110 to S.O.). We thank J. Ludovic Croxford, PhD, from Edanz for editing a draft of this manuscript. We deeply thank Professor James M. Wells of Cincinnati Children's Hospital Medical Center for his helpful comments and suggestions on the manuscript.

## Author contributions

M.M. and K.K. conceptualized the research. S.O. and K.K. performed the study investigation. S.O. performed the experiments. S.O., S.I., M.M., M.N., and K.K. reviewed the data. S.O., S.I., M.M., M.N., and K.K. wrote the manuscript. All authors read and revised the manuscript.

## Competing interests

The authors declare no competing interests.
