## [Transparent Peer Review file · Nature Communications]

Reconstruction of the lymphatic system by transplantation of a centrifuge-based bioengineered lymphatic tissue

Corresponding Author: Dr Kosuke Kusamori

A version of this paper was originally rejected for publication by Nature Communications, however that decision was reconsidered after appeal by the authors.

Version 2:

Reviewer comments:

Reviewer #1

(Remarks to the Author)

Reconstruction of the lymphatic system by transplantation of a centrifuge-based bioengineered lymphatic tissue

This study explores the potential of centrifuge-based bioengineered lymphatic tissue (CeLyTs), composed of lymphatic endothelial cells (LECs) and mesenchymal stem cells (MSCs), in reconstructing lymph node-like structures to treat secondary lymphedema. Using a lymphadenectomy (LD) mouse model, the authors demonstrate that CeLyTs can promote lymphatic flow restoration and reduce edema. The topic is of significant interest and holds great promise for advancing lymphedema treatment. However, the study's conclusions are weakened by insufficient evidence for lymph node reconstruction, inadequate functional validation, and limitations of the experimental model. Below are the primary concerns and specific suggestions for improvement.

General Concerns

1. Lack of structural evidence for lymph node reconstruction:

The study claims to have observed lymph node-like structure formation, but detailed histological evidence demonstrating key lymph node features such as cortex, medulla, follicles, and sinuses is lacking.

2. Limitations of the lymphadenectomy model:

The 21-day recovery period initiated by the LD model makes it difficult to differentiate CeLyTs-induced effects from spontaneous recovery, suggesting the need for a model that better mimics chronic lymphedema.

3. Unclear role of LECs:

Interpreting the results may present a challenge due to the weak-magnification images, which do not clearly show the position of the transplanted CeLyTs in relation to the overall tissue. Additionally, given that Prox-1 is a transcription factor, it might be expected to overlap with DAPI; however, the presence of lymphatic endothelial cells remains uncertain due to the staining of the cytoplasm.

4. Absence of functional validation:

Data demonstrating lymph node-like structures' semi-circulatory functions, such as fluid filtration and foreign particle clearance, are not available at this time.

Specific Comments

1. Evidence for lymph node reconstruction:

It would be beneficial to provide high-resolution histological and immunostaining data (e.g., CD20 for B cells, LYVE-1 for lymphatic vessels) to demonstrate lymph node-specific features.

Employing 3D imaging to visualize the spatial organization of lymphatic networks and lymph node-like structures could be a valuable approach.

2. Addressing model limitations:

Developing a chronic lymphedema model involving iliac lymph node dissection and radiation to prevent spontaneous recovery could be a valuable approach. Ensure a follow-up period of at least six weeks to distinguish natural healing from treatment effects.

Include additional control groups utilizing alternative 3D tissue-engineering methods to benchmark CeLyTs performance.

3. Validating LEC roles:

Conduct experiments comparing CeLyTs with and without LECs to directly assess LEC contributions. Track fluorescently labeled LECs to visualize their integration and functional involvement in lymphatic restoration.

4. Semi-circulatory function validation:

Use FITC-labeled beads to track lymphatic fluid inflow and outflow. Quantify filtration capacity by measuring foreign particle retention and clearance rates.

5. Improving data presentation:

Include wide-field images to illustrate CeLyTs positioning within host tissue. Clarify Prox-1 staining results to confirm lymphatic endothelial cell presence.

Reviewer #2

(Remarks to the Author)

This manuscript, "Reconstruction of the lymphatic system by transplantation of a centrifuge-based bioengineered lymphatic tissue" by Obana et al. describes a method for the generation of artificial lymphatic (lymph-node-like) tissue to treat secondary lymphedema after lymphadenectomy.

The authors describe a centrifugation-based technique, involving human lymphatic endothelial cells and murine mesenchymal stromal/stem cells. After characterization of the artificial tissue, they show the functionality in several in vivo experimental systems (LD mice). The authors conclude that the system, which they call "CeLyT" holds great potential for the treatment of secondary lymphedema.

The reviewer finds the story interesting, and believes that this study is significant for the field of lymphedema research. A solid body of qualitative and quantitative data allows for most of the conclusions drawn.

Some issues, however, should be addressed:

- 1) Why did the authors choose human lymphatic endothelial cells in combination with murine MSCs? Why not having both cell types from the same species? I think this deserves some sentences in the discussion section.
- 2) methodology: Were all experiments performed with one and the same LEC donor? Or did the authors use biological replicates (same for the MSCs?)
- 3) language: please check the whole manuscript in terms of "over-exaggerations", like "great potential (p2, line 41), or "strongly detected" (p4, line 112), or "dramatically increased" (p5, line 144). Such words come rather unscientific.
- 4) p4, line 109: "... a lymphatic endothelial cell marker Prox-1 was detected in these tissues...." - I would think, since LEC are in these tissues, that Prox-1 should be detected, since Prox-1 is a reliable marker for lymphatics.
- 5) p5, line 146: "in addition, the gene expression in hLEC+hMSCs increased...." - which gene expression? please specify.
- 6) p7, line 211: "LD mice were prepared...."? please reword. same as in line 216: "little therapeutic effect", similar in line 220 (".... a superior therapeutic efficacy....") and also at other locations throughout the manuscript.
- 7) p9, line 280: "This study successfully fabricated...." ... not the study, but you authors fabricated
- 8) finally, although the reviewer appreciates the work of many outstanding scientists from Japan, the authors are strongly advised to take care of a more balanced reference list, including also more groundbreaking work from the lab of Kari Alitalo (University Helsinki, Finland) and many others in his scientific surroundings, or the work from Wolfgang Holthöner (LBI Trauma, Austria, e.g. PMID: 28459049) or Ernst Reichmann (e.g. PMID: 24363089)

Reviewer #3

(Remarks to the Author)

The study is presenting an original and useful technology showing that three-layered bioengineered tissue composed of LECs and MSCs was successfully fabricated by the centrifugal cell stacking technique. In this study, the authors suggest that cell transplantation can regenerate functional lymph nodes and suppress lymphedema over an extended period. CeLyTs composed of LECs and MSCs might reconstruct lymph nodes in LD patients and thus be a promising therapy for secondary lymphedema.

This study is very important and could lead to significant therapeutic breakthroughs. However, the experimental data still need to be greatly expanded to better understand how the system is established.

It should be specified in the introduction that the leading cause of secondary lymphedema is parasitic infection due to filariasis. Lymphadenectomy is the primary cause in industrialized countries.

The authors should mention the work from Saaristo with the Lymfactin assay that was promising but did not provide significant improvement of lymphedema with VEGFC delivery (doi: 10.1097/PRS.00000000000011675.).

In Fig 2d, Prox1 staining is cytoplasmic, whereas Prox1 is a transcription factor that exhibits a nuclear localization. The authors should reproduce the staining and add another lymphatic marker such as VEGFR3 to validate sup. Fig2 data. Higher magnification showing the cell shape would be appreciated.

Figure 3c: statistic are needed to validate the data.

Figure 3d and e: A quantification of the cell viability should be provided, after how many days of culture the pictures were taken? This has to be mentioned in the manuscript (same comment for suppl. Fig. 1).

In figure 4, the authors transplanted the CeLyTs subcutaneously in mice. They claim that mMSC/Nluc cells and mMSC/Nluc+NIH3T3 tissues disappeared 3–4 days after transplantation, and those of mMSC/Nluc+hMSC and hLEC+mMSC/Nluc tissues persisted for at least 21 days. However, it seems that at least 5 times less mMSC/Nluc cells and mMSC/Nluc+NIH3T3 material was implanted into mice, which can explain that they disappeared faster. The author should start with equivalent signal in all groups.

Also, the imaging in figure 4d should be improved: lower magnification of the skin should be provided to show all the layers (epidermis/dermis). Importantly, there is a lot of lymphatic vessels into the normal skin. Why do they do not show any? It would be important (crucial) to see if the transplanted CeLyTs may connect to the preexisting network. How do they differentiated endogenous LEC from transplanted ones? Confocal microscopy is needed to answer this question.

Next, the authors evaluated the effect of CeLyTs on the lymphatic flow using ICG footpad injection.

The figure 5 needs control mice with no surgery to compare with basal diffusion of the dye.

We can clearly observe a perfusion of the lymph node-like structure. However, it would be interesting to see if it connects to the lymphatic network and if distant lymph nodes are perfused. The imaging detection might be too limited to observe fluorescence in distant lymph nodes such as axillary or mesenteric nodes across the skin. The authors should dissect them and image them independently. Is there a lymphatic flow?

Also one of the major lymphedema hallmark is dermis thickening induced by chronic inflammation. Do the authors observe a decrease of dermis thickness in the presence of the lymph node-like structure?

Do they observe changes in immune cell populations in the skin?

What are the cell components of the lymph node-like structures after 7, 14, 21 days? A better histological description is recommended. Stainings are not convincing, the authors should provide better characterization by flow cytometry and histology of lymphocytes, macrophages and dendritic cells double stained with lymphatics and quantifications are needed. As many cell therapies failed due to the lack of revascularization of the transplant. Does better results can be observed with VEGFC-overexpressing MSC? Is the system drug-sensitive? What is the impact on adipose tissue accumulation?

Finally, the authors overestimate their conclusions as, so far, the experiments have only been conducted in immunodeficient mice due to the human origin of the transplanted cells.

Also, simple lymph node transplant is not enough to cure lymphedema because the condition involves more than just the absence of lymph nodes; it is a complex disease involving the entire lymphatic system and the surrounding tissues. The authors should discuss about the existing lymphatic vessels that are non-functional or severely damaged. In that context the transplanted nodes may not integrate effectively into the system.

Over time, lymphedema causes chronic changes in the tissues, such as fibrosis (thickening and scarring), fat deposition, and inflammation. These changes reduce tissue elasticity and lymphatic function, making it harder for transplanted nodes to address the underlying problem. Even if some drainage is restored, these structural changes may persist and contribute to ongoing swelling and complications. The authors should discuss this point.

The authors should also discuss about the advantages compare to lymph node transplantation, especially in combination with other treatments like vascularized lymph node transfer (VLNT) or lymphovenous bypass surgery, that is not yet a standalone cure.

Minor points:

What is the origin of hLEC? It is not described in the method.

Why some parts of the manuscript are written in a red color?

Version 3:

Reviewer comments:

Reviewer #1

(Remarks to the Author)

Comment:

As a reviewer, I have carefully re-evaluated the authors' response and revised manuscript, and I must respectfully note that a central concern remains insufficiently addressed.

In my original review, I explicitly requested that the authors perform cell tracking experiments using fluorescent labeling or an equivalent method to directly demonstrate that the transplanted lymphatic endothelial cells (LECs) contributed to the reconstructed tissue.

This level of evidence is a fundamental requirement in regenerative medicine for establishing causal relationships between

transplanted cells and tissue regeneration.

However, the authors have only presented indirect observations using Prox-1 immunostaining, without conducting any direct tracking experiments for the transplanted LECs.

Moreover, the resolution of the immunofluorescence images is insufficient for reliable identification and localization of cells.

Therefore, the current data do not adequately support the authors' claim that transplanted LECs contributed to tissue reconstruction.

I strongly encourage the authors to understand the core of the comment and to respond with scientific rigor and sincerity, beyond a merely formal reply.

Reviewer #2

(Remarks to the Author)

The authors adequately responded to the concerns of the reviewer. The quality of the manuscript has improved. I recommend the manuscript for publication.

Reviewer #3

(Remarks to the Author)

The manuscript has been significantly improved thanks to the comprehensive set of experiments provided by the authors. The authors have carried out all the requested experiments and have addressed the points that required clarification. In conclusion, I give a favorable recommendation for the publication of this article.

Version 4:

Reviewer comments:

Reviewer #1

(Remarks to the Author)

Thank you for your revisions. However, several critical issues remain unaddressed:

1. Insufficient structural evidence of lymph node reconstruction

Key lymph node features (cortex, medulla, follicles, sinuses) are not demonstrated by histology or immunostaining (e.g., CD20, LYVE-1). Additional 3D imaging or higher-resolution sections are required.

2. Optical quality and raw data of GFP/Prox-1 images

Supplementary Figure 9 remains out of focus, preventing clear visualization of nuclear contours and lymphatic lumina. Please re-acquire images with a high-NA objective, appropriate z-stacks, and provide the unprocessed TIFF stacks as Source Data.

3. Lack of direct tracking of transplanted LECs

Reliance on Prox-1 staining alone is insufficient. Direct fluorescent or genetic cell-tracking experiments are needed to confirm that transplanted LECs integrate and function within the reconstructed tissue.

4. Chronic lymphedema model validity

A 42-day follow-up without fibrosis markers (Masson's trichrome, α -SMA) or observation beyond eight weeks does not convincingly model chronic lymphedema. Please include additional markers and extend the follow-up period.

5. Confounding by spontaneous recovery

Edema resolves spontaneously around day 50. Without irradiation or fascial blockade, therapeutic effects may be overestimated. A modified model to suppress spontaneous lymphatic regrowth should be considered.

6. Opaque surgical methodology

The protocol for iliac lymph-node excision lacks detail: incision lines, tissue planes, ligation of vessels/lymphatics, closure steps, and intraoperative photos or schematics are missing. Immediate Evans Blue or NIRF mapping data confirming complete node removal and flow blockade should be provided.

7. Absence of functional validation of lymph-node-like activity

No quantitative assay of inflow, filtration, or clearance (e.g., ICG or FITC-bead assays) is presented. Please include high-resolution, quantitative functional data demonstrating true lymph-node-like function.

8. Improvement of data presentation

Provide wide-field images showing the position of the bioengineered tissue within the host and clarify the spatial relationship of Prox-1 relative to DAPI.

Version 5:

Reviewer comments:

Reviewer #2

(Remarks to the Author)

Response to reviewers' comments

First, we would like to express our sincere gratitude to the reviewers for their insightful comments and constructive suggestions. Their feedback has greatly contributed to refining the structure-function evaluation of lymph node-like structures and has provided valuable insights into the limitations of our study. In response to the reviewers' comments, we conducted additional experiments and made substantial revisions to our manuscript. Below, we provide our point-by-point responses to each comment.

Reviewers' comments:

Reviewer #1:

Reconstruction of the lymphatic system by transplantation of a centrifuge-based bioengineered lymphatic tissue

This study explores the potential of centrifuge-based bioengineered lymphatic tissue (CeLyTs), composed of lymphatic endothelial cells (LECs) and mesenchymal stem cells (MSCs), in reconstructing lymph node-like structures to treat secondary lymphedema. Using a lymphadenectomy (LD) mouse model, the authors demonstrate that CeLyTs can promote lymphatic flow restoration and reduce edema. The topic is of significant interest and holds great promise for advancing lymphedema treatment. However, the study's conclusions are weakened by insufficient evidence for lymph node reconstruction, inadequate functional validation, and limitations of the experimental model. Below are the primary concerns and specific suggestions for improvement.

General Concerns:

1. Lack of structural evidence for lymph node reconstruction:

The study claims to have observed lymph node-like structure formation, but detailed histological evidence demonstrating key lymph node features such as cortex, medulla, follicles, and sinuses is lacking.

Specific Comments:

1. Evidence for lymph node reconstruction:

It would be beneficial to provide high-resolution histological and immunostaining data (e.g., CD20 for B cells, LYVE-1 for lymphatic vessels) to demonstrate lymph node-specific features.

Employing 3D imaging to visualize the spatial organization of lymphatic networks and lymph node-like structures could be a valuable approach.

Response:

We appreciate the reviewer's insightful comment. To address this concern, we conducted additional experiments and incorporated new images in Figure 8a–c and Supplementary Figure 9e, showing an overview of a lymph node and hLEC+hMSC tissue in LD. The distribution of LECs and immune cells exhibited an overall lymph node-like structure. Macrophages were predominantly localized in the center of the tissue, and this distribution was similar to the medullary zone of the lymph nodes. While distinct B cell and T cell regions in the cortex were not clearly observed, histological analysis confirmed the presence of follicular-like structures. These findings suggest that the observed lymph node-like structures represent immature lymph nodes, containing lymphatic vessels and initiating the formation of lymph node-specific features such as follicular structures, cortex, and medulla.

Revisions corresponding to these findings have been appropriately incorporated into the Results, Discussion, Figure legends, and Supplementary Figure legends sections, with changes marked in red in the revised manuscript.

(Results, Line 293-)

Furthermore, the distribution of cells within the lymph node-like structures formed after hLEC+hMSC tissue transplantation was evaluated. Various cells, including LECs and immune cells, were distributed throughout the overall lymph node-like structures, and follicle-like structures were beginning to form, which exhibited an immature structure similar to that of lymph nodes (Fig. 8a-c, Supplementary Fig. 9e).

(Discussion, Line 398-)

Lymph nodes can be characterized by their active immune response mediated by abundant immune cells and lymphatic inflow and outflow [45-48]. The transplantation of CeLyTs into LD mice formed a lymph node-like structure containing a wide range of immune cells, which promoted an immune response to CpG1018, and the inflow and outflow of ICG (Fig. 7c,d, Supplementary Fig. 9 and Supplementary Fig. 10a). Furthermore, the lymph node-like structures were connected to distant lymph nodes and integrated into a lymphatic network with filtration capacity (Supplementary Fig. 10b-d). These results indicate that the lymph node-like structure functions as a lymph node. Although the lymph node-like structures were immature compared with the lymph nodes, they contained lymphatic vessels, and lymph node-specific structures such as follicular structures, cortex, and medulla were beginning to form (Fig. 8a-c, Supplementary Fig. 9e). The lymph node-like structures were missing structures that exist in mature lymph nodes, such as a T cell zone, B cell zone, and follicular structures. However, the function of the lymph node-like structures was almost equivalent to those of lymph nodes, and LECs and various

immune cells were present, indicating that the lymph node-like structures were at a stage prior to maturation. Future studies are needed to elucidate the mechanisms underlying the maturation of lymph node-like structures.

(Figures)

Figure 8a-c

(Figure legends, Line 996-)

Fig. 8. Distribution of cells in lymphatic node-like structure.

a, Representative immunofluorescence staining for human- or mouse-reactive Prox-1 in a lymph

node-like structure formed in hLEC+hMSC tissue-transplanted LD mice (day 21). Blue, DAPI; Green, mouse-reactive Prox-1; Red, human-reactive Prox-1. Scale bars, 1 mm (upper panel) and 200 μ m (lower panel). **b**, Representative immunofluorescence staining for CD3 and B220 in a lymph node-like structure formed in hLEC+hMSC tissue-transplanted LD mice (day 21). Blue, DAPI; Green, CD3; Red, B220. Scale bars, 1 mm (upper panel) and 200 μ m (lower panel). **c**, Representative immunofluorescence staining for CD11b and CD11c in a lymph node-like structure formed in hLEC+hMSC tissue-transplanted LD mice (day 21). Blue, DAPI; Green, CD11b; Red, CD11c. Scale bars, 1 mm (upper panel) and 200 μ m (lower panel). **d**, Proportion of various immune cell populations in lymph nodes or lymph node-like structures of hLEC+hMSC tissue-transplanted LD mice (days 7, 14, and 21). CD3⁺ cells, CD19⁺ cells, CD11c⁺ cells, and F4/80⁺ cells were gated by SSC and FSC. Data represent the mean \pm standard deviation (n=3).

(Supplementary Figures)

Supplementary Figure 9e

(Supplementary Figure legends, Line 284-)

Supplementary Fig. 9. | Analysis of lymph node-like structures after transplantation of CeLyTs.

e, Paraffin-section images of a lymph node and lymph node-like structure formed in hLEC+hMSC tissue-transplanted LD mice (day 21). Scale bars, 500 μ m.

General Concerns:

2. Limitations of the lymphadenectomy model:

The 21-day recovery period initiated by the LD model makes it difficult to differentiate CeLyTs-induced effects from spontaneous recovery, suggesting the need for a model that better mimics chronic lymphedema.

Specific Comments:

2. Addressing model limitations:

Developing a chronic lymphedema model involving iliac lymph node dissection and radiation to prevent spontaneous recovery could be a valuable approach. Ensure a follow-up period of at least six weeks to distinguish natural healing from treatment effects.

Include additional control groups utilizing alternative 3D tissue-engineering methods to benchmark CeLyTs performance.

Response:

We appreciate the reviewer's meaningful comment. To address this, we conducted additional experiments and included data on changes in paw and leg size in bioengineered tissue-transplanted chronic lymphedema model mice, presented in Supplementary Figure 7. Due to technical limitations, it was challenging to establish a chronic lymphedema model using radiation. Instead, we induced chronic lymphedema by removing the iliac, inguinal, and popliteal lymph nodes as previously described (Nakajima, Y., *et al*, *Sci Rep*, 2018. Reference number 49) and subsequently evaluated the therapeutic effect of CeLyTs over a six-week period. Additionally, as controls for alternative 3D tissue-engineering methods, we included hLEC/hMSC spheroids and hMSC+NIH3T3 tissues fabricated using the cell coating method. CeLyTs exhibited a therapeutic effect beginning 10–14 days after transplantation and completely suppressed lymphedema for 28 days in chronic lymphedema model mice. The therapeutic efficacy was superior to that of bioengineered tissues fabricated using other 3D tissue-engineering methods. However, the therapeutic effect of CeLyTs in chronic lymphedema model mice took longer to manifest compared to LD mice. We hypothesize that CeLyTs transplantation in LD mice led to an earlier therapeutic effect due to its direct connection with the iliac lymph nodes. In contrast, in chronic lymphedema model mice, a more severe model of lymphedema, bypass formation may be more difficult. Furthermore, natural healing was observed in chronic model mice, albeit at a lower rate than in LD mice. As a result, it is difficult to completely differentiate between the therapeutic effect and spontaneous recovery, which has been acknowledged as a study limitation in the Discussion section.

Revisions corresponding to these findings have been appropriately incorporated into the Results, Discussion, Supplementary Methods, and Supplementary Figure legends sections, with changes marked in red in the revised manuscript.

(Results, Line 250-)

CeLyTs also exhibited therapeutic effects in a more severe chronic lymphedema model, which was prepared by removing the iliac, inguinal, and popliteal lymph nodes. Moreover, they demonstrated a greater lymphedema-suppressive effect compared with bioengineered tissues fabricated by other tissue engineering methods, such as hLEC/hMSCs spheroids and hMSC+NIH3T3 tissues generated by the cell coating method (Supplementary Fig. 7).

(Discussion, Line 437-)

In addition, CeLyTs exhibited a therapeutic effect that emerged 10–14 days after transplantation and completely suppressed lymphedema for 28 days in chronic lymphedema model mice, in which the iliac lymph nodes were removed in addition to the inguinal and popliteal lymph nodes (Supplementary Fig. 7). Additional removal of the iliac lymph nodes prevents lymphatic bypass through the deep medical system and connections between the deep and superficial lymphatic systems into the inguinal region. The therapeutic effect of CeLyTs in chronic lymphedema model mice took longer to manifest compared with that in LD mice. This suggests that CeLyTs transplantation in LD mice led to an earlier therapeutic effect because of its connection with the iliac lymph nodes. This study has revealed connections between the iliac lymph nodes and lymph node-like structures (Supplementary Fig. 10b, c). However, bypass formation may be more challenging in chronic lymphedema model mice.

(Discussion, Line 450-)

This study had some limitations. First, obtaining primary LECs from mice tissues is challenging. Because primary LECs can only be isolated and harvested in limited quantities from mouse tissues, human-derived primary LECs were purchased and used instead. Second, immunodeficient mice were used in animal experiments, allowing CeLyTs composed of hMSCs and hLECs to survive and form lymph node-like structures in the mice. Finally, the validity of the disease model should be considered. In this study, lymphedema was induced by removing the lymph nodes. However, in limb lymphedema models, bypass pathways for lymphatic fluid can form easily, leading to a gradual reduction in edema over time [49]. Consistent with this, our results also showed spontaneous edema recovery about 50 days after lymph node removal (Supplementary Fig. 11c). Because of these experimental limitations, it is unclear whether this treatment method will be sufficient to treat lymphedema. Therefore, further investigations are needed to develop more effective therapeutic strategies, such as combining CeLyTs transplantation with additional interventions, including pharmacological treatments, lymphovenous anastomosis bypass surgery, or the transplantation of VEGF-C overexpressing MSCs.

(Supplementary Methods, Line 91-)

Therapeutic effect of CeLyTs on chronic lymphedema model mice. hLEC/hMSC spheroids were prepared by hLECs (2×10^5 cells) and hMSCs (1.5×10^6 cells) were seeded in 96 well ultra-low attachment plate (ThermoFisher Scientific) and incubated for 5 days. Chronic lymphedema model mice were induced by removing the iliac, popliteal, and inguinal lymph nodes in the right lower limbs of mice as previously described [49]. Bioengineered tissues were transplanted to the site of the removed popliteal lymph node. For quantitative evaluation, the thickness of the paws and legs of the right lower limbs of LD mice with or without transplantation of bioengineered tissues were measured for 42 days and the change rate to day 0 was calculated.

(Supplementary Figures)

Supplementary Figure 7

(Supplementary Figure legends, Line 241-)

Supplementary Fig. 7. | Therapeutic effect of bioengineered tissues with lymphatic network in chronic lymphedema model mice.

a, b, Size change of paws (a) and legs (b) in chronic lymphedema model mice after transplantation of hLEC/mMSC spheroid, hLEC+NIH3T3 tissue, and hLEC+hMSC tissue. The thickness of paws and legs was measured for 42 days and the change rate for day 0 was calculated. Data represent the mean \pm standard deviation, and P-values were determined by Dunnett's test. *P < 0.05 was considered statistically significant for the No treatment group. ns, not significant.

General Concerns:

3. Unclear role of LECs:

Interpreting the results may present a challenge due to the weak-magnification images, which do not clearly show the position of the transplanted CeLyTs in relation to the overall tissue. Additionally, given that Prox-1 is a transcription factor, it might be expected to overlap with DAPI; however, the presence of lymphatic endothelial cells remains uncertain due to the staining of the cytoplasm.

Specific Comments:

3. Validating LEC roles:

Conduct experiments comparing CeLyTs with and without LECs to directly assess LEC contributions. Track fluorescently labeled LECs to visualize their integration and functional involvement in lymphatic restoration.

Response:

We appreciate the reviewer's insightful comment. In response, we conducted additional experiments and included a high-magnification microscope image of hLEC+hMSC tissue in LD mice in Figure 7b, as well as an overview image of hLEC+hMSC tissue in LD mice immunostained with human- and mouse-reactive Prox-1 in Figure 8a. LECs derived from the transplanted CeLyTs were distributed throughout the lymph node-like structure and were also sparsely interspersed with mLECs derived from the host animal. Furthermore, high-magnification images of lymph node-like structures confirmed the co-localization of transplant-derived LECs with host-derived LECs. We also attempted to improve Prox-1 staining; however, despite multiple experiments and observations, the Prox-1 antibody used in this study predominantly stained the cytoplasm. Therefore, additional images in hMSC tissues in LD group as a control were taken, showing the absence of Prox-1, demonstrating the specificity of Prox-1 antibody for LECs in Figure 7b. Regarding the contribution of LECs to the therapeutic effect of CeLyTs, transplantation of mMSC tissues, which is CeLyTs without LECs had a low therapeutic effect for LD mice in Supplementary Figure 5. Furthermore, LECs were absent in the graft of hMSC tissue in the LD group in Figure 7b, so it might be that LECs are necessary for the therapeutic effect by transplantation of the CeLyTs.

Revisions corresponding to these findings have been appropriately incorporated into the Results, Figures, and Figure legends sections with changes marked in red in the revised manuscript.

(Results, Line 277-)

In addition, lymphatic vessels derived from the host animal and endogenous LECs from the donor co-existed in a mixed state within the tissue (Fig. 7b).

(Figures)

Figure 7b

Figure 8a

(Figure legends, Line 982-)

Fig. 7. Formation of a lymphatic node-like structure after the transplantation of CeLyTs.

b, Representative immunofluorescence staining for a lymphatic endothelial cell marker Prox-1 in a lymph node-like structure formed in hLEC+hMSC tissue-transplanted LD mice (day 21). Blue, DAPI; Green, mouse-reactive Prox-1; Red, human-reactive Prox-1. Scale bars, 50 μ m.

(Figure legends, Line 996-)

Fig. 8. Distribution of cells in lymphatic node-like structures.

a, Representative immunofluorescence staining for human- or mouse-reactive Prox-1 in a lymph node-like structure formed in hLEC+hMSC tissue-transplanted LD mice (day 21). Blue, DAPI; Green, mouse-reactive Prox-1; Red, human-reactive Prox-1. Scale bars, 1 mm (upper panel) and 200 μ m (lower panel).

*General Concerns:**4. Absence of functional validation:*

Data demonstrating lymph node-like structures' semi-circulatory functions, such as fluid filtration and foreign particle clearance, are not available at this time.

*Specific Comments:**4. Semi-circulatory function validation:*

Use FITC-labeled beads to track lymphatic fluid inflow and outflow. Quantify filtration capacity by measuring foreign particle retention and clearance rates.

Response:

We appreciate the reviewer's significant comment. In response, we conducted additional experiments and included data demonstrating the semi-circulatory function of lymph node-like structures in Supplementary Figure 10. We confirmed that these structures were connected to the iliac lymph nodes and perfused with lymphatic fluid. To further assess their functionality, we used fluorescence-labeled beads to track lymphatic inflow and outflow and quantified their foreign particle retention capacity. The lymphatic dynamics in these structures closely resembled those of natural lymph nodes, with comparable foreign particle retention and filtration capabilities. These findings indicate that the lymph node-like structures exhibit semi-circulatory function.

Revisions corresponding to these findings have been appropriately incorporated into the Results, Discussion, Supplementary Methods, and Supplementary Figure legends sections with changes marked in red in the revised manuscript.

(Results, Line 313-)

Furthermore, after the removal of distant lymph nodes and lymph node-like structures, ICG fluorescence signals in the lymph node-like structures increased at 1 h, while at 3 h after ICG administration, the signals were decreased in the lymph node-like structures and increased in the iliac lymph nodes (Supplementary Fig. 10b,c). The use of fluorescence-labeled beads to assess the filtering ability of the lymph node-like structures demonstrated that fluorescence signals in the lymph node-like structures were diminished, whereas a slight increase in signals was observed in the iliac lymph nodes 3 h after fluorescence-labeled bead administration (Supplementary Fig. 10d). These findings confirm that the lymphatic flow behavior and filtration capacity of the lymph node-like structures were comparable to those of popliteal lymph nodes in mice.

(Discussion, Line 398-)

Lymph nodes can be characterized by their active immune response mediated by abundant immune cells and lymphatic inflow and outflow [45-48]. The transplantation of CeLyTs into LD mice formed a lymph node-like structure containing a wide range of immune cells, which

promoted an immune response to CpG1018, and the inflow and outflow of ICG (Fig. 7c,d, Supplementary Fig. 9 and Supplementary Fig. 10a). Furthermore, the lymph node-like structures were connected to distant lymph nodes and integrated into a lymphatic network with filtration capacity (Supplementary Fig. 10b-d). These results indicate that the lymph node-like structure functions as a lymph node.

(Supplementary Methods, Line 111-)

Lymphatic flow in the reconstructed lymph node-like structure. CeLyTs were transplanted into LD mice. Then, fluorescence at the site of the removed popliteal lymph nodes (the transplantation site) was detected every 10 min after ICG administration on day 21 by an IVIS Lumina III (PerkinElmer, Waltham, MA, USA) under continuous isoflurane anesthesia. In addition, the lymph nodes or lymph node-like structures formed by transplantation of the CeLyTs were excised from normal mice or LD mice on day 21. Then, fluorescence at the site of the lymph nodes or lymph node-like structures was detected immediately, 1 h, and 3 h after ICG administration using iBright FL1000 Imaging System (Thermo Fisher Scientific), and the fluorescence intensity of the homogenates was measured by a GloMax[®] Discover Microplate Reader (Promega, Madison, WI, USA). To evaluate the semi-circulatory function of lymph node-like structures, single-cell suspension was prepared from excised lymph nodes and lymph node-like structures formed by transplantation of the CeLyTs immediately, 1 h, and 3 h after fluorescence-labeled beads (FluoSpheres, 0.1 μm , yellow-green, Thermo Fisher Scientific) administration into mice footpad. The total uptake of fluorescence-labeled beads in a monocyte population was calculated using a positive rate of fluorescein isothiocyanate (FITC). Data were acquired using a BD FACSLyric (BD Biosciences Pharmingen) and analyzed using FlowJo software version 10.7.2 (BD Biosciences Pharmingen).

(Supplementary Figures)
Supplementary Figure 10

(Supplementary Figure legends, Line 292-)

Supplementary Fig. 10. | Lymphatic flow in lymph node-like structure after transplantation of bioengineered tissues with lymphatic network

a, *In vivo* imaging of lymph flow in lymph node-like structures formed in hLEC+hMSC tissue-transplanted LD mice (day 21) after ICG injection. **b**, *Ex vivo* imaging of lymph nodes or lymph node-like structures immediately (im), 1 h, and 3 h after ICG administration to normal mice (+ICG), LD mice (LD+ICG), and hLEC+hMSC tissue-transplanted LD mice (day 21). **c**, Fluorescent intensity of excised and homogenized lymph nodes or lymph node-like structures immediately (im), 1 h, or 3 h after ICG administration. Data represent the mean \pm standard deviation (n=3). **d**, Flow cytometric analysis of filtration capacity monocyte population in lymph nodes or lymph node-like structures immediately (im), 1 h, and 3 h after fluorescence-labeled beads administration to normal mice (No treatment) and hLEC+hMSC tissue-transplanted LD mice (day 21). Data represent the mean \pm standard deviation (n=3). AX, Axillary lymph node; PO, Popliteal lymph node; IL, Iliac lymph node.

5. *Improving data presentation:*

Include wide-field images to illustrate CeLyTs positioning within host tissue. Clarify Prox-1 staining results to confirm lymphatic endothelial cell presence.

Response:

We appreciate the reviewer's helpful comment. In response, we conducted additional experiments and included a wide-field image of hLEC+hMSC tissue in LD immunostained for Prox-1 in Figure 8a. Prox-1-positive LECs were observed not only within the lymph node-like structure but also in the surrounding tissues.

Corresponding revisions have been appropriately incorporated into the Figure legends section, with changes marked in red in the revised manuscript.

(Figures)

Figure 8a

(Figure legends, Line 996-)

Fig. 8. Distribution of cells in lymphatic node-like structures.

a, Representative immunofluorescence staining for human- or mouse-reactive Prox-1 in a lymph node-like structure formed in hLEC+hMSC tissue-transplanted LD mice (day 21). Blue, DAPI; Green, mouse-reactive Prox-1; Red, human-reactive Prox-1. Scale bars, 1 mm (upper panel) and 200 μ m (lower panel).

Reviewer #2:

This manuscript, "Reconstruction of the lymphatic system by transplantation of a centrifuge-based bioengineered lymphatic tissue" by Obana et al. describes a method for the generation of artificial lymphatic (lymph-node-like) tissue to treat secondary lymphedema after lymphadenectomy.

The authors describe a centrifugation-based technique, involving human lymphatic endothelial cells and murine mesenchymal stromal/stem cells. After characterization of the artificial tissue, they show the functionality in several in vivo experimental system (LD mice). The authors conclude that the system, which they call "CeLyT" holds great potential for the treatment of secondary lymphedema.

The reviewer finds the story interesting, and believes that this study is significant for the field of lymphedema research. A solid body of qualitative and quantitative data allows for most of the conclusions drawn.

Some issues, however, should be addressed:

1) Why did the authors chose human lymphatic endothelial cells in combination with murine MSCs? Why not having both cell types from the same species? I think this deserves some sentences in the discussion section.

Response:

We apologize for the lack of information. At the start of this study, obtaining primary cultured mouse LECs was challenging. The number of LECs harvested from mouse tissues was very limited, and their expansion in culture was also difficult. Therefore, we purchased and used human LECs from Takara Bio Inc. (Tokyo, Japan). As a result, human LECs, mouse MSCs, and human MSCs were used in this study, as described in the limitations section of the Discussion.

Corresponding revisions have been appropriately incorporated into the Discussion section, with changes marked in red in the revised manuscript.

(Discussion, Line 450-)

This study had some limitations. First, obtaining primary LECs from mice tissues is challenging. Because primary LECs can only be isolated and harvested in limited quantities from mouse tissues, human-derived primary LECs were purchased and used instead. Second, immunodeficient mice were used in animal experiments, allowing CeLyTs composed of hMSCs

and hLECs to survive and form lymph node-like structures in the mice. Finally, the validity of the disease model should be considered. In this study, lymphedema was induced by removing the lymph nodes. However, in limb lymphedema models, bypass pathways for lymphatic fluid can form easily, leading to a gradual reduction in edema over time [49]. Consistent with this, our results also showed spontaneous edema recovery about 50 days after lymph node removal (Supplementary Fig. 11c). Because of these experimental limitations, it is unclear whether this treatment method will be sufficient to treat lymphedema. Therefore, further investigations are needed to develop more effective therapeutic strategies, such as combining CeLyTs transplantation with additional interventions, including pharmacological treatments, lymphovenous anastomosis bypass surgery, or the transplantation of VEGF-C overexpressing MSCs.

2) methodology: Were all experiments performed with one and the same LEC donor? Or did the authors use biological replicates (same for the MSCs?)

Response:

We apologize for the lack of information. The primary cultured human LECs were isolated from a single Caucasian male donor (2 years old). Commodity and lot number information have also been added to the manuscript. Similarly, the primary cultured human MSCs were isolated from a single Japanese donor.

Corresponding revisions have been appropriately incorporated into the Online Methods section, with changes marked in red in the revised manuscript.

(Online Methods, Line 481-)

Materials and animals

The mMSC line C3H10T1/2 was kindly provided by Dr. Hiroki Kagawa (Kyoto, Japan). The mouse fibroblast cell line NIH3T3 (fibroblast) was obtained from RIKEN Bioresource Center (RIKEN BRC) (Ibaraki, Japan). Human primary adipose-derived stem cells (hMSC) were kindly provided by Rohto Pharmaceutical Co., Ltd. (Osaka, Japan). *The hMSCs were isolated from adipose tissue from a single donor.* Human primary lymphatic endothelial cells (hLEC) were purchased from Takara Bio Inc. (Tokyo, Japan, C-12216, lot: 467Z001.2). *The hLECs were isolated from juvenile foreskin (different locations) from a single donor.* All other chemicals used were of the highest commercially available grade. Male nude BALB/c Slc-nu/nu mice (5–8 weeks old) were purchased from Sankyo Labo Service Co., Inc. (Tokyo, Japan) and maintained under specific pathogen-free conditions. All animals were treated under anesthesia using isoflurane, and euthanized by excess anesthesia after experiments. The protocols for animal experimentations

were conducted in accordance with the Institutional Animal Experimentation Committee of the Tokyo University of Science (latest approval number: Y23004, latest approval date: August 8, 2024). All animal experimentations were conducted in accordance with the principles and procedures outlined in the National Institutes of Health Guide for the Care and Use of Laboratory Animals and the ARRIVE guidelines. **The sex of the cells and animals was not considered in the study design of any experiment because it was thought to not affect the results.**

3) language: please check the whole manuscript in terms of "over-exaggerations", like "great potential" (p2, line 41), or "strongly detected" (p4, line 112), or "dramatically increased" (p5, line 144). Such words come rather unscientific.

Response:

We appreciate the reviewer's helpful comment. We have carefully reviewed and revised the manuscript to eliminate unscientific wording.

Corresponding revisions have been appropriately incorporated into the Abstract and Results sections, with changes marked in red in the revised manuscript.

(Abstract, Line 40-)

Taken together, CeLyTs composed of lymphatic endothelial cells and MSCs reconstructed the lymph node and might be a **promising therapy for secondary lymphedema.**

(Results, Line 114-)

Significantly, the expressions of Prox-1 **and VEGFR-3** in the bioengineered tissues prepared by the centrifugal cell stacking technique were **higher than those in other bioengineered tissues** (Fig. 2d **and Supplementary Fig. 1f**).

(Results, Line 148-)

The expressions of PDPN and VEGFR-3 genes, important factors of lymphangiogenesis in hLEC+mMSC tissues, were significantly increased compared with those in an hLEC/mMSC mixed tissue prepared by single seeding without centrifugation, and were also higher than those in monolayered hLEC/mMSC (Supplementary Fig. 2a).

4) p4, line 109: "... a lymphatic endothelial cell marker Prox-1 was detected in these tissues...." - I would think, since LEC are in these tissues, that Prox-1 should be detected, since Prox-1 is a reliable marker for lymphatics.

Response:

We apologize for any confusion regarding the provided details. Prox-1 expression was detected in bioengineered tissues containing hLECs (Fig. 2d). However, its expression in the single-seeding condition (hLEC/mMSC mixed tissue) was lower than that in the centrifugal cell stacking condition (hLEC+mMSC tissue) due to the lower survival rate of hLECs in the hLEC/mMSC mixed tissue (Supplementary Fig. 2c). Additionally, the expression levels of other lymphatic endothelial cell markers, such as Podoplanin (PDPN, also known as gp38 and T1 α) and vascular endothelial growth factor receptor (VEGFR)-3, were also low (Supplementary Fig. 2a). Furthermore, in the single-seeding condition (mMSC tissue), no LECs were present in the bioengineered tissues, and Prox-1 expression was not detected.

Corresponding revisions have been appropriately incorporated into the Results section, with changes marked in red in the revised manuscript.

(Results, Line 154-)

Flow cytometric analysis showed that the geometric mean fluorescence intensity of the hLEC/CFSE+hMSC tissue was significantly higher than that of the hLEC/CFSE/hMSC mixed tissue prepared by single seeding without centrifugation (Supplementary Fig. 2c), indicating the survival of internal LECs was maintained in CeLyTs. **This result corresponded with the expression level of Prox-1 protein in the bioengineered tissues (Fig. 2d).**

5) p5, line 146: "in addition, the gene expression in hLEC+hMSCs increased...." - which gene expression? please specify.

Response:

We apologize for the lack of information. Podoplanin (PDPN) and vascular endothelial growth factor receptor (VEGFR)-3 are key factors in lymphangiogenesis. Their expression levels were increased in hLEC+mMSC tissue. Information regarding PDPN and VEGFR-3 has been added to the manuscript.

Corresponding revisions have been appropriately incorporated into the Results section, with changes marked in red in the revised manuscript.

(Results, Line 148-)

The expressions of PDPN and VEGFR-3 genes, important factors of lymphangiogenesis in

hLEC+mMSC tissues, were **significantly** increased compared with those in an hLEC/mMSC mixed tissue prepared by single seeding without centrifugation, and were also higher than those in monolayered hLEC/mMSC (Supplementary Fig. 2a).

6) p7, line 211: "LD mice were prepared...."? please reword. same as in line 216: "little therapeutic effect", similar in line 220 (".... a superior therapeutic efficacy....") and also at other locations throughout the manuscript.

Response:

We appreciate the reviewer's helpful comment. We have carefully reviewed the manuscript and revised the terminology accordingly.

Corresponding revisions have been appropriately incorporated into the Results, Discussion, Online Methods, and Supplementary Methods sections, with changes marked in red in the revised manuscript.

(Results, Line 191-)

First, we **used** LD mice with impaired lymphatic flow by removing the popliteal and inguinal lymph nodes.

(Results, Line 222-)

LD mice were **used** and various suspended cells and bioengineered tissues were used for transplantation (Supplementary Fig. 4a [top]).

(Results, Line 226-)

The suspended hLECs transplantation group and suspended mMSCs transplantation group had a **low** therapeutic effect, which was equivalent to that of the LD mouse group.

(Results, Line 231-)

The results demonstrated that CeLyTs exhibited a **high** therapeutic effect on the treatment of LD mice compared with MSC transplantation, which is anticipated to be a potential therapeutic strategy for lymphedema.

(Discussion, Line 363-)

Furthermore, we speculated that the survival of the internal cells of CeLyTs was more important for the treatment of lymphedema, because the mMSC tissues or hLEC/mMSC mixed tissues prepared by single seeding without centrifugation had a **low** therapeutic effect on lymphedema

(Supplementary Fig. 5c-e).

(Discussion, Line 473-)

Conversely, CeLyTs resulted in the reconstruction of a lymph node-like structure and long-term suppression of lymphedema, exhibiting a high therapeutic effect compared with conventional treatments over 100 days (Supplementary Fig. 11).

(Online Methods, Line 615-)

LD mice were induced by removing the popliteal and inguinal lymph nodes in the right lower limbs of mice as previously described [49].

(Supplementary Methods, Line 69-)

LD mice were induced by removing the popliteal and inguinal lymph nodes in the right lower limbs of mice.

7) p9, line 280: *"This study successfully fabricated...." "... not the study, but you authors fabricated*

Response:

We appreciate the reviewer's helpful comment. The pointed-out sentence has been revised.

Corresponding revisions have been appropriately incorporated into the Discussion section, with changes marked in red in the revised manuscript.

(Discussion, Line 332-)

In this study, we successfully fabricated CeLyTs composed of LECs and MSCs and demonstrated their usefulness for the treatment of secondary lymphedema, for which no effective radical treatment is available.

8) *finally, although the reviewer appreciates the work of many outstanding scientists from Japan, the authors are strongly advised to take care of a more balanced reference list, including also more groundbreaking work from the lab of Kari Alitalo (University Helsinki, Finland) and many others in his scientific surroundings, or the work from Wolfgang Holthoner (LBI Trauma, Austria, e.g. PMID: 28459049) or Ernst Reichmann (e.g. PMID: 24363089)*

Response:

We appreciate the reviewer's helpful comment. We have reviewed and revised the balance of

references. The number of citations from Japanese scientists has been reduced, and new references, including those from the laboratories of Wolfgang Holnthoner and Ernst Reichmann, have been added.

New reference list has been appropriately incorporated into the revised manuscript, with changes marked in red.

New references

18. Klar, A. S., Böttcher-Haberzeth, S., Biedermann, T., Schiestl, C., Reichmann, S., Meuli, M. Analysis of blood and lymph vascularization patterns in tissue-engineered human dermo-epidermal skin analogs of different pigmentation. *Pediatr Surg Int.* 30, 223–31 (2014). doi: 10.1007/s00383-013-3451-0.
19. Knezevic, L., Schaupper, M., Mühleder, S., Schimek, K., Hasenberg, T., Marx, U., Priglinger, E., Redl, H., Holnthoner, W. Engineering blood and lymphatic microvascular networks in fibrin matrices. *Front Bioeng Biotechnol.* 5, 25 (2017). doi: 10.3389/fbioe.2017.00025.
23. Song, N., Scholtemeijer, M., Shah, K. Mesenchymal stem cell immunomodulation: Mechanisms and therapeutic potential. *Trends Pharmacol Sci.* 41, 653–664 (2020). doi: 10.1016/j.tips.2020.06.009.
40. Feyen, D. A. M., Gaetani, R., Doevendans, P. A., Sluijter, J. P. G., Stem cell-based therapy: Improving myocardial cell delivery. *Adv Drug Deliv Rev.* 106, 104-115 (2016). doi: 10.1016/j.addr.2016.04.023.

Reviewer #3: The study is presenting an original and useful technology showing that three-layered bioengineered tissue composed of LECs and MSCs was successfully fabricated by the centrifugal cell stacking technique. In this study, the authors suggest that cell transplantation can regenerate functional lymph nodes and suppress lymphedema over an extended period. CeLyTs composed of LECs and MSCs might reconstruct lymph nodes in LD patients and thus be a promising therapy for secondary lymphedema.

This study is very important and could lead to significant therapeutic breakthroughs. However, the experimental data still need to be greatly expanded to better understand how the system is established.

It should be specified in the introduction that the leading cause of secondary lymphedema is parasitic infection due to filariasis. Lymphadenectomy is the primary cause in industrialized countries.

The authors should mention the work from Saaristo with the Lymfactin assay that was promising but did not provided significant improvement of lymphedema with VEGFC delivery (doi: 10.1097/PRS.00000000000011675.).

Response:

We appreciate the reviewer's helpful comment. We have added information regarding the role of parasitic infection due to filariasis in causing secondary lymphedema. Additionally, we explain that the delivery of vascular endothelial growth factor (VEGF)-C, a promising approach for treating lymphedema, has not proven to be effective, citing new references.

Corresponding revisions have been appropriately incorporated into the Introduction section, with changes marked in red and yellow in the revised manuscript.

(Introduction, Line 45-)

Lymphatic vessels, which are mainly composed of lymphatic endothelial cells (LECs), connect hundreds of lymphoid organs, transport tissue metabolites, and peripheral antigens [1,2]. Lymph nodes are the organs responsible for immune surveillance and immunomodulatory functions and are connected by numerous lymphatic vessels [3]. Dysfunction of lymphatic vessels caused by congenital mutations or cancer treatment results in primary and secondary lymphedema, respectively [4-6]. Secondary lymphedema is often caused by lymphadenectomy (LD) during the surgical treatment of cancer and parasitic infection related to filariasis [7,8]. Because lymph nodes do not regenerate once removed, a patient's quality of life is severely impaired by recurrent complications, including interstitial fluid retention and limb cellulitis [9]. Compression therapy, which is the gold standard treatment [9], liposuction [10], manual lymphatic drainage [11], and

lymphatic anastomosis are treatment options for secondary lymphedema, but their edema-suppressing effect is limited [12,13]. Additionally, autologous lymph node transplantation for lymph node reconstruction and **vascular endothelial growth factor (VEGF)-C gene therapy** has been attempted, but there is insufficient evidence of a therapeutic effect [14,15].

New references

15. Rannikko, E. H., Pajula, S., Suominen, S. H., Kiiski, J., Mani, M. R., Halle, M., Kaartinen, I. S., Lahdenperä, O., Arnardottir, T. H., Kauhanen, S. M., Kavola, H., Majava, M., Niemi, T. S., Brück, N. M., Mäki, M. T., Seppänen, M. P., Saarikko, A. M., Hartiala, P. Phase II study shows the effect of adenoviral vascular endothelial growth factor C and lymph node transfer in lymphedema. *Plast Reconstr Surg.* 155, 256e-267e. doi: 10.1097/PRS.00000000000011675.

In Fig 2d, Prox1 staining is cytoplasmic, whereas Prox1 is a transcription factor that exhibits a nuclear localization. The authors should reproduce the staining and add another lymphatic marker such as VEGFR3 to validate sup. Fig2 data.

Higher magnification showing the cell shape would be appreciated.

Response:

We appreciate the reviewer's meaningful comment. Prox-1 staining was cytoplasmic in the repeated experiment when using the Prox-1 antibody (Proteintech, Rosemont, IL, USA, 11067-2-AP) employed in this study. We have performed additional experiments and added an image stained with another LEC marker, VEGFR-3, in Supplementary Figure 1f. Higher magnification images have also been included.

Corresponding revisions have been appropriately incorporated into the Results, Online Methods, and Figure legends sections, with changes marked in red in the revised manuscript.

(Results, Line 110-)

To examine differences between hLEC+MSC tissues prepared by the centrifugal cell stacking technique and mMSC tissues or hLEC/mMSC mixed tissues prepared by single seeding without centrifugation, the lymphatic endothelial cell markers Prox-1 and **VEGF receptor (VEGFR)-3** were detected in these tissues by immunofluorescence staining. Significantly, the expressions of Prox-1 and **VEGFR-3** in the bioengineered tissues prepared by the centrifugal cell stacking technique were **higher than those in other bioengineered tissues** (Fig. 2d and Supplementary Fig. 1f).

(Online Methods, Line 585-)

The primary and secondary antibodies were as follows: 1:200 PROX-1 (Proteintech, Rosemont, IL, USA, 11067-2-AP), 1:500 mouse PROX-1 (Abcam, Cambridge, UK, ab101851), 1:500 human PROX-1 (Bio-Techne, Minneapolis, MN, USA, AF2727), 1:500 CD31 (Bio-Techne, NB600-1475), 1:500 mouse CD31 (Abcam, ab256569), 1:500 human CD31 (Abcam, ab76533), 1:100 B220/CD45R (Bio-Techne, NBP2-53303), 1:100 CD3 (Arigo, Zhubei, Taiwan, ARG22819), 1:50 CD11b (Abcam ab8878), 1:50 CD11c (Thermo Fisher Scientific, PA5-90208), 1:500 Collagen Type I (Proteintech, 14695-1-AP), 1:500 VEGFR-3 (Thermo Fisher Scientific, BS-1083R), 1:50 HuNuC (Sigma-Aldrich, MAB1281C3), 1:400 CD45 (Thermo Fisher Scientific, 56-0451-82), 1:100 CD3 (Bio-Techne, FAB4841G), 1:100 CD19 (Bio-Techne, NBP2-24965AF647), 1:100 F4/80 (Cell Signaling Technology, Inc., Danvers, MA, USA, 52267), 1:200 CD11c (Bio-Techne, FAB69501R), 1:500 anti-rabbit Alexa488 (Thermo Fisher Scientific, A21206), 1:500 anti-rat Alexa488 (Thermo Fisher Scientific, A-11006), 1:500 anti-rabbit Alexa647 (Thermo Fisher Scientific, A31573), and 1:500 anti-goat Alexa647 (Thermo Fisher Scientific, A21447).

(Supplementary Figures)

Supplementary Figure 1f

(Supplementary Figure legends, Line 142-)

Supplementary Fig. 1. | Optimization of centrifugation conditions for the preparation of CeLyTs.

f, Immunofluorescence images of a lymphatic endothelial cell marker VEGFR-3 in mMSC tissues (by single seeding without centrifugation), hLEC/mMSC mixed tissue (by single seeding without centrifugation), and hLEC+mMSC tissue (by centrifugal cell stacking technique). Blue, DAPI; Red, VEGFR-3. Scale bars, 100 μ m (left panel) and 50 μ m (right panel).

Figure 3c: statistic are needed to validate the data.

Response:

We appreciate the reviewer's helpful comment. We performed Dunnett's test on the data for each day using Day 1 in Figure 3c. The figure was revised and replaced with a graph showing the static test results.

Corresponding revisions have been appropriately incorporated into the Results and Figure legends sections, with changes marked in red in the revised manuscript.

(Results, Line 138-)

The relative light units derived from live cell-releasing Nluc in the culture supernatant **remained at similar levels** throughout the culture period, **with no statistically significant differences** (Fig. 3c).

(Figures)

Figure 3c

(Figure legends, Line 902-)

Fig. 3. Characteristics of CeLyTs.

c, Relative light unit in the culture supernatant of an mMSC/Nluc+mMSC tissue as an indicator of the cell viability of mMSC/Nluc inside the tissue. Data represent the mean \pm standard deviation (n=3), and P-values were determined by Dunnett's test. *P < 0.05 was considered statistically significant for Day 1. ns, not significant.

Figure 3d and e: A quantification of the cell viability should be provided, after how many days of culture the pictures were taken? This has to be mentioned in the manuscript (same comment for suppl. Fig. 1).

Response:

We apologize for any confusion in the provided details and the lack of information. We have included data on quantifying cell viability in bioengineered tissue in Supplementary Fig. 2c. The survival of internal LECs in hLEC/CFSE+hMSC tissue was higher than that in the hLEC/CFSE/hMSC mixed tissue, as determined by flow cytometry analysis. Additionally, we have added information about the culture days of the bioengineered tissues for the images presented, as well as to the parts you pointed out and other relevant sections.

Corresponding revisions have been appropriately incorporated into the Figure legends and Supplementary Figure legends sections, with changes marked in red in the revised manuscript.

(Figure legends, Line 885-)

a, Fluorescence microscopic observation of suspended and centrifuged mMSC/GFP at different times after seeding (**day 1**). Scale bars, 500 μm . **b**, Typical images of hLEC+hMSC tissue, hLEC+mMSC tissue, and NIH3T3 (fibroblasts) fabricated by a centrifugal cell stacking technique (**day 5**). Scale bars, 5 mm. **c**, Paraffin-section images of an hLEC+hMSC tissue (**day 5**). Scale bars, 50 μm .

(Figure legends, Line 906-)

d, Luminescence images of an mMSC/fluc+mMSC tissue to evaluate the viability of mMSC/Nluc inside the tissue (**day 5**). Scale bar, 5 mm. **e**, Fluorescence microscopic images of an mMSC+mMSC/CFSE tissue to evaluate the viability of mMSC/CFSE outside the tissue (**day 5**). Green, CFSE. Scale bar, 5 mm.

(Supplementary Figure legends, Line 137-)

a, **b**, Fluorescence microscopic observation of mMSCs/GFP after centrifugation under various conditions. (A) Seeding once and (b) twice (**day 1**). Scale bars, 500 μm . **c**, Paraffin-sections and microscopic images of fibroblast cells and mMSC tissue (**day 5**). Scale bars, 50 μm .

(Supplementary Figure legends, Line 160-)

c, Flow cytometric analysis of the viability of hLECs in hLEC/CFSE/mMSC mixed tissues and hLEC/CFSE+hMSC tissues (**day 5**). Monolayered mMSCs were used as a negative control. Data represent the mean \pm standard deviation (n=3), and P-values were determined by the Tukey-Kramer test. *P < 0.05 was considered statistically significant.

In figure 4, the authors transplanted the CeLyTs subcutaneously in mice. They claim that mMSC/Nluc cells and mMSC/Nluc+NIH3T3 tissues disappeared 3–4 days after transplantation, and those of mMSC/Nluc+hMSC and hLEC+mMSC/Nluc tissues persisted for at least 21 days. However, it seems that at least 5 times less mMSC/Nluc cells and mMSC/Nluc+NIH3T3 material was implanted into mice, which can explain that they disappeared faster. The author should start with equivalent signal in all groups.

Response:

We apologize for any confusion in the provided details. The number of luciferase-expressing cells (mMSC/Nluc) in the suspended mMSC/Nluc and hLEC+mMSC/Nluc tissues, mMSC/Nluc+NIH3T3 tissue, and mMSC/Nluc+hMSC tissue is the same. This is a relative evaluation for each group, and the color scale has been adjusted, which may make the results appear different. Additionally, because imaging was performed on Day 1 after transplantation, the appearance may vary depending on the early immune rejection response.

Corresponding revisions have been appropriately incorporated into the Results section, with changes marked in red in the revised manuscript.

(Results, Line 168-)

To confirm the usefulness of CeLyTs fabricated with MSCs for survival after transplantation, the survival periods of suspended mMSC/Nluc cells and various bioengineered tissues were evaluated by *in vivo* imaging. **The same number of Nluc-expressing cells were transplanted, and the survival rates after transplantation were evaluated relatively by adjusting the color scale of each group.**

Also, the imaging in figure 4d should be improved: lower magnification of the skin should be provided to show all the layers (epidermis/dermis). Importantly, there is a lot of lymphatic vessels into the normal skin. Why do they do not show any? It would be important (crucial) to see if the transplanted CeLyTs may connect to the preexisting network. How do they differentiated endogenous LEC from transplanted ones? Confocal microscopy is needed to answer this question.

Response:

We appreciate the reviewer's insightful comment. We have performed additional experiments and added overview section images of hLEC+hMSC tissue in skin tissue in Figure 4e. The transplanted tissue survived and was located between the epidermis and the peritoneum. However, despite several experiments and observations, we could not detect lymphatic vessels in the skin using the confocal laser microscope (Leica Microsystems, Wetzlar, Germany) employed in this

study. In fact, lymphatic vessels are likely present at a level that this study could not detect. Additionally, we have included high-magnification images of hLEC+hMSC tissue in LD in Figure 7b, and overview section images of hLEC+hMSC tissue in LD immunostained with human- and mouse-reactive Prox-1 in Figure 8a. We differentiated LECs using human- and mouse-reactive Prox-1 antibodies. The lymphatic vessels derived from the host animal and endogenous LECs from the donor were randomly distributed, with some co-localizing in the tissue.

Corresponding revisions have been appropriately incorporated into the Results, Figure legends, and Supplementary Figure legends sections, with changes marked in red in the revised manuscript.

(Results, Line 181-)

The expressions of Prox-1 and an endothelial marker CD31 were detected in skin tissues transplanted with hLEC+mMSC tissues 14 days after transplantation, although the expression of Prox-1 in normal skin was beyond the limit of detection (Fig. 4d). Furthermore, the transplanted hLECs survived and were located between the dermis and peritoneum (Fig. 4e).

(Results, Line 277-)

In addition, lymphatic vessels derived from the host animal and endogenous LECs from the donor co-existed in a mixed state within the tissue (Fig. 7b).

(Figures)

Figure 4e

Figure 7b

Figure 8a

(Figure legends, Line 930-)

Fig. 4. Survival of CeLyTs and formation of lymphatic structures after transplantation.

e, Representative immunofluorescence images of hLEC+hMSC tissues 14 days after subcutaneous transplantation into mice. Blue, DAPI; Red, HuNuC; Magenta, human-reactive Prox-1. The dermis is indicated above the dotted lines. Scale bars, 1 mm.

(Figure legends, Line 982-)

Fig. 7. Formation of a lymphatic node-like structure after the transplantation of CeLyTs.

b, Representative immunofluorescence staining for a lymphatic endothelial cell marker Prox-1 in a lymph node-like structure formed in hLEC+hMSC tissue-transplanted LD mice (day 21). Blue, DAPI; Green, mouse-reactive Prox-1; Red, human-reactive Prox-1. Scale bars, 50 μ m.

(Figure legends, Line 996-)

Fig. 8. Distribution of cells in lymphatic node-like structure.

a, Representative immunofluorescence staining for human- or mouse-reactive Prox-1 in a lymph

node-like structure formed in hLEC+hMSC tissue-transplanted LD mice (day 21). Blue, DAPI; Green, mouse-reactive Prox-1; Red, human-reactive Prox-1. Scale bars, 1 mm (upper panel) and 200 μm (lower panel).

Next, the authors evaluated the effect of CeLyTs on the lymphatic flow using ICG footpad injection.

The figure 5 needs control mice with no surgery to compare with basal diffusion of the dye.

Response:

We apologize for any confusion in the provided details. The No treatment group in Figure 5 is the no-surgery group. Additionally, in the No treatment group, ICG was administered into the footpad of healthy mice.

We can clearly observe a perfusion of the lymph node-like structure. However, it would be interesting to see if it connects to the lymphatic network and if distant lymph nodes are perfused. The imaging detection might be too limited to observe fluorescence in distant lymph nodes such as axillary or mesenteric nodes across the skin. The authors should dissect them and image them independently. Is there a lymphatic flow?

Response:

We appreciate the reviewer's meaningful comment. We have performed additional experiments and included data demonstrating the connections to the lymphatic network and distant lymph nodes in Supplementary Figure 10b, c. We confirmed the outflow and connection of lymph flow to the iliac lymph nodes after it flowed into the lymph node-like structures by tracking lymph flow following the administration of ICG to the mice footpad. The transplanted CeLyTs were connected to the lymphatic network, enabling lymphatic fluid perfusion to the iliac lymph nodes.

Corresponding revisions have been appropriately incorporated into the Results, Discussion, Supplementary Methods, and Supplementary Figure legends sections, with changes marked in red in the revised manuscript.

(Results, Line 313-)

Furthermore, after the removal of distant lymph nodes and lymph node-like structures, ICG fluorescence signals in the lymph node-like structures increased at 1 h, while at 3 h after ICG administration, the signals were decreased in the lymph node-like structures and increased in the iliac lymph nodes (Supplementary Fig. 10b,c). The use of fluorescence-labeled beads to assess the filtering ability of the lymph node-like structures demonstrated that fluorescence signals in

the lymph node-like structures were diminished, whereas a slight increase in signals was observed in the iliac lymph nodes 3 h after fluorescence-labeled bead administration (Supplementary Fig. 10d). These findings confirm that the lymphatic flow behavior and filtration capacity of the lymph node-like structures were comparable to those of popliteal lymph nodes in mice.

(Discussion, Line 398-)

Lymph nodes can be characterized by their active immune response mediated by abundant immune cells and lymphatic inflow and outflow [45-48]. The transplantation of CeLyTs into LD mice formed a lymph node-like structure containing a wide range of immune cells, which promoted an immune response to CpG1018, and the inflow and outflow of ICG (Fig. 7c,d, Supplementary Fig. 9 and Supplementary Fig. 10a). Furthermore, the lymph node-like structures were connected to distant lymph nodes and integrated into a lymphatic network with filtration capacity (Supplementary Fig. 10b-d). These results indicate that the lymph node-like structure functions as a lymph node.

(Supplementary Methods, Line 111-)

Lymphatic flow in the reconstructed lymph node-like structure. CeLyTs were transplanted into LD mice. Then, fluorescence at the site of the removed popliteal lymph nodes (the transplantation site) was detected every 10 min after ICG administration on day 21 by an IVIS Lumina III (PerkinElmer, Waltham, MA, USA) under continuous isoflurane anesthesia. In addition, the lymph nodes or lymph node-like structures formed by transplantation of the CeLyTs were excised from normal mice or LD mice on day 21. Then, fluorescence at the site of the lymph nodes or lymph node-like structures was detected immediately, 1 h, and 3 h after ICG administration using iBright FL1000 Imaging System (Thermo Fisher Scientific), and the fluorescence intensity of the homogenates was measured by a GloMax[®] Discover Microplate Reader (Promega, Madison, WI, USA). To evaluate the semi-circulatory function of lymph node-like structures, single-cell suspension was prepared from excised lymph nodes and lymph node-like structures formed by transplantation of the CeLyTs immediately, 1 h, and 3 h after Fluorescence-labeled beads (FluoSpheres, 0.1 μ m, yellow-green, Thermo Fisher Scientific) administration into mice footpad. The total uptake of Fluorescence-labeled beads in a monocyte population was calculated using a positive rate of Fluorescein Isothiocyanate (FITC). Data were acquired using a BD FACSLyric (BD Biosciences Pharmingen) and analyzed using FlowJo software version 10.7.2 (BD Biosciences Pharmingen).

(Supplementary Figures)

Supplementary Figure 10b, c

(Supplementary Figure legends, Line 295-)

Supplementary Fig. 10. | Lymphatic flow in lymph node-like structure after transplantation of bioengineered tissues with lymphatic network

b, *Ex vivo* imaging of lymph nodes or lymph node-like structures immediately (im), 1 h, and 3 h after ICG administration to normal mice (+ICG), LD mice (LD+ICG), and hLEC+hMSC tissue-transplanted LD mice (day 21). **c**, Fluorescent intensity of excised and homogenized lymph nodes or lymph node-like structures immediately (im), 1 h, and 3 h after ICG administration. Data represent the mean \pm standard deviation (n=3). **d**, Flow cytometric analysis of filtration capacity monocyte population in lymph nodes or lymph node-like structures immediately (im), 1 h, and 3 h after Fluorescence-labeled beads administration to normal mice (No treatment) and hLEC+hMSC tissue-transplanted LD mice (day 21). Data represent the mean \pm standard deviation (n=3). AX, Axillary lymph node; PO, Popliteal lymph node; IL, Iliac lymph node.

Also one of the major lymphedema hallmark is dermis thickening induced by chronic inflammation. Do the authors observe a decrease of dermis thickness in the presence of the lymph node-like structure?

Response:

We appreciate the reviewer's insightful comment. We have performed additional experiments and included data showing that the thickness of the skin is reduced by the transplantation of CeLyTs in LD mice in Supplementary Figure 8a, b. Transplantation of CeLyTs suppressed skin thickening, and MSC transplantation also contributed to this improvement. This suggests that the immunoregulatory ability of MSCs may also play a role in the suppression of inflammation associated with lymphedema.

Corresponding revisions have been appropriately incorporated into the Results, Discussion, and Supplementary Figure legends sections, with changes marked in red in the revised manuscript.

(Results, Line 256-)

Furthermore, we confirmed the effect of the CeLyTs on the hallmarks of lymphedema, including skin thickening and adipose tissue accumulation. The hLEC+mMSC tissue groups had a suppressive effect on the thickness of skin compared with the suspended mMSC and hLEC groups (Supplementary Fig. 8a,b).

(Discussion, Line 371-)

CeLyTs had a suppressive effect on the hallmarks of lymphedema induced by chronic inflammation (Supplementary Fig. 8). Suspended MSC transplantation also showed a tendency to improve skin thickening and adipose tissue accumulation (Supplementary Fig. 8a,b,d), indicating the immunoregulatory ability of MSCs may also contribute to improving inflammation associated with lymphedema by the transplantation of CeLyTs.

(Supplementary Figures)

Supplementary Figure 8a, b

(Supplementary Figure legends, Line 254-)

Supplementary Fig. 8. | Suppressive effect on lymphedema hallmarks after transplantation of bioengineered tissues with lymphatic network in LD.

a, Paraffin-section images of skin of Suspended cells- and bioengineered tissue-transplanted LD mice (day 21). Arrows, measurement sites for the thickness of skin. Scale bars, 500 μm. **b**, The thicknesses of skin were measured from five sites. Data represent the mean ± standard deviation (n=5), and P-values were determined by Dunnett's test. *P < 0.05 was considered statistically significant for No treatment. ns, not significant.

Do they observe changes in immune cell populations in the skin?

Response:

We appreciate the reviewer's meaningful comment. We have performed additional experiments and included data demonstrating the proportion of various immune cell populations in the dermis in Supplementary Figure 8c. Transplantation of CeLyTs improved the proportion of various immune cell populations in the dermis, bringing them to levels nearly equivalent to those in

normal mice. In addition, LD mice exhibited dermal inflammation and an increase in macrophages. The inflammation was suppressed by the transplantation of CeLyTs, which may be attributable to the immune regulatory properties of MSCs.

Corresponding revisions have been appropriately incorporated into the Results, Discussion, Online Methods, Supplementary Methods, and Supplementary Figure legends sections, with changes marked in red in the revised manuscript.

(Results, Line 259-)

In addition, immune cell populations were present in the dermis, which showed signs of inflammation [27], including decreased T cell numbers and increased macrophage numbers in the LD group. However, the hLEC+hMSC tissue-transplanted groups showed a recovery of immune cell populations, with T cell and macrophage numbers reaching levels nearly equivalent to those of normal mice (Supplementary Fig. 8c).

(Discussion, Line 371-)

CeLyTs had a suppressive effect on the hallmarks of lymphedema induced by chronic inflammation (Supplementary Fig. 8). Suspended MSC transplantation also showed a tendency to improve skin thickening and adipose tissue accumulation (Supplementary Fig. 8a,b,d), indicating the immunoregulatory ability of MSCs may also contribute to improving inflammation associated with lymphedema by the transplantation of CeLyTs.

(Online Methods, Line 585-)

The primary and secondary antibodies were as follows: 1:200 PROX-1 (Proteintech, Rosemont, IL, USA, 11067-2-AP), 1:500 mouse PROX-1 (Abcam, Cambridge, UK, ab101851), 1:500 human PROX-1 (Bio-Techne, Minneapolis, MN, USA, AF2727), 1:500 CD31 (Bio-Techne, NB600-1475), 1:500 mouse CD31 (Abcam, ab256569), 1:500 human CD31 (Abcam, ab76533), 1:100 B220/CD45R (Bio-Techne, NBP2-53303), 1:100 CD3 (Arigo, Zhubei, Taiwan, ARG22819), 1:50 CD11b (Abcam ab8878), 1:50 CD11c (Thermo Fisher Scientific, PA5-90208), 1:500 Collagen Type I (Proteintech, 14695-1-AP), 1:500 VEGFR-3 (Thermo Fisher Scientific, BS-1083R), 1:50 HuNuC (Sigma-Aldrich, MAB1281C3), 1: 400 CD45 (Thermo Fisher Scientific, 56-0451-82), 1:100 CD3 (Bio-Techne, FAB4841G), 1:100 CD19 (Bio-Techne, NBP2-24965AF647), 1:100 F4/80 (Cell Signaling Technology, Inc., Danvers, MA, USA, 52267), 1:200 CD11c (Bio-Techne, FAB69501R), 1:500 anti-rabbit Alexa488 (Thermo Fisher Scientific, A21206), 1:500 anti-rat Alexa488 (Thermo Fisher Scientific, A-11006), 1:500 anti-rabbit Alexa647 (Thermo Fisher Scientific, A31573), and 1:500 anti-goat Alexa647 (Thermo Fisher Scientific, A21447).

(Supplementary Methods, Line 101-)

Proportion of various immune cell populations in the dermis tissue. Preparation of single-cell suspension from excised dermis performed as previously reported [27]. Briefly, excised lymph nodes and lymph node-like structures were cut into small fragments and digested with DMEM/high glucose (not supplemented with antibiotics or FBS) containing 10 mg/mL dispase (Wako Pure Chemical Industries, Ltd., Osaka, Japan), 0.1 mg/mL collagenase P, and 0.1 mg/mL DNase I (both from Sigma-Aldrich). Cell suspensions were strained through a 70 μ m filter and stained with fluorescently labeled antibodies. Data were acquired using a BD FACSLyric (BD Biosciences Pharmingen) and analyzed using FlowJo software version 10.7.2 (BD Biosciences Pharmingen).

(Supplementary Figures)

Supplementary Figure 8c

(Supplementary Figure legends, Line 260)

Supplementary Fig. 8. | Suppressive effect on lymphedema hallmarks after transplantation of bioengineered tissues with lymphatic network in LD.

c, Proportion of various immune cell populations in the dermis of hLEC+hMSC tissue-transplanted LD mice (day 21). CD3⁺ cells, CD11c⁺ cells, and F4/80⁺ cells were gated by CD45⁺ cells. Data represent the mean \pm standard deviation (n=3).

What are the cell components of the lymph node-like structures after 7, 14, 21 days? A better histological description is recommended. Stainings are not convincing, the authors should provide better characterization by flow cytometry and histology of lymphocytes, macrophages and dendritic cells double stained with lymphatics and quantifications are needed.

Response:

We appreciate the reviewer's meaningful comment. We have performed additional experiments and included data demonstrating the proportion of various immune cell populations in the lymph node-like structures in Figure 8d. Immune cells were abundant in the lymph node-like structures, with T cells, B cells, and dendritic cells present at levels comparable to those in lymph nodes. Macrophages were also more abundant than in lymph nodes. Regarding changes in immune cells, T cells and macrophages infiltrated due to immune rejection of the transplant (Day 7) and subsequently increased again to form lymph node-like tissue (Day 21). B cells and dendritic cells also increased, and immune cells were abundant in the lymph node-like structures. Macrophages, in particular, were especially abundant in these structures, which may be due to the lymphangiogenic properties of CeLyTs. To further clarify the underlying mechanisms, it will be necessary to investigate the chemokines that attract immune cells in greater detail.

Corresponding revisions have been appropriately incorporated into the Results, Discussion, Online Methods, and Figure legends sections, with changes marked in red in the revised manuscript.

(Results, Line 297-)

In addition, the immune cell populations of the lymph node-like tissues were analyzed 7, 14, and 21 days after the transplantation of hLEC+hMSC tissues in LD mice. CD3 positive T cell numbers were increased at 7 days, then decreased at 14 days, but increased to the same level as in the lymph nodes 21 days after transplantation. Numbers of CD19 positive B cells and CD11c positive dendritic cells also rapidly increased to the same levels as in lymph nodes 21 days after transplantation. However, only the numbers of F4/80 positive macrophages had increased 7 days after transplantation, which might be an early immune response. These then increased further to become more abundant than in the lymph nodes 14 and 21 days after transplantation. Images showing abundant immune cells in the lymph node-like structures are shown in Fig. 8d.

(Discussion, Line 414-)

The changes in immune cell populations in the lymph node-like structures included a temporary increase in T cell and macrophage numbers that might be caused by immune rejection 7 days after the transplantation of CeLyTs, followed by a re-increase related to lymph node-like structure formation 21 days after transplantation. However, the numbers of B cells and dendritic cells

increased rapidly 21 days after transplantation (Fig. 8d). Lymph node-like tissues contained abundant immune cells 21 days after transplantation, but it will be necessary to study changes in immune cell populations after 21 days. Further investigation is required to clarify the mechanism of lymph node-like tissue formation and changes in immune cell populations.

(Online Methods, Line 585-)

The primary and secondary antibodies were as follows: 1:200 PROX-1 (Proteintech, Rosemont, IL, USA, 11067-2-AP), 1:500 mouse PROX-1 (Abcam, Cambridge, UK, ab101851), 1:500 human PROX-1 (Bio-Techne, Minneapolis, MN, USA, AF2727), 1:500 CD31 (Bio-Techne, NB600-1475), 1:500 mouse CD31 (Abcam, ab256569), 1:500 human CD31 (Abcam, ab76533), 1:100 B220/CD45R (Bio-Techne, NBP2-53303), 1:100 CD3 (Arigo, Zhubei, Taiwan, ARG22819), 1:50 CD11b (Abcam ab8878), 1:50 CD11c (Thermo Fisher Scientific, PA5-90208), 1:500 Collagen Type I (Proteintech, 14695-1-AP), 1:500 VEGFR-3 (Thermo Fisher Scientific, BS-1083R), 1:50 HuNuC (Sigma-Aldrich, MAB1281C3), 1: 400 CD45 (Thermo Fisher Scientific, 56-0451-82), 1:100 CD3 (Bio-Techne, FAB4841G), 1:100 CD19 (Bio-Techne, NBP2-24965AF647), 1:100 F4/80 (Cell Signaling Technology, Inc., Danvers, MA, USA, 52267), 1:200 CD11c (Bio-Techne, FAB69501R), 1:500 anti-rabbit Alexa488 (Thermo Fisher Scientific, A21206), 1:500 anti-rat Alexa488 (Thermo Fisher Scientific, A-11006), 1:500 anti-rabbit Alexa647 (Thermo Fisher Scientific, A31573), and 1:500 anti-goat Alexa647 (Thermo Fisher Scientific, A21447).

(Online Methods, Line 646)

Flow cytometry analysis of the reconstructed lymph node-like structure. Preparation of a single-cell suspension from excised lymph nodes and lymph node-like structures formed by transplantation of the CeLyTs was performed as previously reported [50]. Briefly, excised lymph nodes and lymph node-like structures were cut into small fragments and digested with RPMI-1640 medium containing 0.8 mg/mL dispase (Wako Pure Chemical Industries Ltd., Osaka, Japan), 0.2 mg/mL collagenase P, and 0.1 mg/mL DNase I (both from Sigma-Aldrich). Cell suspensions were strained through a 70- μ m filter and stained with fluorescently labeled antibodies. Data were acquired using a BD FACSLyric (BD Biosciences Pharmingen, San Diego, CA, USA) and analyzed using FlowJo software version 10.7.2 (BD Biosciences Pharmingen).

(Figures)

Figure 8d

(Figure legends, Line 1006-)

Fig. 8. Distribution of cells in lymphatic node-like structure.

d, Proportion of various immune cell populations in lymph nodes or lymph node-like structures of hLEC+hMSC tissue-transplanted LD mice (days 7, 14, and 21). CD3⁺ cells, CD19⁺ cells, CD11c⁺ cells, and F4/80⁺ cells were gated by SSC and FSC. Data represent the mean ± standard deviation (n=3).

As many cell therapies failed due to the lack of revascularization of the transplant. Does better results can be observed with VEGFC-overexpressing MSC?

Response:

We appreciate the reviewer's attractive comment. We believe that VEGF-C, a lymphangiogenesis factor overexpressed by MSCs, is a promising approach for treating lymphedema. It is possible that expressing VEGF-C in MSCs can improve lymphangiogenesis and angiogenesis. Because angiogenesis is very important for cell therapies, we plan to conduct further studies, including the expression of VEGF-C into MSCs.

Corresponding revisions have been appropriately incorporated into the Discussion section, with changes marked in red in the revised manuscript.

(Discussion, Line 450-)

This study had some limitations. First, obtaining primary LECs from mice tissues is challenging. Because primary LECs can only be isolated and harvested in limited quantities from mouse tissues, human-derived primary LECs were purchased and used instead. Second,

immunodeficient mice were used in animal experiments, allowing CeLyTs composed of hMSCs and hLECs to survive and form lymph node-like structures in the mice. Finally, the validity of the disease model should be considered. In this study, lymphedema was induced by removing the lymph nodes. However, in limb lymphedema models, bypass pathways for lymphatic fluid can form easily, leading to a gradual reduction in edema over time [49]. Consistent with this, our results also showed spontaneous edema recovery about 50 days after lymph node removal (Supplementary Fig. 11c). Because of these experimental limitations, it is unclear whether this treatment method will be sufficient to treat lymphedema. Therefore, further investigations are needed to develop more effective therapeutic strategies, such as combining CeLyTs transplantation with additional interventions, including pharmacological treatments, lymphovenous anastomosis bypass surgery, or the transplantation of VEGF-C overexpressing MSCs.

Is the system drug-sensitive?

Response:

We appreciate the reviewer's helpful comment. Lymph node-like structures produced various proinflammatory cytokines by administering CpG1018, a nucleic acid that has the ability to activate innate immunity (Figure 7d). Based on the result, we think that the lymph node-like structures may be drug-sensitive.

What is the impact on adipose tissue accumulation?

Response:

We appreciate the reviewer's meaningful comment. We have performed additional experiments and included images of the subcutaneous adipose tissue in Supplementary Figure 8d. The results confirm that the amount of subcutaneous adipose tissue is reduced by the transplantation of CeLyTs. Additionally, transplantation of suspended MSCs also reduced adipose tissue accumulation. The suppressive effect of CeLyTs on adipose tissue accumulation is likely due to the anti-inflammatory effect of the MSCs contained within CeLyTs.

Corresponding revisions have been appropriately incorporated into the Results, Discussion, and Supplementary Figure legends sections, with changes marked in red in the revised manuscript.

(Results, Line 264-)

The adipose tissue accumulation in the skin was also reduced by transplantation of the CeLyTs (Supplementary Fig. 8d).

(Discussion, Line 371-)

CeLyTs had a suppressive effect on the hallmarks of lymphedema induced by chronic inflammation (Supplementary Fig. 8). Suspended MSC transplantation also showed a tendency to improve skin thickening and adipose tissue accumulation (Supplementary Fig. 8a,b,d), indicating the immunoregulatory ability of MSCs may also contribute to improving inflammation associated with lymphedema by the transplantation of CeLyTs.

(Supplementary Figures)

Supplementary Figure 8d

(Supplementary Figure legends, Line 263)

Supplementary Fig. 8. | Suppressive effect on lymphedema hallmarks after transplantation of bioengineered tissues with lymphatic network in LD.

d, Paraffin-section images of adipose tissue under the skin of suspended cells- and bioengineered tissue-transplanted LD mice (day 21).

Finally, the authors overestimate their conclusions as, so far, the experiments have only been conducted in immunodeficient mice due to the human origin of the transplanted cells.

Response:

We appreciate the reviewer's helpful comment. As you pointed out, our study was conducted using immunodeficient mice, and the results may therefore be overestimated. Consequently, we have added a discussion of the limitations of our study to the manuscript. We also slightly weakened the conclusion to reflect this consideration.

Corresponding revisions have been appropriately incorporated into the Discussion section, with changes marked in red in the revised manuscript.

(Discussion, Line 450-)

This study had some limitations. First, obtaining primary LECs from mice tissues is challenging. Because primary LECs can only be isolated and harvested in limited quantities from mouse tissues, human-derived primary LECs were purchased and used instead. Second, immunodeficient mice were used in animal experiments, allowing CeLyTs composed of hMSCs and hLECs to survive and form lymph node-like structures in the mice. Finally, the validity of the disease model should be considered. In this study, lymphedema was induced by removing the lymph nodes. However, in limb lymphedema models, bypass pathways for lymphatic fluid can form easily, leading to a gradual reduction in edema over time [49]. Consistent with this, our results also showed spontaneous edema recovery about 50 days after lymph node removal (Supplementary Fig. 11c). Because of these experimental limitations, it is unclear whether this treatment method will be sufficient to treat lymphedema. Therefore, further investigations are needed to develop more effective therapeutic strategies, such as combining CeLyTs transplantation with additional interventions, including pharmacological treatments, lymphovenous anastomosis bypass surgery, or the transplantation of VEGF-C overexpressing MSCs.

(Discussion, Line 472-)

Conversely, CeLyTs resulted in the reconstruction of a lymph node-like structure and long-term suppression of lymphedema, exhibiting a high therapeutic effect compared with conventional treatments over 100 days (Supplementary Fig. 11).

Also, simple lymph node transplant is not enough to cure lymphedema because the condition involves more than just the absence of lymph nodes; it is a complex disease involving the entire lymphatic system and the surrounding tissues. The authors should discuss about the existing lymphatic vessels that are non-functional or severely damaged. In that context the transplanted nodes may not integrate effectively into the system.

Over time, lymphedema causes chronic changes in the tissues, such as fibrosis (thickening and scarring), fat deposition, and inflammation. These changes reduce tissue elasticity and lymphatic function, making it harder for transplanted nodes to address the underlying problem. Even if some drainage is restored, these structural changes may persist and contribute to ongoing swelling and complications. The authors should discuss this point.

Response:

We appreciate the reviewer's insightful comment. As you pointed out, lymphedema can lead to chronic tissue changes because it affects the entire lymphatic system and surrounding tissues. Therefore, simply regenerating lymph nodes may not be sufficient. However, chronic changes in the surrounding tissues will continue to occur with conventional standard treatments, such as lymphatic drainage and compression therapy, due to the absence of functional lymph nodes. In contrast, in this study, we succeeded in forming a structure that functions as a lymph node, and it continued to function without ongoing swelling or complications even after 100 days (Supplementary Fig. 11). This result suggests the potential for long-term effects with a single transplant. Additionally, the lymph node-like structures impacted the hallmarks of lymphedema, including skin thickening, adipose tissue accumulation, and chronic inflammation, and suppressed them to levels comparable to those in normal mice (Supplementary Fig. 8). However, due to experimental limitations, further studies are needed to explore better methods to enhance the therapeutic effects and combinations with drugs.

Corresponding revisions have been appropriately incorporated into the Discussion section, with changes marked in red in the revised manuscript.

(Discussion, Line 450-)

This study had some limitations. First, obtaining primary LECs from mice tissues is challenging. Because primary LECs can only be isolated and harvested in limited quantities from mouse tissues, human-derived primary LECs were purchased and used instead. Second, immunodeficient mice were used in animal experiments, allowing CeLyTs composed of hMSCs and hLECs to survive and form lymph node-like structures in the mice. Finally, the validity of the disease model should be considered. In this study, lymphedema was induced by removing the lymph nodes. However, in limb lymphedema models, bypass pathways for lymphatic fluid can form easily, leading to a gradual reduction in edema over time [49]. Consistent with this, our

results also showed spontaneous edema recovery about 50 days after lymph node removal (Supplementary Fig. 11c). Because of these experimental limitations, it is unclear whether this treatment method will be sufficient to treat lymphedema. Therefore, further investigations are needed to develop more effective therapeutic strategies, such as combining CeLyTs transplantation with additional interventions, including pharmacological treatments, lymphovenous anastomosis bypass surgery, or the transplantation of VEGF-C overexpressing MSCs.

The authors should also discuss about the advantages compare to lymph node transplantation, especially in combination with other treatments like vascularized lymph node transfer (VLNT) or lymphovenous bypass surgery, that is not yet a standalone cure.

Response:

We appreciate the reviewer's helpful comment. This treatment method has limitations due to experimental constraints. Therefore, we must consider the possibility that it could be improved by combining it with lymphovenous anastomosis bypass surgery.

Corresponding revisions have been appropriately incorporated into the Discussion section, with changes marked in red in the revised manuscript.

(Discussion, Line 450-)

This study had some limitations. First, obtaining primary LECs from mice tissues is challenging. Because primary LECs can only be isolated and harvested in limited quantities from mouse tissues, human-derived primary LECs were purchased and used instead. Second, immunodeficient mice were used in animal experiments, allowing CeLyTs composed of hMSCs and hLECs to survive and form lymph node-like structures in the mice. Finally, the validity of the disease model should be considered. In this study, lymphedema was induced by removing the lymph nodes. However, in limb lymphedema models, bypass pathways for lymphatic fluid can form easily, leading to a gradual reduction in edema over time [49]. Consistent with this, our results also showed spontaneous edema recovery about 50 days after lymph node removal (Supplementary Fig. 11c). Because of these experimental limitations, it is unclear whether this treatment method will be sufficient to treat lymphedema. Therefore, further investigations are needed to develop more effective therapeutic strategies, such as combining CeLyTs transplantation with additional interventions, including pharmacological treatments, lymphovenous anastomosis bypass surgery, or the transplantation of VEGF-C overexpressing MSCs.

Minor points:

What is the origin of hLEC? It is not described in the method.

Response:

We apologize for the lack of information. The primary cultured human LECs were isolated from a single Caucasian male (2 years old) donor. Commodity and lot number information has also been added.

Corresponding revisions have been appropriately incorporated into the Online Methods section, with changes marked in red in the revised manuscript.

(Online Methods, Line 481-)

Materials and animals

The mMSC line C3H10T1/2 was kindly provided by Dr. Hiroki Kagawa (Kyoto, Japan). The mouse fibroblast cell line NIH3T3 (fibroblast) was obtained from RIKEN Bioresource Center (RIKEN BRC) (Ibaraki, Japan). Human primary adipose-derived stem cells (hMSC) were kindly provided by Rohto Pharmaceutical Co., Ltd. (Osaka, Japan). **The hMSCs were isolated from adipose tissue from a single donor.** Human primary lymphatic endothelial cells (hLEC) were purchased from Takara Bio Inc. (Tokyo, Japan, C-12216, lot: 467Z001.2). **The hLECs were isolated from juvenile foreskin (different locations) from a single donor.** All other chemicals used were of the highest commercially available grade. Male nude BALB/c Slc-nu/nu mice (5–8 weeks old) were purchased from Sankyo Labo Service Co., Inc. (Tokyo, Japan) and maintained under specific pathogen-free conditions. All animals were treated under anesthesia using isoflurane, and euthanized by excess anesthesia after experiments. The protocols for animal experimentations were conducted in accordance with the Institutional Animal Experimentation Committee of the Tokyo University of Science (latest approval number: Y23004, latest approval date: August 8, 2024). All animal experimentations were conducted in accordance with the principles and procedures outlined in the National Institutes of Health Guide for the Care and Use of Laboratory Animals and the ARRIVE guidelines. **The sex of the cells and animals was not considered in the study design of any experiment because it was thought to not affect the results.**

Why some parts of the manuscript are written in a red color?

Response:

We apologize for the parts before the revision. The parts in red color are the parts that remained in the Editor stage. The red highlighting in the revised manuscript indicates changes during the revision stage.

Other revisions

1. English editing of the revised manuscript

Minor text corrections have been conducted throughout the English editing of the revised manuscript.

2. Update of Acknowledgements

This work was supported by a Grant-in-Aid for Scientific Research (B) from the Japan Society for the Promotion of Science (grant number 23H03749), Project Seeds A from the Japan Agency for Medical Research and Development (AMED), JST SPRING (grant number JPMJSP2151), and a Noguchi Shitagau Research Grant from the Noguchi Institute.

3. Dot plot representation of individual data points in the distribution

Fig. 3c, Fig. 4b, Fig. 5f, Fig. 5g, Fig. 7d, Fig 8d, Supplementary Fig. 2a, Supplementary Fig. 2b, Supplementary Fig. 2c, Supplementary Fig. 2f, Supplementary Fig. 8b, Supplementary Fig. 8c, Supplementary Fig. 10c, Supplementary Fig. 10d, and Supplementary Fig. 11c were modified.

4. Update of author addresses

The authors' institutional affiliations and contact details have been updated with changes marked in red in the revised manuscript to reflect the recent relocation of the campus as outlined below.

¹Laboratory of Cellular Drug Discovery and Development, Faculty of Pharmaceutical Sciences, Tokyo University of Science, 6-3-1 Nijuku, Katsushika, Tokyo 125-8585, Japan

²Laboratory of Biopharmaceutics, Faculty of Pharmaceutical Sciences, Tokyo University of Science, 6-3-1 Nijuku, Katsushika, Tokyo 125-8585, Japan

³Division of Oncology, Research Center for Medical Sciences, The Jikei University School of Medicine, Tokyo, Japan

*Corresponding author

Kosuke Kusamori, Ph.D.

Tel./Fax.: +81-3-5876-1747 (Ext. 6512)

E-mail address: kusamori@rs.tus.ac.jp

We hope that these revisions and corrections are satisfactory and that our paper is now acceptable for publication in *Nature Communications*.

Response to reviewers' comments

We sincerely appreciate the reviewers' valuable feedback and their recommendations for the publication of our manuscript. Their insightful comments prompted us to reexamine the critical role of the transplanted LECs in tissue reconstruction. In response, we conducted additional experiments and made substantial revisions to the manuscript. Below, we provide point-by-point responses to each of the reviewers' comments.

Reviewers' comments:

Reviewer #1:

As a reviewer, I have carefully re-evaluated the authors' response and revised manuscript, and I must respectfully note that a central concern remains insufficiently addressed.

In my original review, I explicitly requested that the authors perform cell tracking experiments using fluorescent labeling or an equivalent method to directly demonstrate that the transplanted lymphatic endothelial cells (LECs) contributed to the reconstructed tissue.

This level of evidence is a fundamental requirement in regenerative medicine for establishing causal relationships between transplanted cells and tissue regeneration.

However, the authors have only presented indirect observations using Prox-1 immunostaining, without conducting any direct tracking experiments for the transplanted LECs.

Moreover, the resolution of the immunofluorescence images is insufficient for reliable identification and localization of cells.

Therefore, the current data do not adequately support the authors' claim that transplanted LECs contributed to tissue reconstruction.

I strongly encourage the authors to understand the core of the comment and to respond with scientific rigor and sincerity, beyond a merely formal reply.

Response:

We appreciate the reviewer's insightful comment. We apologize for the time it took to establish a lentiviral vector system for introducing green fluorescent protein (GFP) into LECs. Using hMSCs and hLEC/GFP, we prepared CeLyTs and transplanted them into LD mice. To evaluate the contribution of the transplanted LECs to the formation of lymph node-like structures,

we directly tracked GFP fluorescence. Since transgenic mice were not available, host-derived LECs were identified indirectly through immunofluorescence staining. On day 7 after transplantation, GFP fluorescence derived from the transplanted LECs was present in the transplanted CeLyTs, and a few host-derived LECs were observed. By day 14, the host-derived LECs were more evident in the transplanted CeLyTs and co-localized and fused with the transplanted LECs. By day 21, both the host-derived and transplanted LECs connected to their extracellular matrix, and some had clearly fused. Notably, LECs originating from the transplanted CeLyTs were also found in areas distant from the transplantation site, where they were observed in close association with host-derived LECs. This suggests that the transplanted LECs infiltrated the surrounding tissue and contributed to the reconstruction of lymphatic vessels by forming a patchwork-like luminal structure of the host-derived and transplanted LECs. Additionally, host-derived LECs were rarely observed in hMSC tissue transplants without LECs. This suggests that their recruitment may be induced by cytokines such as vascular endothelial growth factor (VEGF)-A, VEGF-D, and interleukin (IL)-8, which are produced by MSCs, and are enhanced by co-culture with LECs. Furthermore, VEGFR-3 expression was increased in LECs within CeLyTs, indicating that transplanted LECs might contribute to the reconstruction of lymphatic vessels both *in vitro* and *in vivo* by acting as guide cells for the host-derived LECs under the influence of VEGF-C/D-derived from MSCs and others. In addition, LECs produce chemokines such as CCL21, which attract immune cells. These chemokines likely play a critical role in lymph node development, contributing to the formation of lymph node-like structures through coordinated actions of both transplanted and host-derived LECs. Therefore, the formation of lymphatic luminal structures composed of LECs may be necessary in the mechanism of lymph node-like structure formation. Taken together, these findings suggest that LECs derived from transplanted CeLyTs contribute to the formation of lymph node-like structures and the reconstruction of lymphatic vessels. Further studies are needed to elucidate the underlying mechanisms, particularly in the context of lymph node development.

Revisions corresponding to these findings have been incorporated into the Results, Discussion, Online Methods, Supplementary Methods, Supplementary Figures, and Supplementary Figure legends sections with changes marked in red in the revised manuscript.

(Results, Line 279)

To clarify the contribution of transplanted LECs in the formation of lymphatic vessels and lymph node-like structures, CeLyTs prepared using hLECs expressing green fluorescent protein (hLEC/GFP) and GFP fluorescence was tracked (Supplementary Fig. 9a). A few host-derived LECs were observed at day 7. By day 14, transplanted and host-derived LECs co-localized, and by day 21, they were connected through the extracellular matrix, with some evidence of cell fusion

(Supplementary Fig. 9b, c). In contrast, host-derived LECs were rarely observed in hMSC tissue transplants at day 21, highlighting the importance of LECs in CeLyTs for the formation of lymphatic vessels and lymph node-like structures (Supplementary Fig. 9d).

(Discussion, Line 458)

To evaluate the contribution of the transplanted LECs to the formation of the lymphatic vessels and lymph node-like structures, we established hLEC/GFP and directly tracked GFP fluorescence in the CeLyTs prepared using hMSCs and hLEC/GFP (Supplementary Fig. 9a). The transplanted LECs fused with the host-derived LECs to reconstruct lymphatic vessels (Supplementary Fig. 9b,c). Additionally, some lymphatic luminal structures were formed independently by the transplanted LECs, while others were formed collaboratively with host-derived LECs. These findings are consistent with previous reports and suggest that transplanted LECs play a critical role in the formation of the lymphatic vessels [49]. The host-derived LECs may be induced by cytokines such as VEGF-A, VEGF-D, and IL-8, which were reported to be produced by MSCs in co-culture with LECs [50]. On day 7 after transplantation, only a few host-derived LECs were observed, whereas by day 14, a larger number had appeared and were fused or connected with the transplanted LECs. Furthermore, VEGFR-3 was upregulated in the LECs within CeLyTs, suggesting that the transplanted LECs may contribute to lymphatic vessel reconstruction *in vitro* and *in vivo* by acting as guide cells for the host-derived LECs under the influence of VEGF-C/D derived from MSCs (Supplementary Fig. 2a,b). Regarding the formation of lymph node-like structures, transplanted LECs formed immature lymph node-like structures by forming lymphatic luminal structures together with host-derived LECs on day 14 after transplantation. This process likely facilitates immune cell infiltration, supporting the development of lymph node-like structures. In contrast, transplantation of MSC-only tissue did not induce sufficient LECs to form lymphatic vessels (Supplementary Fig. 9d), suggesting that lymph node-like structures are unlikely to form under such conditions. Recently, it has been reported that LECs induce immune cells by producing chemokines such as CCL21. The formation of lymphatic luminal structures composed of LECs may be important in the formation of lymph node-like structures [51, 52]. Therefore, chemokines such as CCL21 likely play a crucial role in lymph node development, influencing the formation of lymph node-like structures by both transplanted and host LECs. Based on this evidence, the long-term survival of LECs may be critical for successful lymph node regeneration. Further studies are needed to elucidate the functional roles of transplanted LECs in the formation of lymph node-like structures.

(Online Methods, Line 520)

Materials and animals

The mMSC line C3H10T1/2 was kindly provided by Dr. Hiroki Kagawa (Kyoto, Japan). The mouse fibroblast cell line NIH3T3 (fibroblast) was obtained from RIKEN Bioresource Center (RIKEN BRC) (Ibaraki, Japan). Human primary adipose-derived stem cells (hMSC) were kindly provided by Rohto Pharmaceutical Co., Ltd. (Osaka, Japan). The hMSCs were isolated from adipose tissue from a single donor. Human primary lymphatic endothelial cells (hLEC) were purchased from Takara Bio Inc. (Tokyo, Japan, C-12216, lot: 467Z001.2). The hLECs were isolated from juvenile foreskin (different locations) from a single donor. **Lenti-X 293T cells were purchased from Takara Bio Inc.** All other chemicals used were of the highest commercially available grade. Male nude BALB/c Slc-nu/nu mice (5–8 weeks old) were purchased from Sankyo Labo Service Co., Inc. (Tokyo, Japan) and maintained under specific pathogen-free conditions. All animals were treated under anesthesia using isoflurane, and euthanized by excess anesthesia after experiments. The protocols for animal experimentations were conducted in accordance with the Institutional Animal Experimentation Committee of the Tokyo University of Science (latest approval number: Y23004, latest approval date: August 8, 2024). All animal experimentations were conducted in accordance with the principles and procedures outlined in the National Institutes of Health Guide for the Care and Use of Laboratory Animals and the ARRIVE guidelines. The sex of the cells and animals was not considered in the study design of any experiment because it was thought to not affect the results.

Cell culture

mMSCs, firefly luciferase (fluc) gene-expressing mMSCs (mMSC/fluc), NanoLuc luciferase (Nluc) gene-expressing mMSCs (mMSC/Nluc), and green fluorescent protein (GFP) gene-expressing mMSCs (mMSC/GFP) cells were cultured in DMEM supplemented with 15% heat-inactivated FBS and 1% penicillin-streptomycin-L-glutamine solution. NIH3T3 **and Lenti-X 293T** cells were cultured in DMEM supplemented with 10% heat-inactivated FBS and 1% penicillin-streptomycin-L-glutamine solution. hMSCs were cultured in R-STEM Medium for hMSC High Growth (Rohto Pharmaceutical Co., Ltd.). hLECs **and hLEC/GFP** were cultured in Endothelial Cell Media MV2 (Takara Bio Inc.) and maintained in a humidified atmosphere containing 5% CO₂ at 37°C.

Fabrication of bioengineered lymphatic tissues by cell stacking

mMSCs, hMSCs, mMSC/GFP, mMSC/Nluc, carboxyfluorescein succinimidyl ester (CFSE)-labeled mMSCs (mMSC/CFSE), or NIH3T3 (7.5×10^5 cells) were collected with a trypsin-ethylenediaminetetraacetic acid (EDTA) solution (Nacalai Tesque Inc., Kyoto, Japan), seeded into 24-well Transwell inserts (Corning, NY, USA), and set into a culture plate. The culture plate with seeded cells was immediately centrifuged under various conditions (centrifugation speed from 50

×g to 1000 ×g and centrifugation time from 10 sec to 3 min) or 750 ×g and 30 sec to settle the cells at the bottom of the inserts. After 24 h-incubation, hLECs, hLEC/CFSE, hLEC/GFP, mMSC/Nluc, and mMSC/fluc (2×10^5 cells) were gently seeded on the layered MSCs and settled down on the layered cells in the inserts using the same procedure. After another 24 h-incubation, mMSCs, hMSCs, mMSC/GFP, and mMSC/Nluc or mMSC/CFSE (7.5×10^5 cells) were gently seeded on LECs and MSCs and settled down on the layered cells in the inserts using the same procedure. Then, the cells were cultured for 5 days with changes of medium.

New references

49. Landau, S., Newman, A., Edri, S., Michael, I., Shaul B. S., Shandalov, Y., Arye, T. B., Kaur, P., Zheng, M. H., & Levenberg, S. Investigating lymphangiogenesis in vitro and in vivo using engineered human lymphatic vessel networks. *Proc Natl Acad Sci U S A*. 118: e2101931118 (2021). doi: 10.1073/pnas.2101931118.
50. Nagahara, A. I., Homma, j., Ryu, B., Sekine, H., Higashi Y., Shimizu, T., & Kawamata, T. Networked lymphatic endothelial cells in a transplanted cell sheet contribute to form functional lymphatic vessels. *Sci Rep*. 12: 21698 (2022). doi: 10.1038/s41598-022-26041-0.
51. Alitalo, K., Tammela, T., & Petrova, T. V. Lymphangiogenesis in development and human disease. *Nature*. 438: 946-953 (2005). doi: 10.1038/nature04480.
52. Hu, Z., Zhao, X., Wu, Z., Qu, B., Yuan, M., Xing, Y., Song, Y., & Wang, Z. Lymphatic vessel: Origin, heterogeneity, biological functions and therapeutic targets. *Signal Transduct Target Ther*. 9: 9 (2024). doi: 10.1038/s41392-023-01723-x.

(Supplementary Methods, Line 27)

Establishment of hLEC/ green fluorescent protein (GFP)

Lenti-X 293T cells (1×10^7 cells) were seeded in a 10 cm culture dish and cultured overnight at 37 °C in humidified air containing 5% CO₂. Lenti-X 293T cells were co-transfected with the pLVSIIN-GFP plasmid and the Lentiviral Packaging Mix (Takara Bio Inc., Tokyo, Japan) by the calcium phosphate method. After 24 h, the medium was replaced with culture medium containing 10 μM forskolin, and cells were cultured for an additional 48 h. Then, the supernatant was collected and centrifuged at 900× g for 5 min. Furthermore, the supernatant was filtered, mixed with PEG solution (32 w/v % polyethylene glycol #6000, 400 mM NaCl, and 40 mM HEPES), and centrifuged at 2500× g for 40 min. The precipitated viruses were collected by Opti-MEM (Thermo Fisher Scientific, Inc., Waltham, MA, USA) and transduced into hLECs. After 24 h, the medium was replaced, and the cells were cultured in normal culture medium for a few days. These cells were cloned and incubated until they were confluent. hLEC/GFP were observed with BZ-X800 fluorescence microscope (Keyence, Osaka, Japan).

(Supplementary Figures)

Supplementary Figure 9

(Supplementary Figure legends, Line 285)

Supplementary Fig. 9. | Presence of transplanted LECs and their association with host-derived LECs in CeLyTs.

a, Fluorescence images of hLEC/GFP+hMSC tissue by centrifugal cell stacking technique (day 5). Blue, DAPI; Green, GFP. Scale bars, 100 μm . **b**, Representative immunofluorescence images of mouse Prox-1 (a lymphatic endothelial cell marker) and GFP derived from hLEC/GFP in lymph node-like structures formed in hLEC/GFP+hMSC tissue-transplanted LD mice (days 7, 14, and 21). Blue, DAPI; Green, GFP; Red, mouse-reactive Prox-1. Scale bars, 100 μm . **c**, Enlarged images in the white boxed areas of **b**. Blue, DAPI; Green, GFP; Red, mouse-reactive Prox-1. Scale bars, 30 μm . **d**, Representative immunofluorescence images of mouse Prox-1 (a lymphatic endothelial cell marker) in hMSC tissue-transplanted LD mice (day 21). Blue, DAPI; Red, mouse-reactive Prox-1. Scale bars, 100 μm .

Reviewer #2:

The authors adequately responded to the concerns of the reviewer. The quality of the manuscript has improved. I recommend the manuscript for publication.

Response:

We sincerely thank you for carefully reviewing our manuscript. Your thoughtful and precise suggestions have helped us improve it by revising the references, wording, and other aspects.

Reviewer #3:

The manuscript has been significantly improved thanks to the comprehensive set of experiments provided by the authors. The authors have carried out all the requested experiments and have addressed the points that required clarification. In conclusion, I give a favorable recommendation for the publication of this article.

Response:

We sincerely appreciate your thoughtful consideration of our manuscript. Your insightful comments encouraged us to conduct additional experiments, which allowed us to obtain essential data and revisit our study from a new perspective, thereby significantly enhancing the overall quality of the manuscript.

We hope that these revisions and corrections are satisfactory and that our paper is now acceptable for publication in *Nature Communications*.

Response to reviewers' comments

We sincerely thank the reviewer for their insightful and constructive feedback, which has significantly guided the refinement of our manuscript. We also appreciate the editor's additional guidance in prioritizing our revisions. In line with this guidance, we understand that certain concerns, specifically point 3 (direct tracking of transplanted LECs) and point 7 (functional validation), were considered adequately addressed in our previous revision and thus required no further changes. For points 4 (model chronicity) and 5 (spontaneous recovery), we have expanded the discussion to explicitly acknowledge the limitations of our model. For the remaining points, including point 6 (surgical methodology) as well as those related to improved data presentation and higher-resolution imaging, we conducted a series of additional experiments and made substantial revisions. Below, we provide a point-by-point response to each reviewer's comment.

Reviewers' comments:

Reviewer #1:

Thank you for your revisions. However, several critical issues remain unaddressed:

1. Insufficient structural evidence of lymph node reconstruction

Key lymph node features (cortex, medulla, follicles, sinuses) are not demonstrated by histology or immunostaining (e.g., CD20, LYVE-1). Additional 3D imaging or higher-resolution sections are required.

Response:

We appreciate the reviewer's insightful comment. In response to the reviewer's request for more detailed structural characterization, we performed additional histological and immunofluorescence analyses of the reconstructed lymph node-like structure. Specifically, we obtained high-magnification hematoxylin and eosin images and conducted immunostaining for CD3 and B220 to assess T and B cell localization, and for CD11b and LYVE-1 to evaluate macrophage distribution and lymphatic structures. These analyses demonstrated several lymph node-like features, including lymphoid follicles and sinus-like spaces. However, follicles were occasionally observed within medullary regions, and the medulla and cortex were not completely segregated. T and B cells were intermingled without sharply demarcated T and B cell zones characteristic of mature lymph nodes. Macrophages were abundant in central regions, supporting the presence of a medulla. Lymphatic structures were distributed throughout the tissue rather than confined to the subcapsular sinus. Taken together, these findings indicate that the reconstructed tissue possesses multiple lymph node-like features but remain structurally immature compared

with fully developed lymph nodes. Although the structure is not yet fully developed, the organization already demonstrates functional activity, and further optimization of engraftment periods, as well as the use of additional stimulatory factors, may promote the development of a more mature lymph node-like architecture in future studies.

Revisions corresponding to these findings have been appropriately incorporated into the Results, Online Methods, Supplementary Figures, and Supplementary Figure legends sections, with changes marked in red in the revised manuscript.

(Results, Line 306)

Therefore, high-magnification hematoxylin and eosin staining of the reconstructed lymph node-like structures revealed distinct lymphoid structures, including lymphoid follicles and sinus-like spaces. Immunofluorescence staining for CD3, B220, CD11b, and LYVE-1 (LECs) demonstrated partial segregation of T cell and B cell zones; however, both cell types were intermingled, and the boundary between the cortex and the medulla was not clearly defined. Within a region consistent with the medulla, macrophages were abundantly distributed, supporting the presence of a medulla-like structure. Lymphatic structures were broadly distributed throughout the tissue. These findings indicate that the reconstructed tissues possess key features of a lymph node, such as follicles, sinuses, and a medulla region, but remain structurally immature compared with mature lymph nodes (Supplementary Fig. 11a-c).

(Online Methods, Line 652)

Immunohistochemical staining

Bioengineered tissues and cryo-sectioned tissues were fixed with 4% PFA, permeabilized with methanol, dimethyl sulfoxide, and a hydrogen peroxide mixture (6:1:1), and blocked with 1% bovine serum albumin for immunofluorescence staining. Bioengineered tissues and cryo-sectioned tissues were stained using primary antibodies and corresponding secondary antibodies. Nuclei were stained with Vectashield Antifade Mounting Medium with DAPI (Vector Laboratories Inc., Burlingame, CA, USA). Immunofluorescence images were captured using a Leica SP8 laser scanning confocal microscope with a LAS X Life Science software (Leica Microsystems). The primary and secondary antibodies were as follows: 1:200 PROX-1 (Proteintech, Rosemont, IL, USA, 11067-2-AP), 1:500 mouse PROX-1 (Abcam, Cambridge, UK, ab101851), 1:500 human PROX-1 (Bio-Techne, Minneapolis, MN, USA, AF2727), 1:500 CD31 (Bio-Techne, NB600-1475), 1:500 mouse CD31 (Abcam, ab256569), 1:500 human CD31 (Abcam, ab76533), 1:100 B220/CD45R (Bio-Techne, NBP2-53303), 1:100 CD3 (Arigo, Zhubei, Taiwan, ARG22819), 1:50 CD11b (Abcam ab8878), 1:50 CD11c (Thermo Fisher Scientific, PA5-90208), 1:500 Collagen Type I (Proteintech, 14695-1-AP), 1:400 LYVE-1 (Thermo Fisher

Scientific, BS-1311R), 1:500 VEGFR-3 (Thermo Fisher Scientific, BS-1083R), 1:50 HuNuC (Sigma-Aldrich, MAB1281C3), 1:400 CD45 (Thermo Fisher Scientific, 56-0451-82), 1:100 CD3 (Bio-Techne, FAB4841G), 1:100 CD19 (Bio-Techne, NBP2-24965AF647), 1:100 F4/80 (Cell Signaling Technology Inc., Danvers, MA, USA, 52267), 1:200 CD11c (Bio-Techne, FAB69501R), 1:500 anti-rabbit Alexa 488 (Thermo Fisher Scientific, A21206), 1:500 anti-rat Alexa 488 (Thermo Fisher Scientific, A-11006), 1:500 anti-rabbit Alexa 647 (Thermo Fisher Scientific, A31573), and 1:500 anti-goat Alexa 647 (Thermo Fisher Scientific, A21447).

(Supplementary Figures)
Supplementary Figure 11

a

b

c

(Supplementary Figure legends, Line 340)

Supplementary Fig. 11. | Structural analysis of lymph node-like structures after transplantation of CeLyTs.

a, Paraffin-section images of a medulla and a cortex region in a lymph node-like structure formed in hLEC+hMSC tissue-transplanted LD mice (day 21). Scale bars, 200 μ m. Black arrows, sinuses; white arrows, follicles. **b**, Representative immunofluorescence staining for CD3 and B220 in a lymph node-like structure formed in hLEC+hMSC tissue-transplanted LD mice (day 21). Blue, DAPI; Green, CD3; Red, B220. Scale bars, 1 mm (upper panel) and 200 μ m (lower panel). **c**, Representative immunofluorescence staining for CD11b and LYVE-1 in a lymph node-like structure formed in hLEC+hMSC tissue-transplanted LD mice (day 21). Blue, DAPI; Green, CD11b; Red, LYVE-1. Scale bars, 1 mm (upper panel) and 200 μ m (lower panel).

2. Optical quality and raw data of GFP/Prox-1 images

Supplementary Figure 9 remains out of focus, preventing clear visualization of nuclear contours and lymphatic lumina. Please re-acquire images with a high-NA objective, appropriate z-stacks, and provide the unprocessed TIFF stacks as Source Data.

Response:

We sincerely thank the reviewer for pointing out this important issue and apologize for the insufficient image quality in the original Supplementary Figure 9, which hindered clear visualization of nuclear contours and lymphatic lumina. We have replaced the images in Supplementary Figure 9b, c (Day 21) with higher-quality images. We specifically chose Day 21 for replacement because lymphatic lumina were most clearly observed at this time point. After immunostaining the sections for Prox-1, we re-acquired the images using a confocal laser scanning microscope. For the higher-magnification views, we employed a water-immersion high-NA objective to maximize resolution and light collection efficiency, and z-stack acquisition was performed at optimized intervals. The scan speed was set to a low rate to further enhance image quality and reduce noise. These new images clearly visualize nuclear contours and lymphatic lumina. The unprocessed TIFF stacks have been deposited as Source Data.

Revisions corresponding to these findings have been appropriately incorporated into the Supplementary Figures section with changes marked in red in the revised manuscript.

(Supplementary Figures)

Supplementary Figure 9b

Supplementary Figure 9c

3. Lack of direct tracking of transplanted LECs

Reliance on Prox-1 staining alone is insufficient. Direct fluorescent or genetic cell-tracking experiments are needed to confirm that transplanted LECs integrate and function within the reconstructed tissue.

Response:

We appreciate the reviewer's important suggestion regarding direct tracking of transplanted LECs. However, based on the editorial guidance stating that "Our assessment has determined that points 3 (direct tracking of transplanted LECs) and 7 (functional validation) have already been addressed. There is no need to conduct any revisions to address these points", we have not made further revisions to the manuscript for this comment.

4. Chronic lymphedema model validity

A 42-day follow-up without fibrosis markers (Masson's trichrome, α -SMA) or observation beyond eight weeks does not convincingly model chronic lymphedema. Please include additional markers and extend the follow-up period.

Response:

We thank the reviewer for their thoughtful comment regarding the chronic lymphedema model. We fully agree that the use of established fibrosis markers such as Masson's trichrome staining or α -SMA immunohistochemistry, along with longer-term follow-up (beyond 8 weeks), would further reinforce the chronicity of the model. While we acknowledge the importance of these additional experiments, they were not performed in the present study due to constraints related to experimental duration and scope, and we therefore recognize this as a limitation. Nevertheless, we believe our model reflects several key pathological hallmarks associated with chronic lymphedema within the 42-day observation period, based on the following evidence. First, in the LD mice, created by resecting the inguinal and popliteal lymph nodes, we observed features consistent with chronic lymphedema at day 21. These included persistent skin thickening, adipose tissue accumulation, and altered immune cell populations in the dermis, as demonstrated by H&E staining and flow cytometric analysis (Supplementary Fig. 8). Notably, there was a reduction in dermal T cell numbers and an increase in macrophages, consistent with a chronic inflammatory microenvironment reported in previous studies. It should be noted that this LD model is considered less stringent than the chronic lymphedema model. Therefore, while the presence of chronic inflammatory features in the LD model by day 21 may suggest the possibility of similar or more advanced pathological changes, including fibrosis, occurring in the chronic model by day 42, direct evidence for such progression was not obtained in this study. While we did not directly perform Masson's trichrome or α -SMA staining in this study, the presence of sustained tissue remodeling and immune dysregulation supports the notion of a chronic lymphedema state within the 42 day observation period. Nevertheless, we fully acknowledge that the lack of direct fibrosis markers and follow-up beyond 6 weeks represents an important limitation of the current study. In response to the editor's advice that points 4 and 5 can be addressed through clearly discussing any limitations of the model, we have incorporated a thorough discussion of these limitations in the revised manuscript. Due to constraints related to experimental duration and scope, we did not perform additional fibrosis-specific staining or extend the follow up period in this study. Instead, we have addressed these issues through a candid discussion emphasizing the need for future studies to include extended observation times and fibrosis assessments to more robustly characterize the chronicity and fibrotic progression of the model.

Revisions corresponding to these findings have been incorporated into the Discussion

section with changes marked in red in the revised manuscript.

(Discussion, Line 509)

Finally, our chronic lymphedema model involves resection of three key lymph nodes (iliac, inguinal, and popliteal) to suppress both superficial and deep lymphatic drainage, aiming to induce a more persistent and clinically relevant form of lymphedema. While this triple-node resection strategy offers a reproducible and scalable model, it may not completely prevent the development of compensatory lymphatic bypasses or spontaneous lymphatic regeneration, potentially leading to partial edema resolution over time. Alternative approaches, such as local irradiation or fascial dissection, have been used to more stringently inhibit lymphangiogenesis, but these techniques are technically challenging and require specialized equipment not always available in preclinical settings. Thus, although our surgical model balances technical feasibility with chronicity, the possibility of lymphatic rerouting and spontaneous recovery remains a limitation. Accordingly, our interpretation of the current findings is deemed to be valid only within the constraints of this model, and we have sought to avoid overstatement by acknowledging its inherent limitations. Future studies combining additional interventions, including pharmacological treatments, lymphovenous anastomosis surgery, or the transplantation of VEGF-C overexpressing MSCs, may be required to more definitively suppress lymphatic regeneration and better mimic chronic lymphedema.

5. Confounding by spontaneous recovery

Edema resolves spontaneously around day 50. Without irradiation or fascial blockade, therapeutic effects may be overestimated. A modified model to suppress spontaneous lymphatic regrowth should be considered.

Response:

We deeply appreciate the reviewer's thoughtful observation. Indeed, we recognize that in the mouse hindlimb lymphedema model, spontaneous resolution of edema, particularly after approximately 6 to 7 weeks, can occur due to endogenous lymphangiogenesis, which may lead to an overestimation of therapeutic effect when measures to suppress lymphatic regrowth are not employed. In this study, we selected a non-irradiated model to enable reproducible and technically feasible evaluation of lymphatic repair within a moderate timeframe. However, we fully acknowledge that without additional interventions such as irradiation or fascial barrier application, spontaneous lymphangiogenesis is not entirely suppressed. This limitation can make it challenging to distinguish true treatment effects from natural recovery, especially beyond day 50. In response to the editor's advice that points 4 and 5 can be addressed through clearly discussing any limitations of the model, we have revised the manuscript to incorporate a more detailed discussion of this issue. Accordingly, our current findings should be interpreted within the context of this limitation, with the understanding that the observed therapeutic effects may partly reflect the inherent regenerative capacity of the model. We also emphasize the importance of developing or employing more stringent models, such as those incorporating irradiation or surgical fascial modification, for future validation of long-term therapeutic efficacy. We greatly value the reviewer's suggestion, as it highlights a key consideration in model design for translational lymphedema research.

Revisions corresponding to these findings have been incorporated into the Discussion section with changes marked in red in the revised manuscript.

(Discussion, Line 509)

Finally, our chronic lymphedema model involves resection of three key lymph nodes (iliac, inguinal, and popliteal) to suppress both superficial and deep lymphatic drainage, aiming to induce a more persistent and clinically relevant form of lymphedema. While this triple-node resection strategy offers a reproducible and scalable model, it may not completely prevent the development of compensatory lymphatic bypasses or spontaneous lymphatic regeneration, potentially leading to partial edema resolution over time. Alternative approaches, such as local irradiation or fascial dissection, have been used to more stringently inhibit lymphangiogenesis, but these techniques are technically challenging and require specialized equipment not always

available in preclinical settings. Thus, although our surgical model balances technical feasibility with chronicity, the possibility of lymphatic rerouting and spontaneous recovery remains a limitation. Accordingly, our interpretation of the current findings is deemed to be valid only within the constraints of this model, and we have sought to avoid overstatement by acknowledging its inherent limitations. Future studies combining additional interventions, including pharmacological treatments, lymphovenous anastomosis surgery, or the transplantation of VEGF-C overexpressing MSCs, may be required to more definitively suppress lymphatic regeneration and better mimic chronic lymphedema.

6. Opaque surgical methodology

The protocol for iliac lymph-node excision lacks detail: incision lines, tissue planes, ligation of vessels/lymphatics, closure steps, and intraoperative photos or schematics are missing. Immediate Evans Blue or NIRF mapping data confirming complete node removal and flow blockade should be provided.

Response:

We thank the reviewer for this important suggestion to enhance the transparency and reproducibility of our iliac lymph node excision protocol, and apologize for the insufficient methodological detail in the original manuscript. In response, we have revised the Supplementary Methods section to provide detailed descriptions of the surgical procedure, including the incision site, dissection planes, and wound closure. Furthermore, we have added representative intraoperative photographs to the revised Supplementary Figures to visually document the procedural steps. To validate the efficacy of lymph node removal and lymphatic flow blockade, we additionally performed *in vivo* imaging following footpad injection of ICG, as it provides real-time, sensitive visualization of lymphatic flow. The ICG signals did not reach the iliac region postoperatively, confirming successful excision and flow interruption. We believe this provides sufficient real-time functional validation.

Revisions corresponding to these findings have been incorporated into the Results, Supplementary Methods, Supplementary Figures, and Supplementary Figure legends sections with changes marked in red in the revised manuscript.

(Results, Line 250)

CeLyTs also exhibited therapeutic effects in a more severe chronic lymphedema model, which was prepared by removing the iliac, inguinal, and popliteal lymph nodes (Supplementary Fig. 7a).

(Supplementary Methods, Line 110)

Therapeutic effect of CeLyTs on chronic lymphedema model mice. hLEC/hMSC spheroids were prepared by hLECs (2×10^5 cells) and hMSCs (1.5×10^6 cells) were seeded in 96 well ultra-low attachment plate (ThermoFisher Scientific) and incubated for 5 days. Chronic lymphedema model mice were induced by removing the iliac, popliteal, and inguinal lymph nodes in the right lower limbs of mice as previously described [53]. Under isoflurane anesthesia, a ~1.5 cm skin incision was made along the inguinal crease. Subcutaneous tissue and fascia were gently dissected to expose the lymph nodes. Vascular and lymphatic structures were carefully separated but not ligated, as bleeding and lymphorrhea were minimal and self-limiting in this model. The surgical site was irrigated with saline and closed using interrupted sutures. To confirm complete lymph

node removal and lymphatic flow disruption, 1 mg/mL indocyanine green (ICG; MP Biomedicals, Irvine, CA, USA) was injected into the right footpad immediately after surgery, and lymphatic flow was assessed using the IVIS Lumina III (PerkinElmer). Bioengineered tissues were transplanted to the site of the removed popliteal lymph node. For quantitative evaluation, the thickness of the paws and legs of the right lower limbs of LD mice with or without transplantation of bioengineered tissues were measured for 42 days and the change rate to day 0 was calculated.

(Supplementary Figures)

a

Popliteal lymph node (PO) removing procedure

Inguinal (IN) and iliac lymph node (IL) removing procedure

After lymph node removed imaging

(Supplementary Figure legends, Line 265)

Supplementary Fig. 7. | Therapeutic effect of bioengineered tissues with lymphatic network in chronic lymphedema model mice.

a, Surgical procedures for popliteal (PO), inguinal (IN), and iliac (IL) lymph node removal and corresponding Indocyanine green (ICG) imaging. For PO removal, ICG was injected into the footpad, and the lower limb was incised along the white dotted line (~1.5 cm). Guided by ICG imaging, the PO was identified and excised. Fluorescence in the PO region disappeared immediately after removal. Arrows: PO. For IN and IL removal, the IN, located at the confluence of three major vessels, was excised first. Arrows: IN. The peritoneum was then incised to expose the IL, which was resected under real-time ICG imaging guidance. Arrows: IL. The peritoneum was closed with nylon sutures, followed by skin closure. After lymph node removal, *ex vivo* imaging confirmed the removal of PO, IN, and IL. *In vivo* imaging immediately after removal demonstrated dispersal of ICG signal, consistent with interruption of lymphatic flow.

7. Absence of functional validation of lymph-node-like activity

No quantitative assay of inflow, filtration, or clearance (e.g., ICG or FITC-bead assays) is presented. Please include high-resolution, quantitative functional data demonstrating true lymph-node-like function.

Response:

We appreciate the reviewer's valuable comment regarding the functional validation of lymph-node-like activity. However, in accordance with the editorial guidance that "Our assessment has determined that points 3 (direct tracking of transplanted LECs) and 7 (functional validation) have already been addressed. There is no need to conduct any revisions to address these points", we have not made further revisions to the manuscript for this comment.

8.Improvement of data presentation

Provide wide-field images showing the position of the bioengineered tissue within the host and clarify the spatial relationship of Prox-1 relative to DAPI.

Response:

We thank the reviewer for this helpful suggestion to improve data presentation. In response, we acquired wide-field images that include the lymph node-like structure formed in hLEC+hMSC tissue-transplanted LD mice together with the surrounding host muscle tissue. These sections were stained for Prox-1 and DAPI to clarify the spatial relationship between Prox-1 and nuclei. Prox-1 positive cells were distributed across various regions of the tissue but were most abundant in the central area of the lymph node-like structure. The inclusion of surrounding muscle tissue in these images allows the anatomical position of the transplanted construct within the host to be clearly visualized.

Revisions corresponding to these findings have been incorporated into the Results, Supplementary Figures, and Supplementary Figure legends sections with changes marked in red in the revised manuscript.

(Results, Line 316)

Wide-field imaging confirmed the location of the lymph node-like structure within the host hindlimb, adjacent to skeletal muscle. LECs were distributed across various tissue compartments but were most densely localized in the central region of the lymph node-like structure, consistent with enrichment of the lymphatic lineage within the graft core (Supplementary Fig. 11d).

(Supplementary Figures)

(Supplementary Figure legends, Line 340)

Supplementary Fig. 11. | Structural analysis of lymph node-like structures after transplantation of CeLyTs.

d, Representative immunofluorescence staining for Prox-1 in a lymph node-like structure formed in hLEC+hMSC tissue-transplanted LD mice with the surrounding host muscle tissue (day 21). Blue, DAPI; Red, Prox-1. Scale bars, 1 mm. Within the white dot circle: lymph node-like structure.

We hope that these revisions and corrections are satisfactory and that our paper is now acceptable for publication in *Nature Communications*.